# Identification and characterization of early human photoreceptor states and cell-state-specific retinoblastoma-related features

Dominic WH Shayler[1,2], Kevin Stachelek[1,3], Linda Cambier[1], Sunhye Lee[1], Jinlun Bai[1,2], Bhavana Bhat[1], Mark W Reid[1], Daniel J Weisenberger[4,5], Jennifer G Aparicio[1], Yeha Kim[1], Mitali Singh[1], Maxwell Bay[2], Matthew E Thornton[6], Eamon K Doyle[7,8], Zachary Fouladian[1,2], Stephan G Erberich[7,8], Brendan H Grubbs[6], Michael A Bonaguidi[9], Cheryl Mae Craft[10,11], Hardeep P Singh[1,11], David Cobrinik[1,4,5,11]*

[1]The Vision Center, Department of Surgery, and Saban Research Institute, Children's Hospital Los Angeles, Los Angeles, United States; [2]Development, Stem Cell, and Regenerative Medicine Program, Keck School of Medicine, University of Southern California, Los Angeles, United States; [3]Cancer Biology and Genomics Program, Keck School of Medicine, University of Southern California, Los Angeles, United States; [4]Department of Cancer Biology, Keck School of Medicine, University of Southern California, Los Angeles, United States; [5]Norris Comprehensive Cancer Center, Keck School of Medicine, University of Southern California, Los Angeles, United States; [6]Maternal-Fetal Medicine Division of the Department of Obstetrics and Gynecology, Keck School of Medicine, University of Southern California, Los Angeles, United States; [7]Department of Radiology and The Saban Research Institute, Children's Hospital Los Angeles, Los Angeles, United States; [8]Department of Radiology, Keck School of Medicine, University of Southern California, Los Angeles, United States; [9]Department of Development, Stem Cell, and Regenerative Medicine, Keck School of Medicine, University of Southern California, Los Angeles, United States; [10]Department of Integrative Anatomical Sciences, Keck School of Medicine, University of Southern California, Los Angeles, United States; [11]USC Roski Eye Institute, Department of Ophthalmology, Keck School of Medicine, University of Southern California, Los Angeles, United States

*For correspondence:
dcobrinik@chla.usc.edu

## eLife Assessment

In this **important** paper, the authors use single-cell RNA sequencing to understand post-mitotic cone and rod developmental states and identify cone-specific features that contribute to retinoblastoma genesis. The authors report findings that have practical implications for retinal development, gene expression, and cell fate specification. The evidence is **compelling** as the experimental design and analysis are exceptionally rigorous.

**Abstract** Human cone photoreceptors differ from rods and serve as the retinoblastoma cell-of-origin, yet the developmental basis for their distinct behaviors is poorly understood. Here, we used deep full-length single-cell RNA-sequencing (scRNA-seq) to distinguish post-mitotic cone

and rod developmental states and identify cone-specific features related to retinoblastomagenesis. The analyses revealed nascent, immediately post-mitotic cone and rod precursors characterized by higher THRB or NRL regulon activities, immature and maturing cone and rod precursors with concurrent cone- and rod-related gene and regulon expression, and distinct early and late cone and rod maturation states distinguished by maturation-associated declines in RAX regulon activity. Cell-state-specific gene expression features inferred from full-length scRNA-seq were consistent with past 3' scRNA-seq analyses. Beyond the cell state characterizations, full-length scRNA-seq revealed that both L/M cone and rod precursors co-expressed *NRL* and *THRB* RNAs yet differentially expressed functionally antagonistic *NRL* isoforms and prematurely terminated *THRB* transcripts. Moreover, early L/M cone precursors sequentially expressed several lncRNAs along with *MYCN*, which composed the seventh most L/M-cone-specific regulon, and *SYK*, which was implicated in the cone precursors' proliferative response to *RB1* loss. These findings reveal previously unresolved photoreceptor precursor states and suggest a role for early cone-precursor-intrinsic *SYK* expression in retinoblastoma initiation.

## Introduction

Vertebrate photoreceptors develop from optic vesicle retinal progenitor cells (RPCs) through progressive RPC lineage restriction, fate determination, and post-mitotic maturation (*Shiau et al., 2021*; *Brzezinski and Reh, 2015*). While several transcription factors that govern these events have been identified, important aspects of photoreceptor fate determination and maturation remain unclear. For example, it is unclear if fate is determined in RPCs, where OTX2 and ONECUT1 are thought to control the post-mitotic expression of long- or medium-wavelength (L/M) cone determinant TRβ2 and rod determinant NRL (*Emerson et al., 2013*), or is determined in post-mitotic photoreceptor precursors with concurrent TRβ2 and NRL expression (*Ng et al., 2011*). Following fate commitment, post-mitotic developmental stages have been defined based on morphologic features and phototransduction-related gene or protein expression (*Hendrickson et al., 2012*; *Hoshino et al., 2017*), but it is unclear if progression through such stages is subdivided into distinct cell states governed by unique transcription factor combinations or represents a developmental continuum.

An improved understanding of photoreceptor development may provide insight into the pathogenesis of retinal dystrophies, retinal degenerations, and the retinal cone precursor cancer, retinoblastoma (*Georgiou et al., 2024*; *Cobrinik, 2024*). In the latter case, L/M cone precursors lacking the retinoblastoma protein (pRB) were shown to proliferate in a manner dependent on the L/M-cone lineage factors RXRγ and TRβ2 and the intrinsically highly expressed MDM2 and MYCN oncoproteins, likely representing the first step of retinoblastoma tumorigenesis (*Cobrinik, 2024*; *Xu et al., 2009*; *Xu et al., 2014*; *Singh et al., 2018*). Similarly, retinoblastoma cell proliferation depends on RXRγ, TRβ2, MDM2, and MYCN (*Xu et al., 2009*), implying that intrinsic L/M-cone factors contribute to the oncogenic state. However, retinoblastoma cells also express rod lineage factor *NRL* RNAs, which – along with other evidence – suggested a heretofore unexplained connection between rod gene expression and retinoblastoma development (*McEvoy et al., 2011*; *Khanna et al., 2006*). Improved discrimination of early photoreceptor states is needed to determine if co-expression of rod- and cone-related genes is adopted during tumorigenesis or reflects the co-expression of such genes in the retinoblastoma cell of origin.

The cone precursors' propensity to form retinoblastoma is a human-specific feature whose study requires analysis of developing human retina (*Cobrinik, 2024*). Single-cell RNA-sequencing (scRNA-seq) is well suited to such analyses as it enables discrimination of cell-type-specific fate determination and maturation features. scRNA-seq studies employing 3' end-counting have defined age-related post-mitotic transition populations, fate-determining features of post-mitotic photoreceptor precursors, and gene expression changes associated with the cone fate decision and early development (*Clark et al., 2019*; *Lu et al., 2020*; *Sridhar et al., 2020*; *Buenaventura et al., 2019*; *Lo Giudice et al., 2019*; *Lyu et al., 2021*; *Zuo et al., 2024*). However, 3' end-counting cannot be used to interrogate transcript isoforms, and the relatively low number of genes detected per cell limits the ability to distinguish closely related states in individual cells.

In this study, we sought to further define the transcriptomic underpinnings of human photoreceptor development and their relationship to retinoblastoma tumorigenesis. We generated deep,

full-length single-cell transcriptomes of human retinal progenitor cells (RPCs) and developing photoreceptors from fetal week (FW) 13–19 retinae, with enrichment of rare cone precursor populations, and applied long-read cDNA sequencing, RNA velocity, pseudotemporal trajectory reconstruction, and single-cell regulatory network inference and clustering (SCENIC) to interrogate individual cell states. These analyses discriminated previously unresolved photoreceptor developmental states, identified photoreceptor precursor states with cone and rod-related RNA co-expression, uncovered cell-type-specific expression of RNA isoforms of photoreceptor fate-determining genes, elucidated post-mitotic photoreceptor developmental trajectories, and revealed retinoblastoma cell-of-origin features that may contribute to retinoblastoma genesis.

## Results

### Regulon-defined RPC and photoreceptor precursor states

To interrogate transcriptomic changes during human photoreceptor development, dissociated RPCs and photoreceptor precursors were FACS-enriched from 18 retinae, ages FW13-19 (*Figure 1—figure supplement 1A*), and isolated using microfluidics or direct sorting into microliter droplets, followed by full-length cDNA synthesis, paired-end sequencing, and alignment to Ensembl transcript isoforms (*Figure 1A*). The FACS enrichment was based on a prior cone isolation method (*Xu et al., 2014*) but with wider gating to include rods and RPCs. After sequencing, we excluded all cells with <100,000 read counts and 18 cells expressing one or more markers of retinal ganglion, amacrine, and/or horizontal cells (*POU4F1, POU4F2, POU4F3, TFAP2A, TFAP2B, ISL1*) and concurrently lacking photoreceptor lineage marker *OTX2*. This yielded 794 single cells with averages of 3,750,417 uniquely aligned reads, 8278 genes detected, and 20,343 Ensembl transcripts inferred (*Figure 1—figure supplement 1A–C*). Sequencing batches were normalized and transcriptomes clustered and visualized in uniform manifold approximation and projection (UMAP) plots that integrated cells across different retinae, ages, isolation methods, and sequencing runs (*Figure 1B and C and Figure 1—figure supplement 1D–F*).

Low-resolution Louvain clustering (level 0.4) generated six clusters that segregated into mostly distinct UMAP domains (*Figure 1B*). One cluster was comprised of RPCs and Müller glia (MG), with specific expression of *LHX2, VSX2, SOX2,* and *SLC1A3*, while five clusters were comprised of cells with photoreceptor features, with wide expression of *OTX2* and *CRX* and cluster-specific rod- and cone gene expression (*Figure 1D and Figure 1—figure supplement 2*). In UMAP space, the cluster designated immature photoreceptor precursors (iPRPs) intermixed with the RPC/MG population, extended towards early L/M cone precursors, and was predominantly comprised of cells expressing the L/M cone determinant *THRB* (*Figure 1B and D*).

Two clusters highly expressing the rod determinant *NR2E3* were designated early-maturing rod (ER) and late-maturing rod (LR) (*Figure 1B*), based on the latter's increased expression of rod phototransduction genes *GNAT1, CNGB1, PDE6G, GNGT1,* and *RHO* (*Figure 1D and Figure 1— figure supplement 2C*). Few rods were detected at FW13, whereas both early and late rods were detected from FW15 to FW19 (*Figure 1C*), corroborating prior reports (*Lu et al., 2020*; *Zuo et al., 2024*). Differential expression analysis revealed other genes upregulated in late rods (*GNB1, SAMD7, NT5E*) (*Omori et al., 2017*; *Gagliardi et al., 2018*) as well as the downregulated *CRABP2, DCT,* and *FABP7* (*Figure 1—figure supplement 3A and Supplementary file 1A*). Genes upregulated in the LR cluster were enriched for photoreceptor and light sensing gene ontologies (*Figure 1—figure supplement 3B*) including spectrin binding, likely relating to proteins that control photoreceptor polarity and synapse formation (*Chen et al., 2009*; *Burger et al., 2021*), and purine biosynthesis and ribonucleotide metabolic processes, potentially related to the developing photoreceptors' high NAD$^+$ requirements (*Sokolov et al., 2021*).

Cones segregated into distinct S- and L/M-cone clusters, with differential expression of cone subtype markers (*OPN1SW, THRB*), previously identified S-cone enriched genes (*CCDC136, UPB1*; *Lukowski et al., 2019*; *Peng et al., 2019*; *Kallman et al., 2020*), novel S-cone enriched genes (*MEGF10, NRXN3, ACKR3*), and the L/M cone transcription factor *ISL2* (*Lu et al., 2020*), among others (*Figure 1B and D, Figure 1—figure supplement 3C, Supplementary file 1B*). Gene ontology analysis did not reveal relevant terms enriched in S cones, whereas L/M cones were enriched for protein translation related ontologies (*Figure 1—figure supplement 3D*) due to increased expression of ribosomal protein genes *RPL23A, RPLP0, RPS19, RPS27, RPS27A, RPS29, RPS3A* (*Supplementary file 1B*).

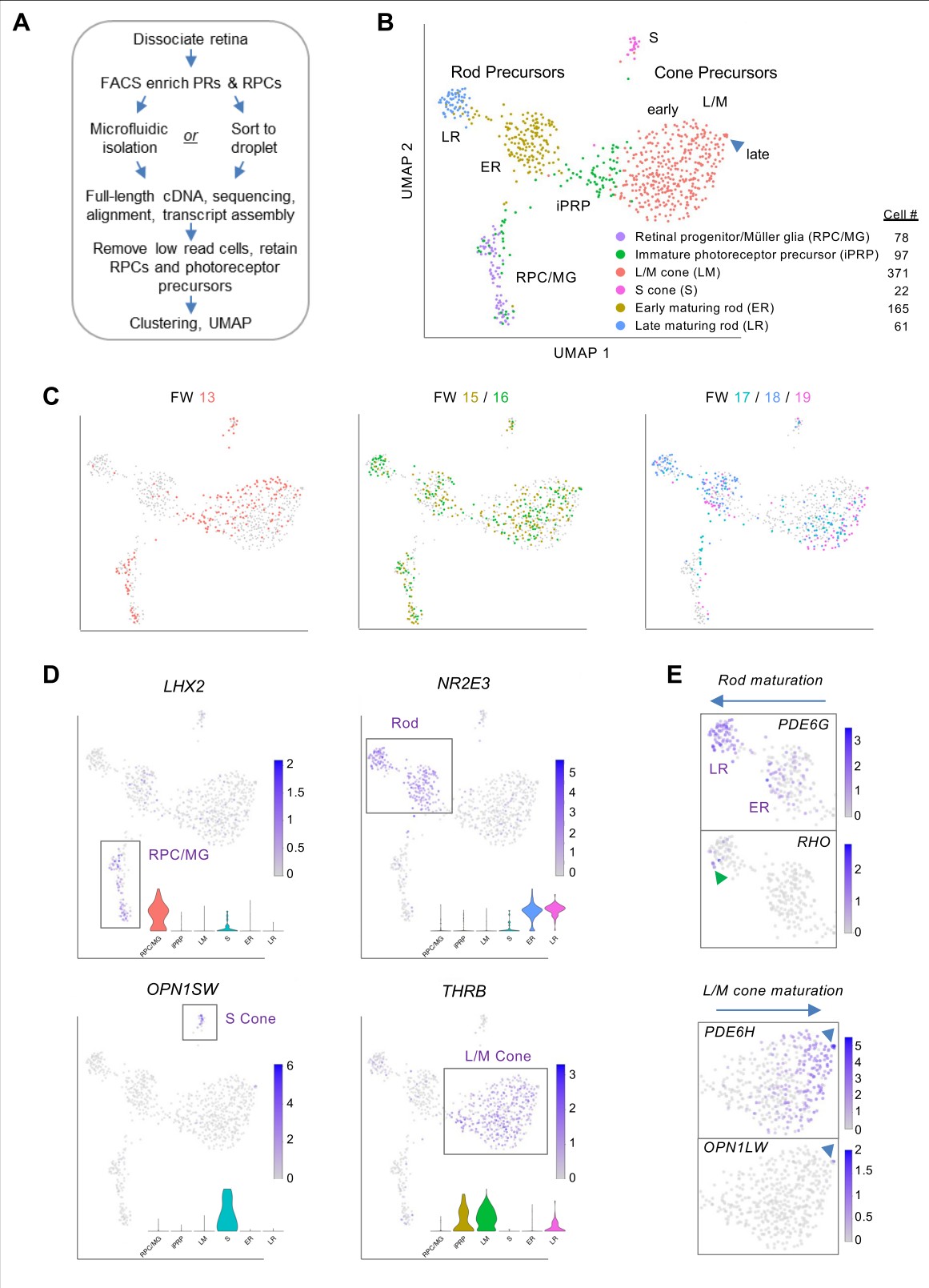

**Figure 1.** Photoreceptor-enriched full-length scRNA-seq of developing human retina. (**A**) Overview of sample collection and sequencing. (**B, C**) UMAP plots showing low-resolution cell type clusters (**B**) and ages (**C**). (**D**) Expression of marker genes for RPC/MGs (*LHX2*), rods (*NR2E3*), S cones (*OPN1SW*), L/M cones (*THRB*). *Insets*: Gene expression violin plots (from *left* to *right*): RPC/MG (red), iPRP (brown), LM cone (green), S cone (teal), early rod (blue), late rod (pink). (**E**) Expression of markers of rod maturation (*PDE6G*, *RHO*) and cone maturation (*PDE6H*, *OPN1LW*). Arrowheads: Late-maturing *RHO*⁺

*Figure 1 continued on next page*

Figure 1 continued

rods (*top*), late-maturing *OPN1LW*⁺ cones (*bottom*). See **Figure 1—figure supplement 2** for additional examples. UMAP and violin plots for any gene or transcript isoform can be produced at https://docker.saban.chla.usc.edu/cobrinik/app/seuratApp/.

The online version of this article includes the following figure supplement(s) for figure 1:

**Figure supplement 1.** scRNA-seq sample and sequencing summary.

**Figure supplement 2.** Expression of marker genes of RPCs, Müller glia, photoreceptors, rods, and cones.

**Figure supplement 3.** Differential expression between rod and cone maturation states.

In UMAP space, the *THRB*⁺ L/M-cone cluster segregated into a large 366 cell proximal population and a five-cell distal population inferred to represent early-maturing and late-maturing stages, respectively, based on the latter's increased expression of cone phototransduction genes *OPN1LW* (encoding L-opsin), *PDE6H*, and *GUCA1C* (**Figure 1E** and **Figure 1—figure supplement 3E and F**), analogous to *RHO*, *PDE6G*, and *GNGT1* upregulation in late rods. L/M cone precursors from different age retinae occupied different UMAP regions, suggesting age-related differences in L/M cone precursor maturation (**Figure 1C**). Compared to the early L/M population, late L/M cones had upregulation of three M-opsin genes, *TTR*, encoding the retinol-binding protein transthyretin, *PCP4*, a small protein that binds calmodulin previously noted in foveal cones (**Voigt et al., 2019**), and *MYL4*, a myosin light chain gene upregulated in retinal organoid L/M cones (**Kallman et al., 2020**), among others (**Figure 1—figure supplement 3E and F**, **Supplementary file 1C**). The low proportions of *OPN1MW*⁺ and *OPN1LW*⁺ late-maturing L/M cones are consistent with a prior analysis of similar-age retinae and with the further upregulation of these proteins in later maturation (**Lu et al., 2020**).

We next asked whether similar distinctions between early-maturing and late-maturing L/M cone and rod precursors were observed in prior studies. Indeed, a 3' single nucleus (sn) RNA-seq analysis of ~220,000 retinal cells from post-conception week (pcw) 8–23 distinguished the *cone precursor* versus *ML cone* and the *rod precursor* versus *rod* subclasses (**Zuo et al., 2024**). (*N.b.*, we retain the '*pcw*' and '*ML cone*' terms of Zuo et al. when describing their data and the synonymous '*FW*' and '*L/M cone*' for our data to maintain continuity with past publications.) The Zuo et al. cone and rod precursor versus cone and rod photoreceptor comparisons were not strictly analogous to our early-maturing versus late-maturing precursor comparisons in that our early-maturing precursors excluded immature cone and rod precursors. Still, the comparisons revealed many of the same differentially expressed genes (**Figure 1—figure supplement 3A, E, G and H**; **Supplementary file 1A, C, D and E**).

To further interrogate cell identities, we used SCENIC to identify cluster-specific transcription factor regulons, which represent the overall expression of single transcription factors and their likely coregulated target genes (**Van de Sande et al., 2020**). The highest specificity regulons defined major cell populations including RPC/MG-specific E2F2, E2F3, VSX2, and PAX6; pan-photoreceptor NEUROD1, OTX2, and CRX; rod-specific NRL; and L/M cone-specific THRB and ISL2 (**Figure 2** and **Supplementary file 1F**). SCENIC also distinguished ER and LR states via the latter's increased NRL, CRX, ATF4, and LHX3 and decreased HMX1 and RAX activities (p<0.0005 for each, Dunn test). RAX activity also decreased in the 5 cell late-maturing L/M cone group (**Figure 2B**), supporting its distinct transcriptomic identity and suggesting a similar mode of late cone and late rod maturation. Additionally, iPRPs expressed their most specific regulons, LHX9 and OLIG2, at levels similar to RPC/MGs along with photoreceptor-related regulons, consistent with the transitional nature of this population (**Figure 2A** and **Supplementary file 1F**). However, SCENIC did not identify S cone-specific regulons, in keeping with the notion that S cones represent a default photoreceptor state induced by pan-photoreceptor factors such as OTX2, CRX, and NEUROD1 in the absence of NRL and THRB (**Swaroop et al., 2010**). Thus, deep, full-length scRNA-seq enabled identification of regulons underlying RPC and developing photoreceptor states at the single-cell level.

## Differential expression of *NRL* and *THRB* isoforms in rod and cone precursors

Although cone and rod precursors segregated into distinct clusters, mRNAs encoding rod-determining factor NRL, L/M cone-determining factor TRβ, and cone marker RXRγ were co-expressed in both rod and cone precursor populations, with mean *NRL* expression only 4.3-fold higher in the ER vs LM cluster, mean *THRB* expression 5.0-fold higher in LM vs LR, and mean *RXRG* expression 4.3-fold

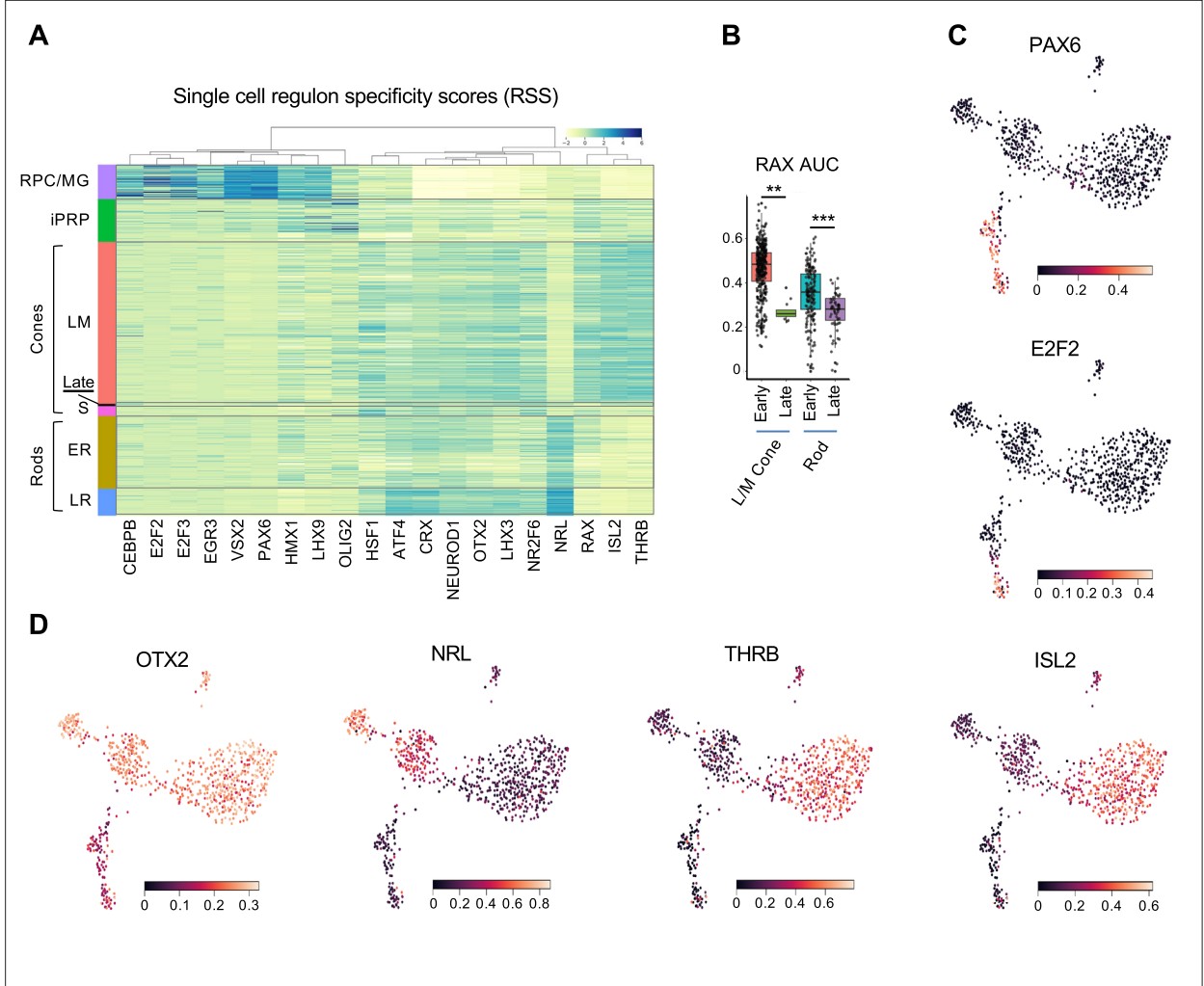

**Figure 2.** Regulon-defined RPC and photoreceptor precursor states. (**A**) Ward-clustered heatmap of the highest scoring SCENIC regulons in each cluster, displaying Z-score normalized regulon activities. Late = late-maturing L/M cones. (**B**) Box plot of RAX regulon area under the curve (AUC) values for early and late L/M cones and rods. *, p<0.005; ***, p<0.0005, Dunn test. (**C,D**) UMAP plots of regulon AUC values for (**C**) PAX6 (RPC/MG) and E2F2 (RPC), and (**D**) OTX2 (photoreceptors and photoreceptor-committed RPCs), NRL (rod) and THRB and ISL2 (L/M cone).

higher in LM vs ER (*Figure 1—figure supplement 2C and D*). Cone *NRL* expression was unexpected given NRL's role in rod fate determination (*Kallman et al., 2020*) and rod-specific NRL regulon activity (*Figure 2A and D*). Similarly, rod *RXRG* and *THRB* expression were unexpected given their roles in cone gene expression and fate determination (*Ng et al., 2001*; *Roberts et al., 2005*) and L/M cone-specific THRB regulon activity (*Figure 2A and D*). Accordingly, we used full-length scRNA-seq data to determine if cone and rod precursors differentially express *NRL*, *THRB*, and *RXRG* transcript isoforms.

For *NRL*, three assigned transcript isoforms (*ENST00000397002*, *ENST00000561028*, and *ENST00000558280*) are predicted to encode the canonical full-length NRL protein (FL-NRL) (RefSeq NP_001341697.1), while two others (*ENST00000560550* and ENST00000396995) are previously uncharacterized transcripts predicted to use an alternative 'P2' promoter and first exon, here termed exon 1T (*Figure 3A–C*). The novel transcripts are predicted to encode an N-terminally truncated NRL protein (Tr-NRL) retaining the leucine zipper DNA binding domain but lacking the minimal transactivation domain (*Friedman et al., 2004*; *Figure 3C and Figure 3—figure supplement 1A and B*). While all transcript isoforms were inferred to be more highly expressed in the ER rod cluster versus the LM cone cluster, the ratio of all FL-NRL:Tr-NRL transcripts was 2.9:1 in early rod precursors and 2.2:1 in late rod precursors, in contrast to 0.67:1 in L/M cone precursors (*Figure 3B*). Consistent with the assigned isoform ratios, mean read coverage of the Tr-NRL-specific exon 1T was higher in S and LM cones, while coverage of FL-NRL-specific exon 1 was higher in rods (*Figure 3C*, red *vs.* black

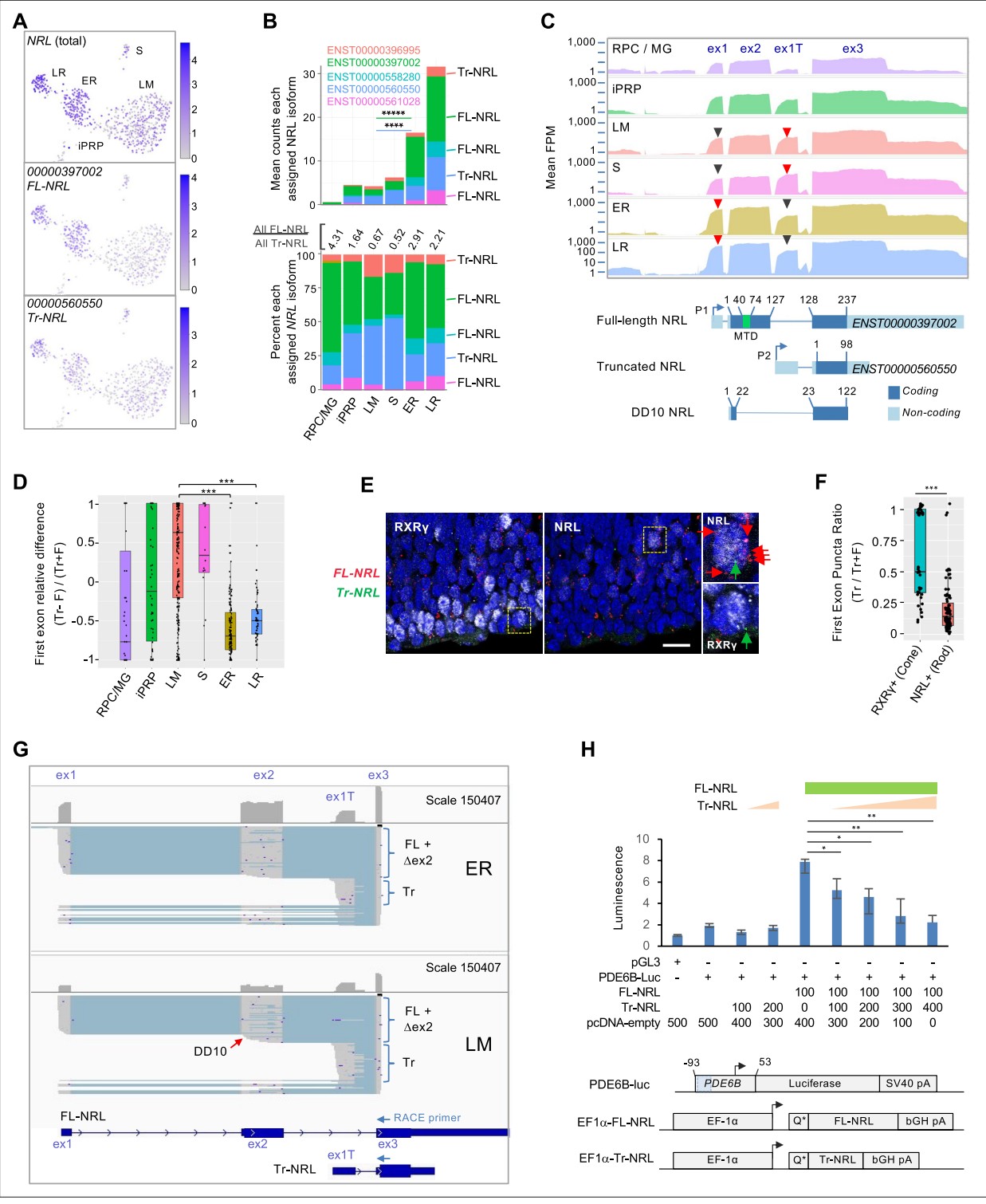

**Figure 3.** Differential expression of *NRL* isoforms in rod and cone precursors. (**A**) Expression of *NRL* gene and the most highly assigned Ensembl isoforms *ENST00000397002* (FL-NRL) and *ENST00000560550* (Tr-NRL). (**B**) Mean *NRL* isoform assignments for clusters defined in ***Figure 1B***, presented as total counts (*top*) and percentage of total counts (*bottom*). Significance for LM vs. ER fold change, colored by isoform. ****, p<0.0002; *****, <0.000001 (bootstrapped Welch's t-test). Ensemble transcript IDs shown in color with structures shown in ***Figure 3—figure supplement 1B***. (**C**) *Top*: Mean read counts (fragments per million, FPM) across Ensembl *NRL* exons for each cluster. *Bottom*: Transcript structures numbered according to amino acid positions. Minimal transactivation domain (MTD) in green. Arrowheads: Red/black: First exons where red is higher of two peaks. (**D**) Relative difference box plot of raw reads mapping to truncated (Tr) and full length (**F**) transcript first exons in each cell, according to cluster. Relative difference

*Figure 3 continued on next page*

*Figure 3 continued*

is the difference in reads mapping to truncated and full-length *NRL* first exons (Tr-F) divided by the sum of both (Tr + F). Values >0 indicate more reads assigned to truncated isoform, values <0 indicate more reads assigned to full-length isoform. ***, p<0.0001 (post-hoc Dunn test). (**E**) NRL and RXRγ immunostaining and RNA FISH with probes specific to truncated Tr-NRL exon 1T (green puncta) and FL-NRL exons 1 and 2 (red puncta) in FW16 retina. Boxed regions enlarged at right show an RXRγ$^{lo}$, NRL$^+$ rod with one Tr-NRL and six FL-NRL puncta (*top*) and an RXRγ$^{hi}$, NRL$^-$ cone with one Tr-NRL and no FL-NRL puncta (*bottom*), indicated with same-color arrows. Scale bar: 10 μm. (**F**) Ratio of fluorescent puncta observed in experiment depicted in (**E**) for NRL$^+$ or RXRγ$^{hi}$ cells where Tr puncta >0. ***; p<0.0005 (Welch's t-test). (**G**) Long-read nanopore sequencing of pooled 5' RACE reactions initiated with *NRL* exon 3 primers and performed on cDNA libraries from 23 ER cells (*top*) and 21 LM cells (*bottom*). Each schematic shows total exon coverage (*above*) and individual transcripts (*below*), where expressed sequences are gray and introns light blue. Full-length (FL), alternatively spliced or internally initiated exon 2 (Δex2), and truncated (Tr) transcripts are indicated by brackets. Red arrow: Transcripts resembling DD10, with internal exon 2 transcription initiation and premature splicing to exon 3. Ensembl FL-NRL and Tr-NRL transcript isoforms and RACE primer positions are shown below. (**H**) *Top*: PDE6B-luciferase reporter activity in NIH-3T3 cells transfected with indicated amounts (ng) of pcDNA4-C-EF1α and derived FL-NRL and Tr-NRL constructs. *Bottom:* PDE6B-luc reporter and pcDNA4-C-EF1α expression constructs. Blue box = NRL response element. Error bars = standard deviation of triplicate measurements. *, p<0.05; **, <0.005 (Student's t-test). Data representative of two experiments in NIH-3T3 and one in HEK-293T.

The online version of this article includes the following source data and figure supplement(s) for figure 3:

**Figure supplement 1.** Differential *NRL* transcript isoform expression in early rod and cone precursors.

**Figure supplement 2.** Cell-type-specific NRL protein expression.

**Figure supplement 2—source data 1.** PDF file containing original western blots for *Figure 3—figure supplement 2*, indicating the relevant bands and treatments.

**Figure supplement 2—source data 2.** Original files for western blot analysis displayed in *Figure 3—figure supplement 2*.

arrowheads). Comparing the reads mapped to each first exon relative to total reads further confirmed that the Tr-NRL exon 1T predominated in individual cones whereas the FL-NRL exon 1 predominated in rods (*Figure 3D*). The cone cells' higher proportional expression of Tr-NRL first exon sequences was validated by RNA fluorescence in situ hybridization (FISH) of FW16 fetal retina in which NRL immunofluorescence was used to identify rod precursors, RXRγ immunofluorescence was used to identify cone precursors, and FISH probes specific to Tr-NRL exon 1T or to FL-NRL exons 1 and 2 were used to assess Tr-NRL and FL-NRL expression (*Figure 3E and F*).

While the Tr-NRL-encoding *NRL* isoforms were not to our knowledge previously described, another *NRL* isoform that initiated within exon 2 and lacked the NRL transactivation domain due to alternative exon 2 splicing, termed DD10 (*Figure 3C*), was previously identified in adult retina (*Swaroop et al., 1992*). Concordantly, we detected reads spanning the unique DD10 splice junction, yet at lower levels than the unique Tr-NRL junction (5,942 vs 57,048).

As transcript isoforms inferred from short-read sequencing do not necessarily reflect the original transcript structures, we further defined the *NRL* isoforms expressed in early cone and rod precursors by performing 5' rapid amplification of cDNA ends (RACE) on the already generated single cell cDNA libraries from 23 early rod (ER) and 21 early L/M cone (LM) cells, followed by nanopore long-read sequencing of the pooled RACE products. The long-read sequencing revealed isoforms consistent with FL-NRL, Tr-NRL, DD10, and several other *NRL* isoforms with alternative transcription initiation and alternative splicing within exon 2 as well as within the Tr-NRL exon 1T (*Figure 3G and Figure 3—figure supplement 1C*). In keeping with the computationally assigned isoforms and differential exon usage, ER libraries had a higher proportion of FL-NRL exon 1 and exon 2 reads, and LM libraries had a higher proportion of Tr-NRL exon 1T reads (*Figure 3G and Figure 3—figure supplement 1C*). Moreover, alternative splicing within NRL exon 2 was more prevalent in LM libraries, affecting 6788 of 29,177 (23 %) of exon 2 reads compared to 5378 of 85,860 (6 %) in ER cells (p<0.0001; Chi-square with Yates correction), resulting in rarer full-length (exon 1-2-3) transcripts than inferred from short-read sequencing. Thus, long-read sequencing revealed cell type-biased *NRL* isoform expression with a preponderance of FL-NRL transcripts in early rods and disrupted FL-NRL and Tr-NRL transcript isoforms in L/M cones.

Despite our detection of L/M cone RNAs encoding Tr-NRL and FL-NRL, cone expression of NRL protein has not been reported. To assess endogenous Tr-NRL expression, we performed immunoblot analysis of CHLA-VC-RB31 retinoblastoma cells (*Stachelek et al., 2023*), which were predicted to express FL-NRL and Tr-NRL transcripts in a cone-like 0.73:1 ratio, with an antibody that recognizes both FL-NRL and Tr-NRL proteins (*Figure 3—figure supplement 2A and B*). As in other retinoblastoma cells (*Khanna et al., 2006*), FL-NRL increased in response to retinoic acid and proteasome

inhibition, whereas Tr-NRL was not detected (*Figure 3—figure supplement 2C*). Similarly, Tr-NRL was not detected in EGFP+, RXRγ+ cones following lentiviral transduction of an explanted fetal retina with an EGFP-P2A-Tr-NRL cassette (*Figure 3—figure supplement 2D–G*). These findings suggest that Tr-NRL is poorly translated or unstable in most cone precursors.

As Tr-NRL might be expressed in contexts that were not examined in our analyses, we explored the function of the putative Tr-NRL protein. As both Tr-NRL and the previously described NRL DD10 lack the NRL transactivation domain and as DD10 interferes with FL-NRL transactivation (*Rehemtulla et al., 1996*), we examined if Tr-NRL similarly opposes FL-NRL transcriptional activity. Indeed, in luciferase reporter assays, Tr-NRL suppressed FL-NRL activation of a *PDE6B* promoter (*Figure 3H*).

For *THRB*, the most highly assigned transcript isoforms encoded the L/M cone-specific TRβ2 (*ENST00000280696*) and the more widely expressed TRβ1 (*ENST00000396671* and others; *Figure 4A-C*, *Figure 4—figure supplement 1*). While both TRβ1 and TRβ2 promote L/M cone fate determination (*Eldred et al., 2018*), the isoform encoding TRβ2 predominated in L/M cones while isoforms encoding TRβ1 predominated in early rods (*Figure 4B*). Moreover, late rods preferentially expressed *THRB* exons 1–6 and the *THRB* 3' untranslated region (UTR), implying that RNAs encoding full-length TRβ proteins were rare (*Figure 4B and C*). Notably, a higher percentage of reads extended from the exon 4 and exon 6 splice donor sequences into the subsequent introns in LR versus LM cells (*Figure 4D and E*; p≤0.001 for both, two-tailed Chi square test), suggesting that premature transcription termination (PTT) in introns 4 and 6 preferentially limits full-length TRβ expression in the LR population. The inferred PTT events are consistent with structures of the assigned Ensembl isoforms (*Figure 4B*, *Figure 4—figure supplement 1*).

To further evaluate PTT events, we used *THRB* 3' RACE and long-read sequencing on single-cell cDNA libraries from 23 L/M cone cells and five LR cells selected for high *THRB* expression. RACE reactions were performed separately with primers complementary to the TRβ1-specific exon 4 and to the TRβ2-specific first exon (*Figure 4F*). Sequencing of LM and LR RACE products corroborated pronounced PTT in introns 4 and 6, with greater intron 6 PTT in LR versus LM cells (*Figure 4F and Figure 4—figure supplement 2*). However, we did not corroborate the late rods' proportionately higher intron 4 PTT, likely due to the small number of LR cells examined and heterogeneity in PTT frequency in individual cells. Long read sequencing also revealed PTT following the TRβ2-specific exon and a novel transcription-terminating exon following the canonical exon 5 observed solely in TRβ2 transcripts (*Figure 4F and Figure 4—figure supplement 2*). 3' RACE transcripts rarely extended into the 3' UTR in LM or LR cells, suggesting that reads mapping to this region do not reflect protein-coding RNAs and confound the assessment of protein-coding *THRB* mRNA expression. These analyses demonstrate that *THRB* is regulated by multiple PTT events in rod and cone precursors as well as by cell type-specific promoter utilization and independent 3' UTR RNA expression.

For *RXRG*, short read sequencing reads were assigned to several isoforms that differed in their 5' promoter position and exon utilization (*Figure 4—figure supplement 3A–C*). However, 5' RACE and long-read sequencing did not support differential isoform expression, and quantitative imaging revealed an average ~3.5-fold higher RXRγ protein in cones compared to rods (*Figure 4—figure supplement 3D and E*). Thus, long-read sequencing clarified that *RXRG* expression is moderately higher in human cone versus rod precursors without evidence of cell-type-specific isoforms.

## Two post-mitotic immature photoreceptor precursor populations

To further define photoreceptor developmental states, we subdivided the initial six cell populations with higher resolution clustering (level 1.6), identified high resolution cluster-associated genes and regulons, and inferred each cell's rate and direction of transcriptomic change using RNA velocity (*La Manno et al., 2018*; *Figure 5A and B*). Increased clustering resolution divided the RPC/MG cluster into separate RPC and Müller glia (MG) groups, divided L/M cones into four subgroups (LM1 – LM4), which partially overlapped in UMAP space, and divided the iPRP cluster into two clusters here designated immature cone precursors (iCPs) and immature rod precursors (iRPs), the latter also drawing cells from the low-resolution ER cluster (*Figure 5A and C*). Similar clusters were observed at reduced k.param values that define nearest neighbors.

The distinction between RPCs and MG was corroborated by the expression of known marker genes, with the RPC cluster having increased expression of cell cycle markers (*CCNE2*, *CCNA2*, *CCNB2*) and the cell cycle-associated *PBK* and *E2F7* (*Figure 5D*, *Figure 5—figure supplement 1A*,

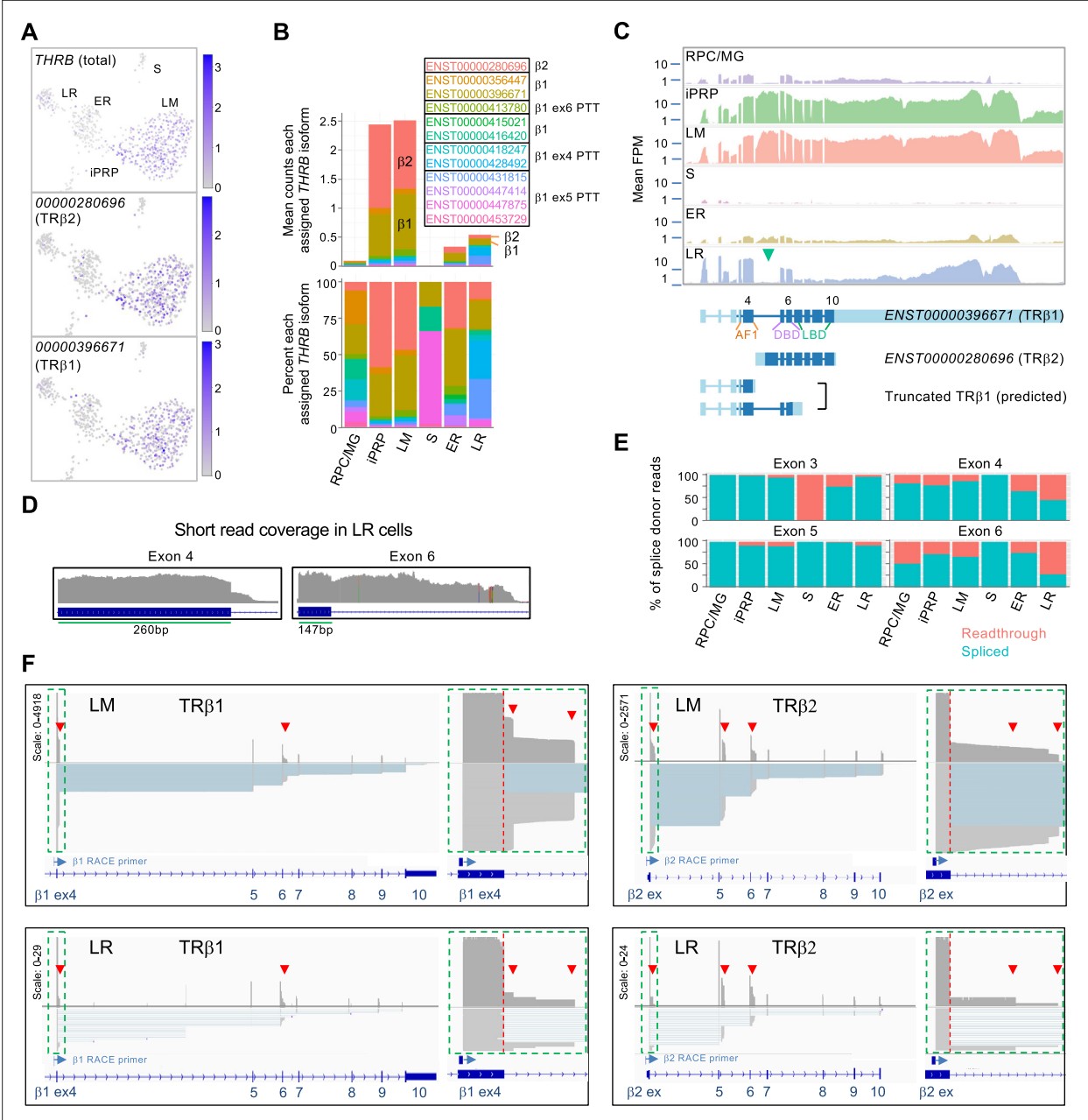

**Figure 4.** Differential expression of *THRB* isoforms in rod and cone precursors. (**A**) Expression of *THRB* and highly assigned isoforms *ENST00000280696* (encoding TRβ2) and *ENST00000396671* (TRβ1). (**B**) Mean *THRB* isoform assignments for each cluster presented as counts (*top*) and percentage of counts (*bottom*); Ensemble transcript IDs shown in color with β2, β1, and β1 PTT isoform structures as in *Figure 4—figure supplement 1*. (**C**) *Top*: Mean read counts across Ensembl *THRB* exons. *Bottom*: Transcript structures for TRβ1, TRβ2, and two TRβ1 truncations. Green arrowhead: First TRβ2 exon. *ENST00000396671* exon numbers are indicated above and protein domains (AF1, DNA-binding (DBD), and ligand binding (LBD)) below. (**D**) Read coverage for LR cells across *THRB* exons 4 and 6 splice donor sites. (**E**) Percentage of exon splice donor reads that are spliced or readthrough to the subsequent intron. (**F**) Long-read sequencing of pooled 3' RACE reactions initiated with exon 4 (*left*) or TRβ2 exon 1 (right) performed on cDNA libraries from 21 LM cells (*top*) and 5 LR cells (*bottom*). Schematics show total coverage (*above*) and individual transcripts (*below*). TRβ1 and TRβ2 first exons (green boxes) are enlarged at right. Red arrowheads: intronic premature transcription termination (PTT).

The online version of this article includes the following figure supplement(s) for figure 4:

**Figure supplement 1.** Differential *THRB* isoform assignments in rod and cone precursors.

**Figure supplement 2.** THRB isoforms and premature transcription termination in cone and rod precursors.

**Figure supplement 3.** *RXRG* isoform expression in rod and cone precursors.

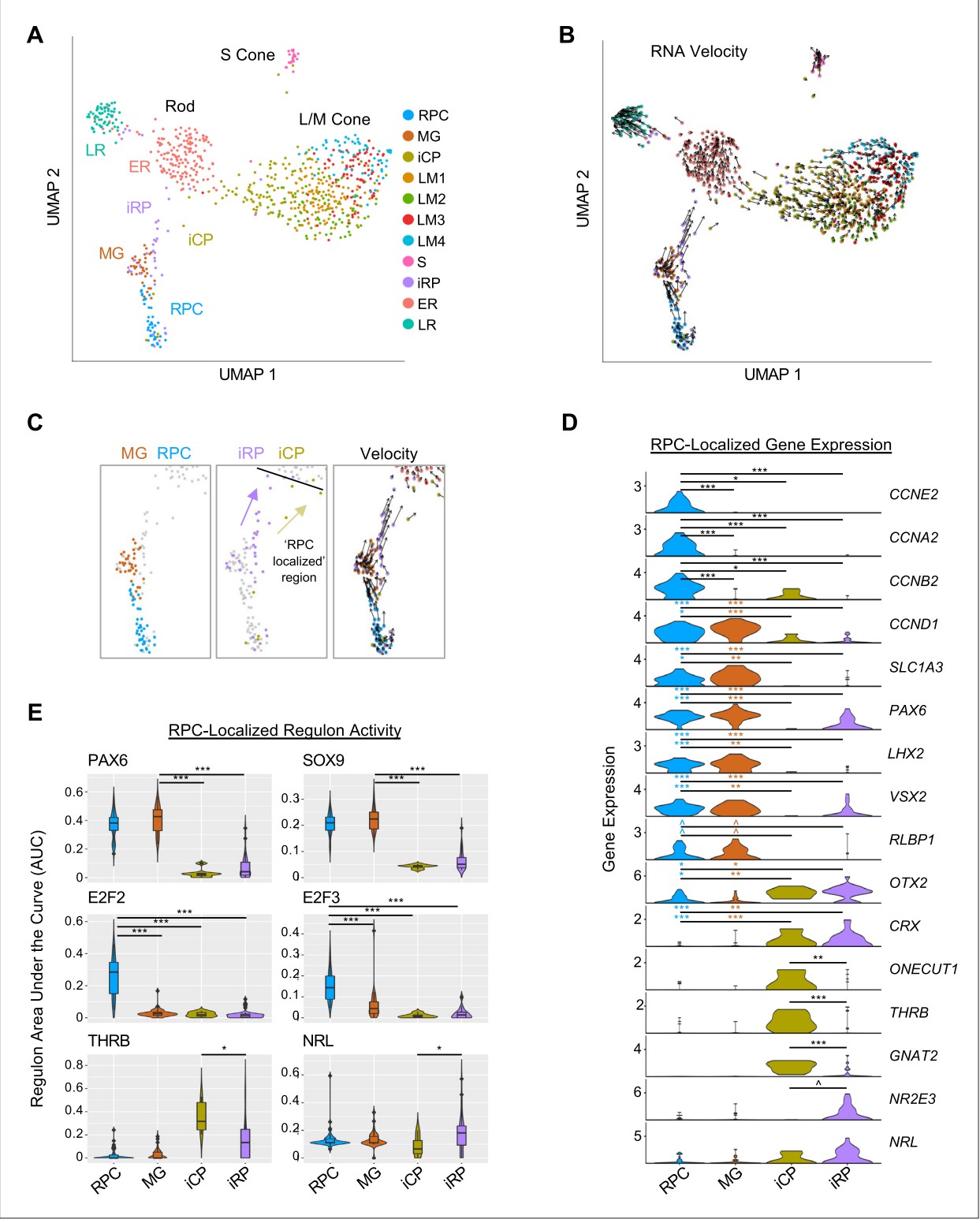

**Figure 5.** Two post-mitotic photoreceptor precursor populations expressing rod or cone markers. (**A**) UMAP plot colored by high resolution clusters. (**B**) RNA velocity plots with cell clusters as in A. (**C**) Enlarged view highlighting RPC and MG clusters (*left*), RPC-localized iCP and iRP clusters (*middle*), and RNA velocity (*right*). Black line: limit of RPC-localized region. Arrows depict inferred trajectories. (**D**) Violin plots depict expression of selected genes in RPC, MG, and RPC-localized iCP and iRP cells. Colored asterisks compare clusters of the same color to the cluster at the right of the line. (**E**) SCENIC regulon violin and box plots for RPC-localized cells in each cluster, selected from most specific regulons for MG, RPC, ER, and LM clusters. ^, pAdj <0.1; *, <0.05; **, <0.005; ***, <0.0005 (post-hoc Dunn test).

*Figure 5 continued on next page*

*Figure 5 continued*

The online version of this article includes the following figure supplement(s) for figure 5:

**Figure supplement 1.** High-resolution cluster marker genes and differential RPC versus MG gene expression.

**Figure supplement 2.** Immature photoreceptor precursor populations identified with deep full-length scRNA-seq and 3' snRNA-seq.

**Figure supplement 3.** Cone and rod precursor cell type assignments, UMAP positions, and gene expression in a large 3' snRNA-seq analysis.

*and Supplementary file 1G*). The MG cluster was spatially segregated, lacked expression of most cell cycle genes, and expressed genes known to be expressed in both populations (*CCND1*, *SLC1A3*, *PAX6*, *LHX2*, *VSX2*) as well as the MG marker *RLBP1* (*Roesch et al., 2008*; *Pereiro et al., 2020*; *Blackshaw et al., 2004*; *Figure 5C and D and Figure 5—figure supplement 1A*). Differential gene expression analyses revealed upregulation of cell-cycle-related genes and G2/M checkpoint and E2F target ontologies in the RPC cluster but no significantly upregulated ontologies in the MG population (*Figure 5—figure supplement 1B and C*), consistent with early MGs resembling quiescent RPCs (*Walcott and Provis, 2003*). In contrast to other studies, we did not distinguish primary RPCs (PRPCs) from neurogenic RPCs (NRPCs; *Clark et al., 2019*; *Lu et al., 2020*; *Zuo et al., 2024*), likely due to the underrepresentation of RPCs in our dataset.

The high-resolution iCP and iRP clusters included 'RPC-localized' cells positioned adjacent to RPCs and 'non-RPC-localized' cells adjacent to the LM and ER clusters in UMAP space (*Figure 5C*). RNA velocity suggested that the RPC-localized cell groups flowed towards the larger early maturing L/M cones and rods, respectively (*Figure 5B and C*). Compared to RPCs and MG, RPC-localized iCP and iRP cells had minimal expression of cyclin RNAs or RPC markers *PAX6*, *LHX2*, and *VSX2*, and had increased expression of photoreceptor determinants *OTX2* and *CRX* (*Figure 5D*), consistent with their being immediately post-mitotic photoreceptor precursors. iCP cells in this region upregulated the early cone cell fate determinant *ONECUT1* (*Emerson et al., 2013*; *Finkbeiner et al., 2022*), the L/M cone determinant *THRB*, and the cone differentiation marker *GNAT2*, whereas iRP cells trended towards higher expression of the rod determinant *NR2E3* (*Figure 5D and Figure 5—figure supplement 2A*); however, such RNAs may be more highly expressed in cone- and rod-fated NRPCs than in the unspecified RPCs in our study. Notably, *THRB* and *GNAT2* expression did not significantly change while *ONECUT1* declined in the subsequent non-RPC-localized iCP and LM1 stages, whereas *NR2E3* and *NRL* dramatically increased on transitioning to the ER state (*Figure 5—figure supplement 2A*).

In the RPC-localized region, iCPs had higher *ONECUT1*, *THRB*, and *GNAT2*, whereas iRPs trended towards higher *NRL* and *NR2E3* (p=0.19, p=0.054, respectively; *Figure 5D*). While we detected differential expression when selectively examining these genes of interest, no genes were differentially expressed (Padj <0.05) in a transcriptome-wide comparison, likely due to the lack of statistical power given the small number of cells examined.

To our knowledge, past studies have not distinguished immature cone and rod precursors (i.e. iCPs and iRPs) from the subsequent maturing cone and rod precursor states. To explore whether early cone and rod precursors with similar properties are present in droplet-based scRNA-seq studies, we examined gene expression in spatiotemporally segregated cone and rod precursor populations as well as in cone and rod NRPCs in the *Zuo et al., 2024* 3' snRNA-seq dataset. This revealed that cone NRPCs and the earliest (pcw 8–10) cone precursors had high *ONECUT1* and *THRB* yet minimal *NR2E3* (*Figure 5—figure supplement 2B*). Similarly, rod NRPCs had high *NRL* and *NR2E3*, yet it was not possible to examine the earliest (pcw 10–13) rod precursors, as many had cone-like gene expression profiles, suggesting they may have been mis-assigned (*Figure 5—figure supplement 2B*, *gray boxes*). Further evaluation suggested misassignment of a smaller proportion of rod and cone precursors at other ages (*Figure 5—figure supplement 3*). In particular, cells potentially misassigned as rod precursors were located in the cone precursor UMAP region and highly expressed *THRB* (*Figure 5—figure supplement 3D*, **), while cells potentially misassigned as cone precursors were located in the rod precursor UMAP region and highly expressed *NR2E3* (*Figure 5—figure supplement 3E*, *). When considering only the main cone and rod precursor UMAP regions, early (pcw 8–13) cone precursors expressed *THRB* and lacked *NR2E3* (*Figure 5—figure supplement 3D and E*, *blue arrows*), while early (pcw 10–15) rod precursors expressed *NR2E3* and lacked *THRB* (*Figure 5—figure supplement 3D and E*, *red arrows*), similar to RPC-localized iCPs and iRPs in our study (*Figure 5D*). We conclude

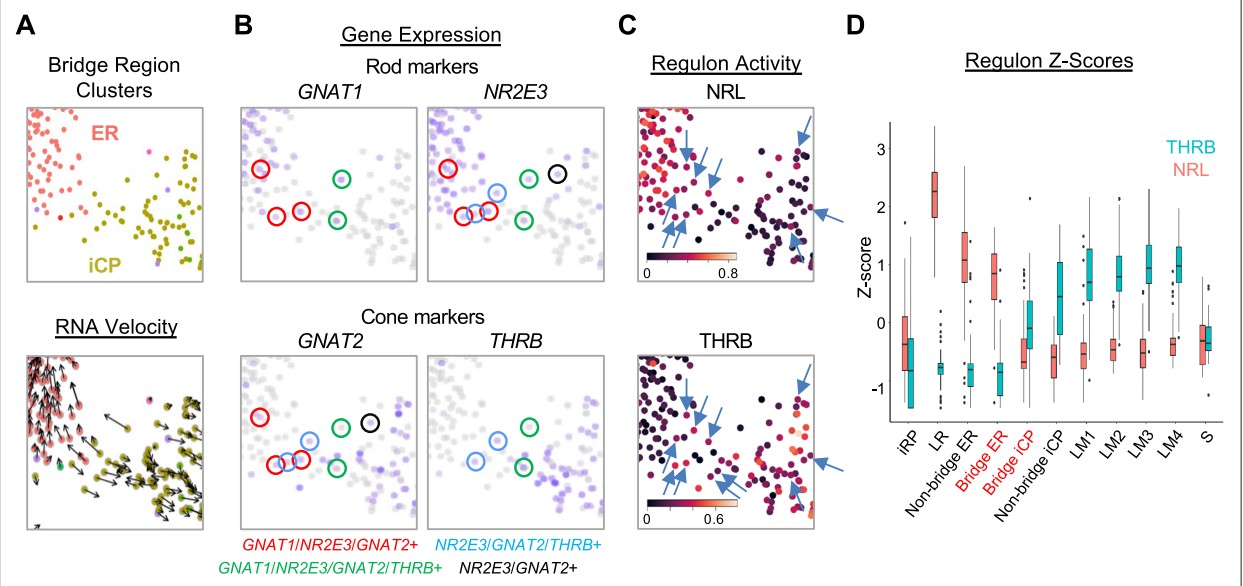

**Figure 6.** An iCP sub-population with cone- and rod-related RNA co-expression. (**A–C**) UMAP 'bridge' region cells colored by ER and iCP cluster and RNA velocity (**A**), rod and cone marker gene expression (**B**), and NRL and THRB regulon activity (C, arrows indicate cells with both regulon signals). (**D**) Box plot of Z-score-normalized NRL and THRB regulon AUCs for each cluster; Bridge ER and Bridge iCP represent cells present in the UMAP region in panels (**A-C**).

that Louvain clustering of deep full-length scRNA-seq distinguished immediately post-mitotic iCP and iRP populations with features similar to early cone and rod precursors in a large 3' snRNA-seq dataset.

In a further comparison, we noted lower detection of *GNAT2* in L/M cone precursors and lower detection of *CCNE2*, *CCNA2*, and *CCNB2* in RPCs in 3' snRNA-seq versus full-length scRNA-seq datasets (*Figure 5—figure supplement 2A and B*). This suggests that deep full-length scRNA-seq enabled more sensitive detection of certain cell-type-specific RNAs along with improved discrimination of immature cone versus rod precursor states.

SCENIC regulons further clarified the identities of the four RPC-localized clusters. The MG cluster was best specified (*i.e.*, had highest regulon specificity scores) by PAX6 and SOX9, both previously described in RPCs and MGs (*Roesch et al., 2008*; *Marquardt et al., 2001*; *Poché et al., 2008*), whereas RPCs were best specified by E2F2 and E2F3 (*Figure 5E*). These RPC and MG regulons were low or absent in the early post-mitotic iCPs and iRPs, suggesting they undergo an abrupt cell state change. Moreover, RPC-localized iCPs had greater THRB regulon signal than RPC-localized iRPs, while RPC-localized iRPs had higher NRL regulon signal (p<0.05 for both; *Figure 5E*). Interestingly, RPC-localized and/or non-localized iCPs also had higher activity of the pan-photoreceptor regulons LHX3, OTX2, and NEUROD1 compared to their iRP counterparts (*Figure 5—figure supplement 2C and D*). Finally, activities of the cone-specific THRB and ISL2 regulons, the rod-specific NRL regulon, and the pan-photoreceptor LHX3, OTX2, CRX, and NEUROD1 regulons increased to varying extents on transitioning from immature iCP or iRP states to the early-maturing LM1 or ER states (*Figure 5—figure supplement 2C and D*).

## Early cone and rod precursors with rod- and cone-related RNA co-expression

In addition to the immediately post-mitotic RPC-localized iCPs, the iCP cluster included cells bridging the UMAP region between early maturing cones and early maturing rods (*Figure 6A*). Many iCP and ER cells in this bridge region expressed RNAs encoding cone markers (*GNAT2*, *THRB*), rod markers (*GNAT1*, *NR2E3*), or, in many cases, both (*Figure 6B*), suggestive of a proposed hybrid cone/rod precursor state more extensive than implied by the co-expression of different *THRB* and *NRL* isoforms (*Ng et al., 2011*).

To determine whether the intermixed cone and rod gene expression reflects a hybrid cone-rod precursor state, we examined bridge region NRL and THRB regulon activities, which embody the

overall transcriptomic effects of these factors. Indeed, in the bridge region, cells with NRL and THRB regulon signals intermixed, and some cells showed signals for both (*Figure 6C*, *arrows*). However, iCP cells had low NRL regulon z-scores similar to L/M cones and most had THRB z-scores above that of adjacent ER cells (*Figure 6D*). The difference between regulon signals increased in rod and cone clusters outside the bridge region as NRL or THRB regulons became more dominant (*Figure 6D*).

To identify cells with early cone- and rod-related RNA co-expression in the developing retina, we performed multiplex RNA FISH for early cone marker *GNAT2* and rod marker *NR2E3* combined with immunofluorescence (IF) staining for RXRγ, which has high expression in outermost neuroblastic layer (NBL) cone precursors and low expression in middle NBL rod precursors (as earlier shown in *Figure 4—figure supplement 3E*), and for NR2E3, which is detected solely in rods. To infer the spatiotemporal pattern of such expression across human retinal development, we examined *GNAT2* and *NR2E3* RNA co-expression in RXRγ⁺ cone precursors in the outermost NBL and in RXRγ⁺ rod precursors in the middle NBL across 13 regions of a FW14 retina section (*Figure 7*; see *Figure 7—figure supplement 1* for IF + FISH images). Limiting our analysis to the outer and middle NBL allowed us to disregard RXRγ⁺ retinal ganglion cells in the retinal ganglion cell layer or inner NBL.

The analyses revealed that most of the far peripheral (hence, nascent) outer NBL RXRγ⁺ cone precursors (*Figure 7A and B*, regions 2 and 13) lacked detectable *GNAT2* and *NR2E3* RNA (green bars in *Figure 7C*), whereas those in the more mature central retina (regions 3–12) were uniformly *GNAT2*⁺ (yellow and red bars in *Figure 7C*). However, starting with regions 3 and 11, some outermost NBL *GNAT2*⁺ cones were also *NR2E3*⁺ (red bars in *Figure 7C*). The proportion of outermost NBL cells that co-expressed *GNAT2* and *NR2E3* RNA increased in the more central retina (regions 5–10), yet all of these cells had strong RXRγ staining and lacked NR2E3 protein, consistent with their having a cone identity (illustrated in *Figure 7E*). In contrast, most middle NBL RXRγ⁺ cells had either a low number of *NR2E3* RNA puncta without NR2E3 protein (*NR2E3*⁺/NR2E3⁻, magenta bars in *Figure 7D*), likely in nascent rods, or prominent *NR2E3* RNA with NR2E3 protein (*NR2E3*⁺/NR2E3⁺, purple bars in *Figure 7D*), in maturing rods (*Figure 7D and E*). However, we did not detect *GNAT2* puncta in middle NBL RXRγ⁺, *NR2E3*⁺ cells, including the most peripheral RXRγ⁺ cells with low-level *NR2E3* RNA (*Figure 7E*). In summary, most photoreceptor precursors with *GNAT2* and *NR2E3* RNA co-expression had high RXRγ, no detectable NR2E3 protein, and outermost NBL positions expected of cone precursors. This supports the notion that, in our scRNA-seq analyses, bridge region iCP cells with combined *GNAT2* and *NR2E3* RNA expression are likely cone-directed.

Cone and rod precursor populations that co-express cone and rod marker genes were also evident in the Zuo et al. 3' snRNA-seq dataset. While the earliest cone precursors were *NR2E3*-negative and the earliest rod precursors were *THRB*-negative (as described above), starting at pcw 15, later cone precursors expressed *NR2E3* (*Figure 5—figure supplement 3E*, *green arrows*) and later rod precursors expressed *THRB* (*Figure 5—figure supplement 3D*, *purple arrows*). *THRB* was highest in the most distal rod precursors (yet still lower than in cone precursors), corroborating our findings in *Figures 1D and 4B–C*. In contrast, in the 3' snRNA-seq analysis, *GNAT2* expression was sparse in rod precursors of all ages (*Figure 5—figure supplement 3F*), consistent with the lack of *GNAT2* in rod precursors in our RNA FISH analyses and implying that different cone- and rod-related genes have different tendencies to be expressed in the other photoreceptor precursor type.

Thus, a 3' snRNA-seq analysis confirmed the initial production of immature photoreceptor precursors with either L/M cone-precursor-specific *THRB* or rod-precursor-specific *NR2E3* expression, followed by lower-level co-expression of their counterparts, *NR2E3* in cone precursors and *THRB* in rod precursors. However, in the Zuo et al. analyses, the co-expression was first observed in well-separated UMAP regions, as opposed to a region that bridges the early cone and early rod populations in our UMAP plots. These findings are consistent with the notion that cone- and rod-related RNA co-expression begins in already fate-determined cone and rod precursors, and that such precursors aberrantly intermixed in our UMAP 'bridge region' due to their insufficient representation in our dataset.

## Developmental expression of photoreceptor precursor markers and fate determinants

To further assess relationships between iCP subpopulations, we examined the expression and UMAP distributions of the four iCP marker genes identified in *Figure 5—figure supplement 1A*. Among

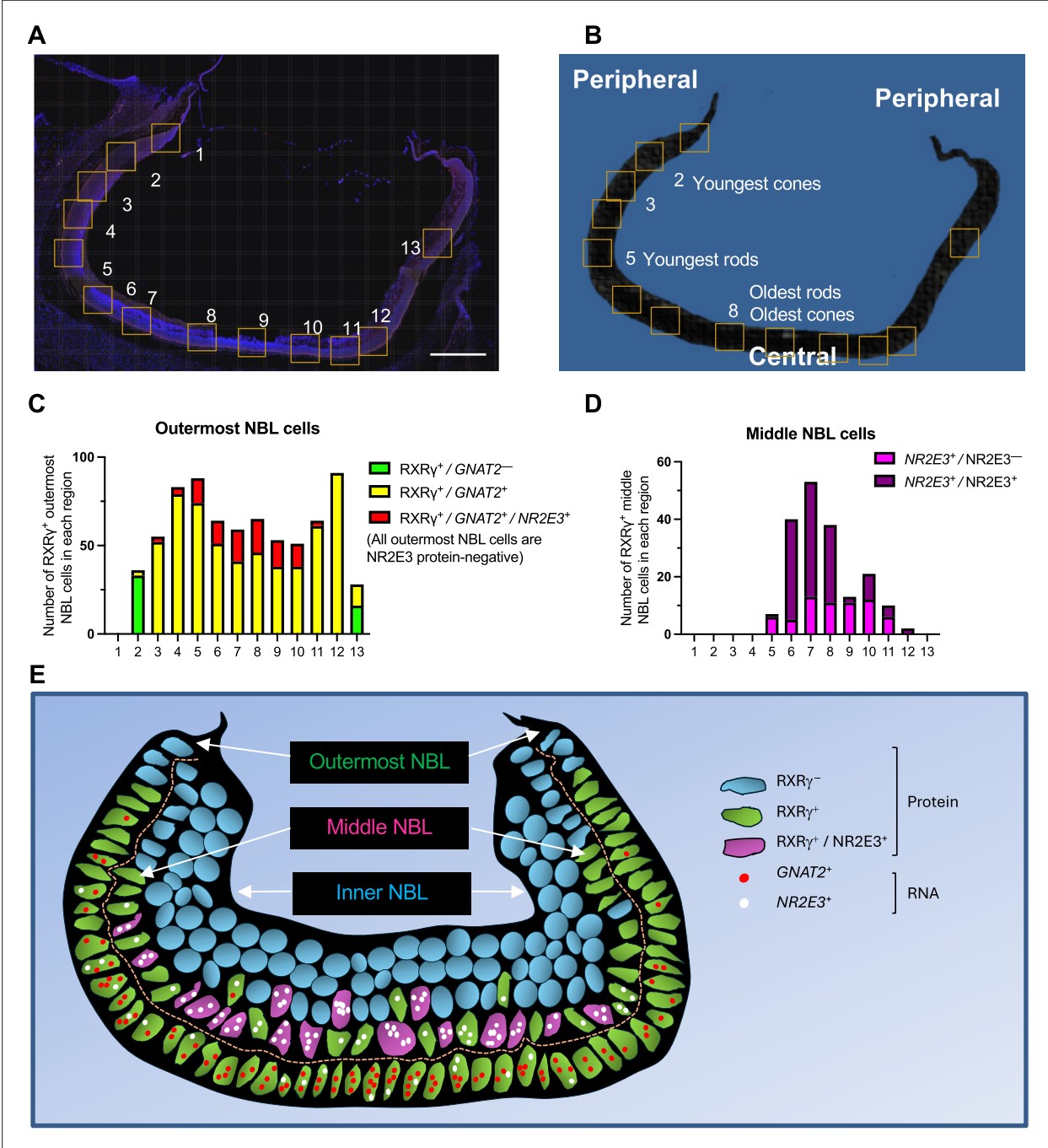

**Figure 7.** Cone-related *GNAT2* and rod-related *NR2E3* RNA co-expression in human cone precursors. (**A, B**) Tiled composite fluorescence image of FW14 retinal section after immunofluorescence staining of RXRγ and NR2E3 and RNA FISH of *GNAT2* and *NR2E3* (**A**) and diagram of the same section indicating the most peripheral (*i.e.*, youngest and least mature) and most central (*i.e.*, oldest and most mature) rods and cones (**B**). Scale bar in A = 500 μm. Boxes indicate regions further evaluated as shown in *Figure 7—figure supplement 1* and quantitated in panels C and D. (**C, D**) Quantitation of (**C**) outermost and (**D**) middle (i.e. sub-outermost) NBL photoreceptor precursors expressing combinations of RXRγ and NR2E3 proteins and *GNAT2* and *NR2E3* RNAs (*n.b.*, italics are used for RNAs, non-italics for proteins). (**E**) Patterns of *GNAT2* and *NR2E3* RNA and RXRγ and NR2E3 protein expression inferred from in situ hybridization and immunofluorescence staining. RXRγ is expressed in the outermost NBL starting in the far periphery, consistent with cone precursors, and in middle NBL cells, starting more centrally, consistent with rod precursors. *GNAT2* and *NR2E3* RNA co-expression in outermost NBL cells lacked NR2E3 protein, representing putative cone precursors. RXRγ+ retinal ganglion cells in the inner NBL and ganglion cell layer are not shown.

The online version of this article includes the following figure supplement(s) for figure 7:

**Figure supplement 1.** *NR2E3* and *GNAT2* co-expression in photoreceptor precursors.

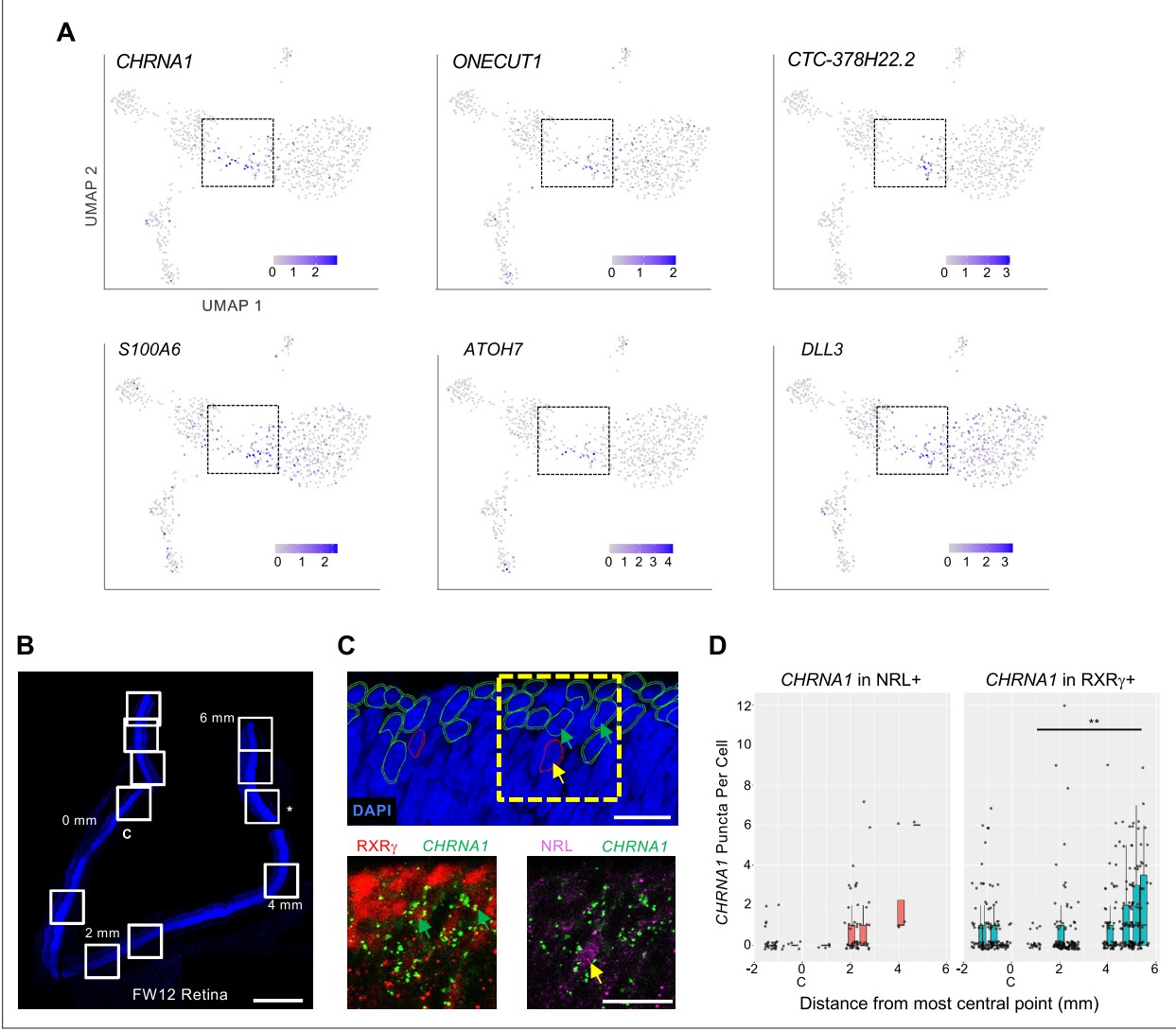

**Figure 8.** Developmental expression of early cone and rod precursor markers. (**A**) UMAP plots of iCP marker genes *CHRNA1*, *ONECUT1*, *CTC-378H22.2*, and *S100A* (see *Figure 5—figure supplement 1A*) and previously identified photoreceptor precursor markers *ATOH7* and *DLL3*. (**B–D**) Combined RXRγ/NRL immunohistochemical staining and *CHRNA1* RNA FISH of FW12 retina. (**B**) Tiled images of retina section with nuclei stained with DAPI. White boxes: fields used for quantitative fluorescent imaging. Distances along apical edge of tissue marked in mm from midpoint of central image (0 mm, **C**). *: Imaged region shown in **C**. Scale bar = 500 μm. (**C**) *Top:* Retinal nuclear and cellular segmentation and identification of cells as RXRγ⁺ (green outline) or NRL⁺ (red outline). Yellow box: Field shown below. *Bottom:* RXRγ or NRL immunofluorescence staining with *CHRNA1* FISH. Arrows: RXRγ⁺, *CHRNA1⁺* (green), NRL⁺, *CHRNA1⁺* (yellow). Scale bars = 15 μm. (**D**) Quantitation of fluorescent puncta in RXRγ⁺ and NRL⁺ cells by image field. X-axis: Distance from the midpoint of each image to retina center (0 mm, **C**). **, p<0.005 (Wald test, images from 0 to 6 mm).

The online version of this article includes the following figure supplement(s) for figure 8:

**Figure supplement 1.** Developmental expression of photoreceptor precursor markers in a large 3' snRNA-seq dataset.

**Figure supplement 2.** iCP-enriched regulon activities.

these genes, *CHRNA1* was mainly expressed in bridge region iCP and ER cells, whereas *ONECUT1* was biased to cone-directed iCPs, lncRNA *CTC-378H22.2* was expressed in a narrow zone of *THRB⁺*, *GNAT2⁺* iCPs, and *S100A6* was more widely expressed (*Figure 8A*). iCP cells also expressed *ATOH7* and *DLL3*, which were previously proposed to define transitional photoreceptor precursors and promote cone fate determination (*Clark et al., 2019*; *Lu et al., 2020*; *Sridhar et al., 2020*). *ATOH7* was largely restricted to RPC-localized and bridge region iCPs, whereas *DLL3* was broadly expressed similar to *S100A6* (*Figure 8A*).

A similar pattern was seen in the *Zuo et al., 2024* 3' snRNA-seq dataset, where *CHRNA1*, *ONECUT1*, *S100A*, *ATOH7*, and *DLL3* were most highly expressed in the youngest and developmentally earliest ML cone and rod precursors and persisted to various extents during cone and rod maturation, whereas *CTC-378H22.2* (ENSG00000259436) had more restricted and cone-specific expression (*Figure 8—figure supplement 1A*). However, the early cone and rod precursor UMAP regions did not adjoin one another (in contrast to our UMAP bridge region), suggesting that the cone and rod precursors expressing the above markers had begun their distinct trajectories and that their juxtaposition in our UMAP analyses is spurious. Also, different iCP markers had different spatiotemporal expression: *CHRNA1* and *ATOH7* were most prominent in peripheral retina with *ATOH7* strongest at pcw 10 and *CHRNA1* strongest at pcw 13; *CTC-378H22.2* was prominently expressed from pcw 10–13 in both the macula and the periphery; and *DLL3* and *ONECUT1* showed the earliest, strongest, and broadest expression (*Figure 8—figure supplement 1B*). The distinct patterns suggest that these factors have spatiotemporally distinct roles in cone precursor differentiation.

As *CHRNA1* appeared to be the most specific marker of both early cone and early rod precursors, we evaluated whether *CHRNA1* RNA marked specific cone and rod populations in the developing retina using RNA-FISH combined with RXRγ and NRL immunofluorescent staining. In a FW12 retina, we detected the highest *CHRNA1* in the earliest, most peripheral NRL$^+$ rods and RXRγ$^+$ cones and fewer *CHRNA1$^+$* cells in more mature, central regions (p<0.005 in cones; *Figure 8B–D*), consistent with its expression in early cone and rod precursors.

To identify factors that regulate transcriptomic states during cone cell fate determination, we examined the UMAP distributions of the most iCP-specific transcription factor regulons, OLIG2 and LHX9 (*Figure 5—figure supplement 2D*, *Supplementary file 1H*). As Olig2 was previously detected in mouse RPCs in which Onecut1 enabled *Thrb* expression and cone fate determination (*Emerson et al., 2013*; *Hafler et al., 2012*), the OLIG2 regulon was expected to be most active in RPCs and in early, immediately post-mitotic iCPs. While the OLIG2 and LHX9 regulons were indeed active in RPCs, both were also active in a narrow zone of *ONECUT1$^+$* iCPs positioned farthest from S cones and immediately preceding the upregulation of the THRB regulon, a location with sparse *OLIG2* RNA expression (*Figure 8—figure supplement 2*). These data indicate that the OLIG2 and LHX9 regulons are active in and potentially relevant to human L/M cone fate determination in a post-mitotic iCP subpopulation.

## An early L/M cone trajectory marked by successive lncRNA expression

After fate commitment, L/M cones initiate a maturation process with upregulation of RNAs and proteins related to phototransduction, axonogenesis, synaptogenesis, and outer segment morphogenesis (*Hendrickson et al., 2012*; *Hoshino et al., 2017*). To evaluate whether early L/M cone maturation is comprised of distinct transcriptomic cell states, we assessed marker gene and regulon differences between high-resolution clusters LM1, LM2, LM3, and LM4. While clusters LM1-4 were distinguished by sequential increasing expression of *ACOT7*, *RTN1*, *PDE6H*, *OLAH*, and *NPFF*, they showed only subtle differences among the three regulons with highest LM1-4 specificity scores, THRB, ISL2, and LHX3 (*Figure 9A and B*). The lack of cluster-specific marker genes and regulons suggests that LM1-LM4 represent different stages of a graded maturation process.

We further evaluated gene expression changes in maturing cones by pseudotemporally ordering iCP and L/M cone precursors (*Figure 9C*). This identified 967 pseudotime-correlated genes (q-value <0.05, expression >0.05 in >5% of cells; *Supplementary file 1I*) in seven gene modules (*Figure 8—figure supplement 1*). Among the top 20 pseudotime-correlated genes in each module, we identified four lncRNAs that were sequentially expressed (*Figure 9D*). To determine if these lncRNAs distinguish developmentally distinct maturing cones in vivo, we probed their expression via multiplex RNA FISH in co-stained RXRγ$^+$ cone precursors across a FW16 retina (*Figure 9E and F*). Quantitation of FISH puncta defined four cone maturation zones based on expression peaks and significant count differences for each lncRNA as color-coded in *Figure 9G*: the most peripheral cones with high *CTC-378H22.2* and *HOTAIRM1* (blue), peripheral cones with high *HOTAIRM1* only (green), cones with low expression of all four lncRNAs (red), and parafoveal/foveal cones with upregulated *RP13143G15.4*, *CTD-2034I21.2*, and *CTC-378H22.2* (purple) (*Figure 9G*). The detection of foveal *CTC-378H22* by ISH but not in late LM transcriptomes may relate to a lack of the most mature cones in our scRNA-seq analyses. These data support the concept that maturing L/M cones sequentially express specific lncRNAs as they develop.

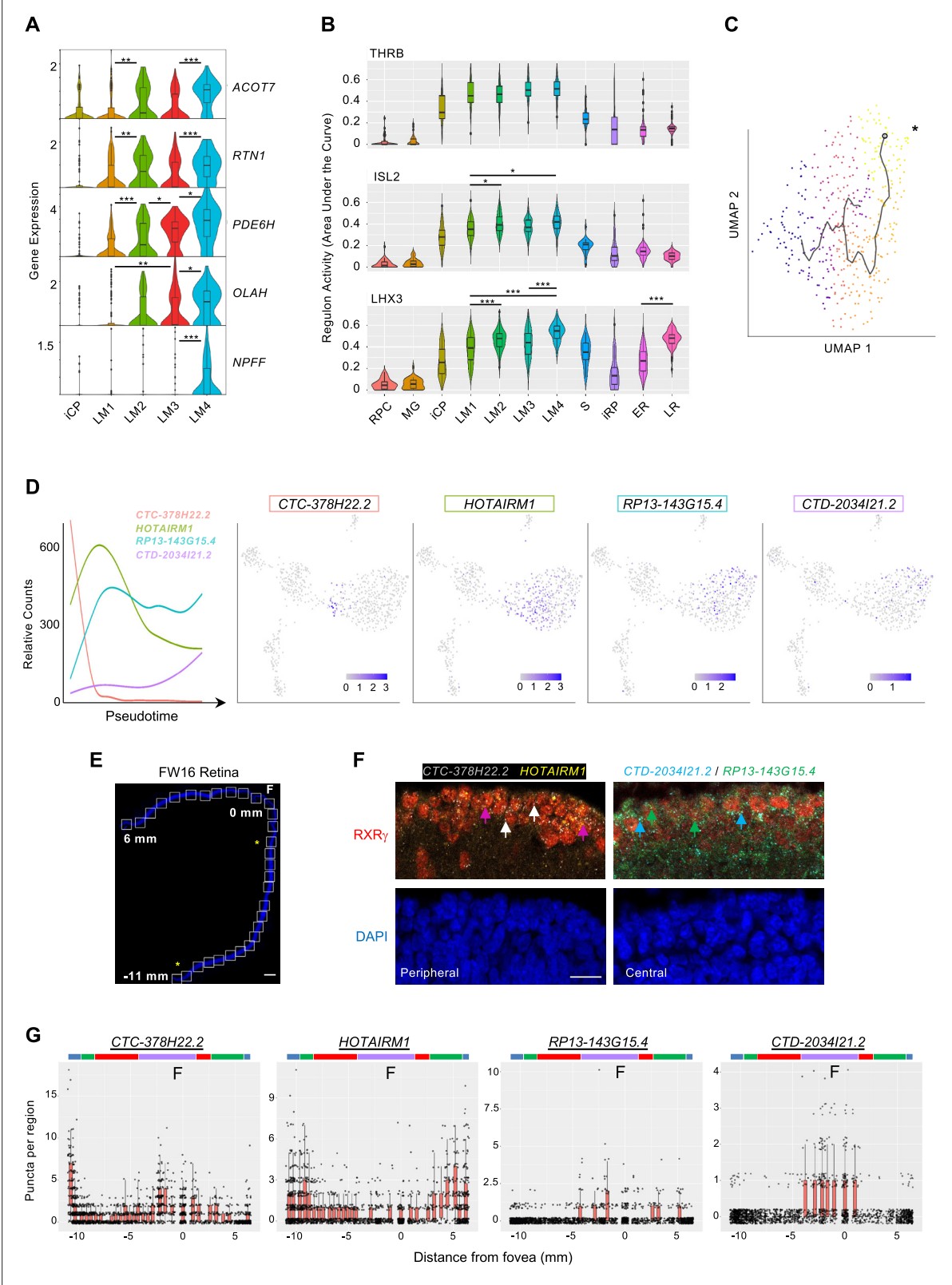

**Figure 9.** L/M cone subcluster marker genes, regulons, and pseudotemporal trajectory with successive lncRNA gene expression. (**A**) Violin plots of high-resolution cone cluster marker genes with increasing maturation-associated expression. All significant differences between adjacent clusters are indicated. (**B**) Violin plots of regulons with highest LM1-4 cone cluster specificity scores. *, p<0.05; **,<0.005; ***,<0.0005 (post-hoc Dunn test). (**C**) Pseudotemporal trajectory through the L/M cone population derived with Monocle 3. *: Root cell used to define endpoint. The pseudotime trajectory

*Figure 9 continued on next page*

*Figure 9 continued*

may be related to LM1-LM4 subcluster distributions in *Figure 5A*. (**D**) Trendlines of relative count expression (*left*) and UMAP plots for lncRNAs correlating with early or late-upregulating modules. Line color matched to labels. (**E, F**) Combined RXRγ immunohistochemical staining and FISH of lncRNAs on FW16 retina. (**E**) Tiled images of retina with nuclei stained with DAPI. White boxes: fields used for quantitative fluorescent imaging. Distances along the apical edge of tissue marked in mm from fovea to ciliary margins. Scale bar = 500 μm. Asterisks identify fields shown in F. (**F**) Combined RXRγ immunostaining and multiplex FISH for four lncRNAs, of which two are shown in peripheral and central retina regions. Arrows: White: RXRγ/*CTC-378H22.2*[+]. Magenta: RXRγ/*HOTAIRM1*[+]. Blue: RXRγ/*CTD-2034I21.2*[+]. Green: RXRγ/*RP13-143G15.4*[+]. Scale bar = 15 μm. (**G**) Quantitation of lncRNA fluorescent puncta assigned to RXRγ[+] cells after segmentation. Colored bars mark lncRNA expression regions as described in the text.

The online version of this article includes the following figure supplement(s) for figure 9:

**Figure supplement 1.** Heatmaps of coregulated gene modules across cone precursor pseudotime.

## Cone-intrinsic SYK expression associated with the proliferative response to pRB loss

We next assessed whether early-maturing cone transcriptomic features are conducive to the proliferative response to pRB loss (*Xu et al., 2014*; *Singh et al., 2018*). Given the high transcriptomic similarity across the L/M cone population, we compared gene expression between all early-maturing L/M cones and early-maturing rods. This identified 422 genes upregulated and 119 downregulated in cones ($p<0.05$, $log2FC>|0.4|$) (*Figure 10—figure supplement 1A*, *Supplementary file 1J*). Among cone-enriched genes, the top three enriched ontologies related to translation initiation, protein localization to membrane, and MYC targets (*Figure 10—figure supplement 1B*). The upregulation of MYC target genes was of interest given that many MYC target genes are also targets of MYCN, that MYCN protein is highly expressed in maturing (ARR3[+]) cone precursors but not in NRL[+] rods (*Figure 10A*), and that MYCN is critical to the cone precursor proliferative response to pRB loss (*Cobrinik, 2024*; *Xu et al., 2009*; *Xu et al., 2014*). Indeed, whereas *MYC* RNA was not detected, the LM cone cluster had increased *MYCN* RNA ($log_2FC = 0.54$) and MYCN regulon activity, representing the seventh highest LM cluster regulon specificity score (*Figure 10B and C*, *Supplementary file 1F*).

Among other differentially expressed genes, we noted the L/M cone-specific upregulation of *SYK* (*Figure 10D*, *Supplementary file 1J*), which encodes a non-receptor tyrosine kinase. Whereas SYK was previously implicated in retinoblastoma genesis and proposed to be induced in response to pRB loss (*Zhang et al., 2012*), its expression was not previously reported in developing fetal retina. Indeed, our scRNA-seq analyses revealed that *SYK* RNA expression increased from the iCP stage through cluster LM4, in contrast to its minimal expression in rods (*Figure 10E*). Moreover, *SYK* expression was abolished in the five-cell group with properties of late-maturing cones (characterized in *Figure 1E*), here displayed separately from the other LM4 cells and designated LM5 (*Figure 10E*). Similarly, immunohistochemical staining revealed high SYK protein expression in immature (ARR3[-]) and early-maturing (ARR3[+]) cones from the retinal periphery to the maturing foveal cones at FW16, while SYK was not detected in the most mature foveal cones at FW18 (*Figure 10F*). The loss of SYK protein and RNA expression with cone maturation is consistent with the lack of SYK in cones of normal retina adjacent to retinoblastoma tumors (*Zhang et al., 2012*) and implies that SYK is a defining feature of the human early cone precursor state. While *MYCN* RNA was also preferentially expressed in early cones, it did not increase as much relative to RPCs nor decline as much in late-maturing cone precursors when compared to *SYK* RNA dynamics (*Figure 10E*).

To determine if SYK might contribute to retinoblastoma initiation, dissociated fetal retinal cells were transduced with an *RB1*-directed shRNA (shRB1-733) known to induce cone precursor proliferation (*Xu et al., 2014*), treated with the selective SYK inhibitor GS-9876 (*Blomgren et al., 2020*) for 12 days, and examined for cone precursor cell cycle entry by co-staining for RXRγ and Ki67. GS-9876 treatment suppressed the proliferative response to pRB knockdown at all concentrations from 1.0 to 5.0 μM (*Figure 10G*), consistent with the notion that cone precursor intrinsic SYK activity contributes to the proliferative response to pRB loss and is retained in retinoblastoma cells (*Figure 10H*). However, given potential SYK inhibitor off-target effects, validation of the role of SYK in retinoblastoma initiation will require gene ablation studies.

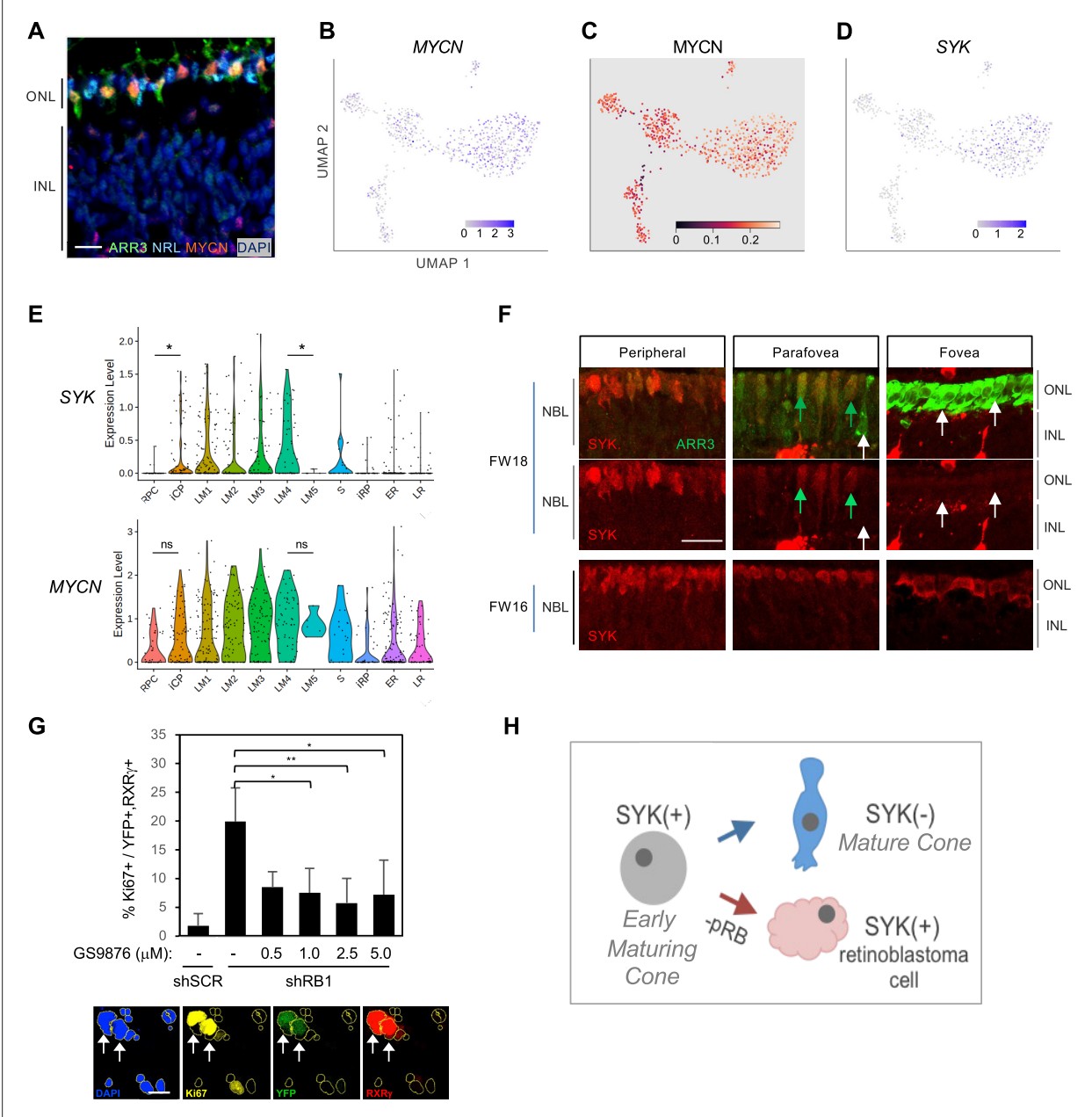

**Figure 10.** Cone intrinsic MYCN and SYK associated with proliferative response to pRB loss. (**A**) Immunofluorescent staining shows high MYCN in ARR3+ cones but not in NRL+ rods in FW18 retina. Scale bar = 10 μm. (**B–D**) UMAP plots of *MYCN* expression (**B**), MYCN regulon activity (**C**), and *SYK* expression (**D**). (**E**) *SYK* and *MYCN* gene expression violin plots by cluster. *, p<0.05; ns = not significant (t-test). (**F**) Immunohistochemical staining of SYK and cone arrestin (ARR3) in FW18 and FW16 retinae. Green arrow: ARR3+, SYK+. White arrow: ARR3+, SYK−. Scale bar = 25 μm. (**G**) *Top:* Effect of SYK inhibitor GS-9876 on Ki67 expression in RXRγ+ cones from FW16.5 retina co-transduced with YFP and shRB1- or control shSCR-shRNA. Values represent means of three analyses from two treatment replicates. Error bars: standard deviation. *, p<0.05; **,<0.005 (Student's T-test with equal variance, 2-tailed). RXRγ+ cells: Experiment 1, n=1340. Experiment 2, n=804. Range 107–366 cells per condition. *Bottom:* Example of Ki67, YFP, and RXRγ co-immunostaining with DAPI+ nuclei (yellow outlines). Arrows: Ki67+, YFP+, RXRγ+ nuclei. Scale bar = 20 μm. (**H**) Model of SYK expression in cone maturation and retinoblastoma development.

The online version of this article includes the following figure supplement(s) for figure 10:

**Figure supplement 1.** Differential gene expression in early cone and rod precursors.

# Discussion

This study evaluated cell state changes associated with human cone and rod photoreceptor development using deep full-length scRNA-seq. Whereas prior scRNA-seq studies provided insights into RPC fate determination and trajectories using 3' end counting (*Clark et al., 2019*; *Lu et al., 2020*; *Sridhar et al., 2020*; *Buenaventura et al., 2019*; *Lo Giudice et al., 2019*; *Zuo et al., 2024*; *Lukowski et al., 2019*), deep full-length sequencing enabled more precise resolution of cell states and identification of cell-type-specific regulon activities and transcript isoforms. Additionally, our cell enrichment strategy provided insight into the transcriptomic profiles and potential cancer-predisposing features of developing cones, a rare population with unique cancer cell-of-origin properties (*Cobrinik, 2024*; *Xu et al., 2014*; *Singh et al., 2018*).

One advantage of full-length scRNA-seq is that it enables the detection of differential transcript isoform expression. Here, we show that the rod determinant *NRL*, the L/M-cone determinant *THRB*, and the cone marker *RXRG* are all co-expressed in cone and rod precursors at the RNA level yet use distinct cell type-specific mechanisms to enable appropriate differential protein expression. Long-read cDNA sequencing revealed that L/M cones preferentially express novel *NRL* transcript isoforms predicted to encode a truncated NRL protein (Tr-NRL) lacking a transactivation domain as well as *NRL* isoforms with intra-exon-2 transcription initiation and alternative splicing. As with DD10 (*Rehemtulla et al., 1996*), Tr-NRL opposed transactivation by full-length NRL and thus may suppress rod-related transcription. However, our inability to detect intrinsic Tr-NRL protein or to overexpress Tr-NRL protein in cone precursors suggests there are additional layers of regulation, possible effects of untranslated Tr-NRL RNA, and alternative contexts in which these isoforms act. A similar assortment of *NRL* isoforms was expressed in rods, yet a far higher proportion of rod *NRL* transcripts encoded full-length (FL-) NRL protein. Thus, the L/M cone precursors' greater use of the Tr-NRL first exon and *NRL* exon 2 disruptions reveals a multipronged approach to suppress FL-NRL function while downregulating overall *NRL* RNA by only ~fourfold.

Similarly, we uncovered cell-type-specific differences in *THRB* transcript isoforms, albeit generated through premature transcription termination (PTT) and 3' UTR expression in late rod precursors. PTT is widely used to govern expression of transcription regulators (*Kamieniarz-Gdula and Proudfoot, 2019*), and while PTT-shortened *THRB* transcripts had been identified (*Master and Nauman, 2014*), their retinal cell specificity was not previously recognized. In contrast, the expression of *THRB* 3' UTR sequences independent of *THRB* protein-coding sequences was not previously reported and may enable additional regulation (*Mercer et al., 2011*). As a general matter, the expression of 3' UTR transcripts that are detached from protein-coding sequences may be inferred from full-length scRNA-seq but may be misinterpreted to represent protein-coding transcripts and thus confound the interpretation of scRNA-seq performed with 3' end-counting. To address this issue, our publicly available Shiny app displays exon coverage for all genes expressed in the RPC and photoreceptor clusters in this study.

One limitation of these studies is that individual cells may express a small and varying spectrum of transcript isoforms. While this variability was mitigated by combining cDNAs of cells deemed to be in similar states based on short-read sequencing, analysis of more cells might better reveal the spectrum of isoforms expressed by specific cell populations.

Understanding human photoreceptor development requires the identification of photoreceptor lineage developmental states and elucidation of mechanisms underlying their transitions. While cell states may be defined in various ways, at the transcriptome level they are perhaps best defined by unique combinations of transcription factor regulon activities (*Van de Sande et al., 2020*). Our deep sequencing approach uncovered discrete cell states distinguished by gene expression as well as by regulon activities. For example, early- and late-maturing rod populations were distinguished by the latter's increased expression of phototransduction genes together with increased activity of the NRL, ATF4, CRX, and LHX3 regulons and decreased HMX1 and RAX regulons (*Figure 2*). Similarly, L/M cones formed an early-maturing cone cluster characterized by high THRB and ISL2 regulons, consistent with THRB and ISL2 driving L/M cone identity (*Ng et al., 2011*; *Lu et al., 2020*; *Fischer et al., 2008*) and a late-maturing cone group with decreased RAX activity. Downregulation of the RAX regulon in late-maturing rod and cone precursors is consistent with decreasing RAX protein during photoreceptor maturation and with RAX-mediated suppression of the cone opsin and rhodopsin promoters (*Irie et al., 2015*; *Chen and Cepko, 2002*).

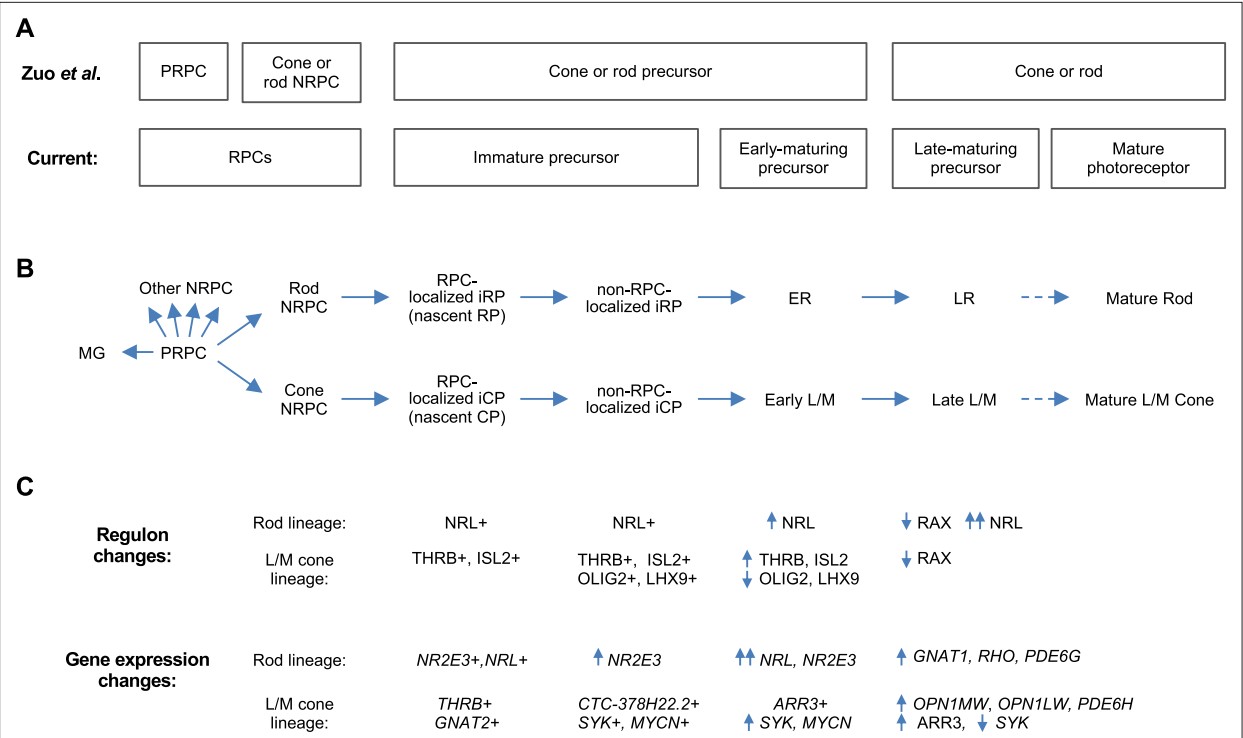

**Figure 11.** Proposed cell state trajectories in human L/M cone and rod development. (**A**) L/M cone and rod developmental stages discerned in the 3' snRNA-seq analysis of *Zuo et al., 2024* (*top*) and in the current deep, full-length scRNA-seq study (*bottom*). Note the discrimination of PRPC and NRPC populations in Zuo et al. and the discrimination of immature and early-maturing precursors in the current work. Late maturing precursors in fetal retina are hypothesized to be distinguishable from fully mature photoreceptors in postnatal retina. (**B**) RPC and L/M cone and rod developmental states referred to in the current study. (**C**) Selected cell state features identified in the current work.

Increased clustering resolution further divided the L/M cone cluster into four subgroups with subtle maturation-associated changes in marker gene expression and regulon activity, consistent with LM1-4 comprising different stages of a graded maturation process (*Figure 9*). However, trajectory analyses revealed successive expression of lncRNAs that was validated in developing retinal tissue. Three of these lncRNAs (*HOTAIRM1*, *CTD-2034I21.1* (neighbor gene to *CTD-2034I21.2*), and the mouse ortholog of *CTC-378H22.2* (D930028M14Rik)) were previously observed in cone scRNA-seq (*Lu et al., 2020*; *Buenaventura et al., 2019*; *Welby et al., 2017*). Their sequential expression in the absence of discrete regulon changes suggests that they demarcate early L/M cone precursor substates.

High-resolution clustering also segregated immature cone precursor (iCP) and immature rod precursor (iRP) populations, which were further partitioned according to their UMAP positions. *RPC-localized* iCP and iRP cells (those positioned adjacent to RPCs and MG in UMAP plots) lacked cell-cycle-related gene expression, showed higher L/M cone-like (THRB) or rod-like (NRL) gene expression and regulon activities, respectively, and had RNA velocities directed toward distinct cone and rod populations (*Figure 5*). iCPs also upregulated core photoreceptor regulons OTX2 and NEUROD1 to a greater extent than iRPs (*Figure 5—figure supplement 2C*), supporting their distinct trajectories. Prior droplet-based scRNA-seq analyses have not, to our knowledge, distinguished immediately post-mitotic immature cone and rod precursors from the subsequent early-maturing cone and rod precursor stages.

Whereas deep full-length scRNA-seq enabled discrimination of immature cone and rod precursors, 3' scRNA-seq and snRNA-seq enabled discrimination of the PRPC and cone- and rod-fated NRPCs (*Clark et al., 2019*; *Lu et al., 2020*; *Lyu et al., 2021*; *Zuo et al., 2024*; *Figure 11A*). Combining these observations supports a model in which cone- and rod-fated NRPCs give rise to immediately post-mitotic (i.e. *nascent*) and subsequent immature cone and rod precursors (iCPs and iRPs), which transition to early-maturing and then late-maturing L/M cone and rod precursors, with each state having unique gene expression and regulon properties (*Figure 11B and C*). It will be important

to corroborate this model with lineage tracing, to extend the model to S cones, and to determine whether late-maturing cone and rod states in fetal retina are distinct from mature post-natal cone and rod photoreceptors at the transcription factor regulon level.

Although not identified as a distinct cluster, our analyses also revealed cone and rod precursor subpopulations that co-express rod and cone genes and regulons (*Figures 6 and 7*, see also *Figure 5—figure supplement 3* for corroborating data of *Zuo et al., 2024*). In both the current full-length scRNA-seq and a recent 3' snRNA-seq analysis (*Zuo et al., 2024*), the earliest cone and rod precursors lacked such co-expression, implying that this property is acquired after cone versus rod fate-determination. In support of this notion, *NR2E3* RNA was detected in more mature central retina cone precursors but not in nascent peripheral cone precursors in fetal retina tissue (*Figure 7*). However, more information is needed to assess whether such co-expression serves a developmental purpose. For example, it is unknown if cone precursor *NR2E3* RNA ever produces NR2E3 protein, as in zebrafish retina, which could suppress cone-related gene expression (*Chen et al., 2005*) and delay terminal differentiation. Similarly, it is unclear if the *THRB* RNA expressed in rod precursors – which is largely truncated or comprised of non-coding 3' UTR sequences (*Figure 4*) – has direct RNA effects (*Mercer et al., 2011*).

Our characterization of cone and rod-related RNA co-expression may help resolve questions about the retinoblastoma cell of origin. Past studies suggested that retinoblastoma cells co-express RNAs associated with rods, cones, or other retinal cells due to a loss of lineage fidelity (*McEvoy et al., 2011*). However, the early L/M cone precursors' expression of *NR2E3* and *NRL* RNAs suggests that their presence in retinoblastomas (*McEvoy et al., 2011*; *Khanna et al., 2006*) reflects their normal expression in the L/M cone precursor cells of origin. This idea is further supported by the retinoblastoma cells' preferential expression of cone-enriched truncated *NRL* transcript isoforms (*Figure 3—figure supplement 2B*).

Our analyses also refine understanding of gene expression in the earliest cone and rod precursors. Early cone and rod precursors shared expression of neurogenic precursor markers *ATOH7*, *DLL3*, and the new marker *CHRNA1*, albeit with higher expression of each in the cone lineage and preferential expression of *ATOH7* and *CHRNA1* in the retinal periphery (*Figure 8 and Figure 8—figure supplement 1*). *ATOH7*, *DLL3*, and *CHRNA1* RNAs persist in some maturing cone and rod precursors, whereas *CTC-378H22.2* more precisely marks iCPs. iCPs with velocity directed towards early-maturing L/M cones expressed *THRB* and *ONECUT1* RNAs and had OLIG2 regulon activity (*Figure 8—figure supplement 2*); as these elements were proposed to promote cone fate in lineage-restricted RPCs (*Emerson et al., 2013*), their expression in iCPs suggests a similar role, such as in L/M cone fate determination, in post-mitotic cells.

We also mined gene expression differences in early cone versus rod photoreceptors to identify factors that impact human cone sensitivity to *RB1* loss (*Xu et al., 2014*; *Singh et al., 2018*). We detected upregulation of genes targeted by MYC, some of which are similar to those targeted by MYCN (*Baluapuri et al., 2020*), along with upregulated *MYCN* RNA, MYCN protein, and MYCN regulon activity. As MYCN is required for the proliferative response to pRB loss (*Xu et al., 2014*) and triggers retinoblastoma when amplified and overexpressed (*Cobrink, 2024*), these findings suggest that increased *MYCN* RNA expression and regulon activity contribute to pRB-deficient cone precursor proliferation and retinoblastoma genesis. We further found that cone precursors highly express SYK, an oncoprotein previously detected in human retinoblastomas and *RB1*-null retinal organoids but not in healthy tumor-associated retina (*Zhang et al., 2012*; *Liu et al., 2020*). The high SYK expression preceding early cone precursor maturation and its loss during late maturation implies that high-level SYK expression is a retinoblastoma cell-of-origin-specific feature. Moreover, the pRB-depleted cone precursors' sensitivity to a SYK inhibitor suggests that native SYK expression rather than de novo induction contributes to the cone precursors' initial proliferation (*Figure 10H*), although genetic ablation of *SYK* is needed to confirm this notion.

In summary, through deep, full-length RNA sequencing, we identified photoreceptor cell-type-specific differences in transcript isoform expression and post-mitotic photoreceptor precursor states with distinctive gene expression and regulon activities. The discrimination of these states enabled the identification of developmental stage-specific cone precursor features associated with the cone precursors' predisposition to form retinoblastoma tumors.

# Materials and methods

## Key resources table

| Reagent type (species) or resource | Designation | Source or reference | Identifiers | Additional information |
|---|---|---|---|---|
| Strain, strain background (*Escherichia coli*) | NEB 10-beta | New England Biolabs | C3019H | Competent cells |
| Cell line (*Homo sapiens*) | CHLA-VC-RB31 | *Stachelek et al., 2023*; https://doi.org/10.1002/gcc.23120 | | Authenticated by STR analysis |
| Cell line (*Mus musculus*) | NIH-3T3 | American Type Culture Collection (ATCC) | CRL-1658.2 | Authenticated by STR analysis |
| Cell line (*Homo sapiens*) | HEK-293T | American Type Culture Collection (ATCC) | CRL-11268 | Authenticated by STR analysis |
| Biological sample (*Human*) | Fetal eyes | Family Planning Associates, Los Angeles, CA | | Isolated from fetal tissue |
| Biological sample (*Human*) | Fetal eyes | Advanced Bioscience Resources, Alameda, CA | | Isolated from fetal tissue |
| Antibody | anti-NRL (Goat polyclonal) | R&D Systems | CAT# AF2945, RRID:AB_2155098 | WB: 1:2000–4000 |
| Antibody | anti-NRL (Mouse monoclonal) | Santa Cruz Biotechnology | CAT# SC-374277, RRID:AB_10991100 | WB:1:250, IF:1:50 |
| Antibody | anti-RXRγ (Mouse monoclonal) | Santa Cruz Biotechnology | CAT# SC-514134, RRID:AB_2737293 | IF: 1:200 |
| Antibody | anti-SYK (Mouse monoclonal) | Santa Cruz Biotechnology | CAT# SC1240, RRID:AB_628308 | IF:(1:200) |
| Antibody | anti-ARR3 (Rabbit polyclonal) | *Zhang et al., 2001*; *Li et al., 2003*; https://doi.org/10.1167/iovs.02-0434; *Lou et al., 2012*; https://doi.org/10.1007/978-1-4615-1355-1_33 | LUMI-F - hCAR | IF: (1:5000) Cheryl Craft |
| Antibody | anti-RXRγ (Rabbit polyclonal) | Santa Cruz Biotechnology | CAT# SC-555, RRID:AB_2269865 | IF: (1:800) |
| Antibody | anti-KI67 (Mouse monoclonal) | BD Bioscience | CAT# 550609, RRID:AB_393778 | IF: (1:200) |
| Antibody | anti-GFP and YFP (Goat polyclonal) | Abcam | CAT# ab6673, RRID:AB_305643 | IF: (1:500) |
| Antibody | anti-CD133-PE (Mouse monoclonal) | Miltenyi Biotec | CAT# 130-113-108, RRID:AB_2725937 | FACS: (1:50) |
| Antibody | Mouse monoclonal anti-CD44-FITC (1:50) | BD Biosciences | CAT# 555478, RRID:AB_395870 | FACS: (1:50) |
| Antibody | Mouse monoclonal anti-PNR/NR2E3 (1:50) | R&D Systems | CAT# PP-H7223-00, RRID:AB_ 2155481 | IF: (1:50) |
| Antibody | Mouse monoclonal anti-CD49b-FITC (1:10) | BD Biosciences | CAT# 555498, RRID:AB_395888 | FACS: (1:10) |
| Antibody | anti-goat IgG Alexa Fluor 488 (1:300) | Jackson Laboratories | CAT# 705-545-147, RRID:AB_2336933 | IF: (1:300) |
| Antibody | anti-mouse IgG Alexa Fluor 680 (Donkey polyclonal) | Life Technologies | CAT# A10038, RRID:AB_2534014 | IF: (1:500) |
| Antibody | Donkey polyclonal anti-mouse IgG Alexa Fluor 680 (Donkey polyclonal) | Invitrogen | CAT# A11057, RRID:AB_2534104 | IF: (1:500) |
| Recombinant DNA reagent | pLKO.1C-YFP-shSCR | *Lee and Cobrinik, 2020*; https://doi.org/10.2144/btn-2019-0155 | RRID:Addgene_139647 | |

*Continued on next page*

*Continued*

| Reagent type (species) or resource | Designation | Source or reference | Identifiers | Additional information |
|---|---|---|---|---|
| Recombinant DNA reagent | pLKO.1C-YFP-shRB1-733 | *Lee and Cobrinik, 2020*; https://doi.org/10.2144/btn-2019-0155 | RRID:Addgene_244458 | |
| Recombinant DNA reagent | pcDNA4-His-Max-C-Nrl | *Cheng et al., 2004*; https://academic.oup.com/hmg/article-abstract/13/15/1563/581552?redirectedFrom=fulltext&login=false | | Gift from A. Swaroop. |
| Recombinant DNA reagent | pcDNA4-His-Max-C-EF1α-FL-NRL | This paper | RRID:Addgene_239094 | See Materials and methods NRL isoform analyses |
| Recombinant DNA reagent | pcDNA4-C-EF1α-FL-NRL | This paper | RRID:Addgene_239095 | See Materials and methods NRL isoform analyses |
| Recombinant DNA reagent | pcDNA4-C-EF1α-Tr-NRL | This paper | RRID:Addgene_239096 | See Materials and methods NRL isoform analyses |
| Recombinant DNA reagent | pcDNA4-C-EF1α | This paper | RRID:Addgene_239097 | See Materials and methods NRL isoform analyses |
| Recombinant DNA reagent | pGL3-SV40 | Promega | AT# E1761, RRID:Addgene_173953 | |
| Recombinant DNA reagent | pGL3-PDE6B-146 | This paper | RRID:Addgene_239098 | See Materials and methods NRL isoform analyses |
| Recombinant DNA reagent | pGL3-empty | This paper | RRID:Addgene_239099 | See Materials and methods NRL isoform analyses |
| Recombinant DNA reagent | pUltra-EGFP-P2A-Tr-NRL | This paper | RRID:Addgene_239100 | See Materials and methods NRL isoform analyses |
| Recombinant DNA reagent | pUltra-EGFP | *Lou et al., 2012* | RRID:Addgene_24129 | |
| Sequence-based reagent | Gipc1_F | This paper | PCR primers | GGGAAAGGACAAAAGGAACCC |
| Sequence-based reagent | Gipc1_R | This paper | PCR primers | CAGGGCATTTGCACCCCATGCC |
| Sequence-based reagent | Subcloning PCR primer | Integrated DNA Technologies | Del-His F | 5'- CCGAAACCATGGCCCTGCCCCCCAGC |
| Sequence-based reagent | Subcloning PCR primer | Integrated DNA Technologies | Del-His R | 5'- GGGCCATGGTTTCGGAGGCCGTCCG |
| Sequence-based reagent | Subcloning PCR primer | Integrated DNA Technologies | NRL-no-His F | 5'- CCGAAACCATGTCTGTGCGGGAGCTAAACC |
| Sequence-based reagent | Subcloning PCR primer | Integrated DNA Technologies | NRL-no-His R | 5'- CAGACATGGTTTCGGAGGCCGTCCG |
| Sequence-based reagent | Subcloning PCR primer | Integrated DNA Technologies | pcDNA del NRL F | 5'- CCGAAACCGCCGTTCAGAGCACCTTGTGG |
| Sequence-based reagent | Subcloning PCR primer | Integrated DNA Technologies | pcDNA del NRL R | 5'- GAACGGCGGTTTCGGAGGCCGTCCG |
| Sequence-based reagent | Subcloning PCR primer | Integrated DNA Technologies | PDE –93 F IF | 5'- TCTTACGCGTGCTAGAGC GCAGGCCCCCATTTG |
| Sequence-based reagent | Subcloning PCR primer | Integrated DNA Technologies | PDE +53 R IF | 5'- CTTAGATCGCAGATCGGT GGCTGCCTGTCCCTG |
| Sequence-based reagent | Subcloning PCR primer | Integrated DNA Technologies | pGL3-sv40 SDM F | 5'- CTGCGATCAAGCTTGGCA TTCCGGTACTG |
| Sequence-based reagent | Subcloning PCR primer | Integrated DNA Technologies | pGL3-sv40 SDM R | 5'- CAAGCTTGATCGCAGATCGGTGGCTG |
| Sequence-based reagent | Subcloning PCR primer | Integrated DNA Technologies | pGL3 del SV40 R | 5'- CAAGCTTGATCGCAGATCTCGAGCCC |

*Continued on next page*

*Continued*

| Reagent type (species) or resource | Designation | Source or reference | Identifiers | Additional information |
|---|---|---|---|---|
| Sequence-based reagent | Subcloning PCR primer | Integrated DNA Technologies | Tr-NRL IF pU-G F | 5'- GCCTTCTAGAGGATCCATGT CTGTGCGGGAGCTAAACC |
| Sequence-based reagent | Subcloning PCR primer | Integrated DNA Technologies | Tr-NRL IF pU-G R | 5'-- CGCCGGAGCCGGATCCTCAGA GGAAGAGGTGGGAGGG |
| Sequence-based reagent | SmartSeq library amplification primer | Integrated DNA Technologies | 5' PCR Primer II A | 5'- AAGCAGTGGTATCAACGCAGAGT |
| Sequence-based reagent | RACE primer | Integrated DNA Technologies | 5' RACE primer for SmartSeq cDNA | 5'- AAGCAGTGGTATCAACGCAGAGTACGGG |
| Sequence-based reagent | NRL reverse primer for 5' RACE | Integrated DNA Technologies | Rev NRL-Full ex3 | 5'- GGTTTAGCTCCCGCACAGACATCGAGAC |
| Sequence-based reagent | RXRG reverse primer for 5' RACE | Integrated DNA Technologies | Rev RXR ex5 | 5'- GAAGAACCCTTTGCAGCCTTCACAACTG |
| Sequence-based reagent | RACE primer | Integrated DNA Technologies | 3' RACE primer for SmartSeq cDNA | 5'- AAGCAGTGGTATCAACGCAGAGTACTTTT |
| Sequence-based reagent | THRB1 forward primer for 3' RACE | Integrated DNA Technologies | Forw TRb1 ex4 | 5'- GCCTTACAGCCTGGGACAAACC |
| Sequence-based reagent | THRB2 forward primer for 3' RACE | Integrated DNA Technologies | Forw TRb2 ex1 | 5'-CCCTGGAAACATGTTTAAAAGCAAGGACT |
| Sequence-based reagent | In situ hybridization chain reaction probes | Molecular Instruments Inc | FL-NRL exon 1, 2 | Target sequence Ensembl ID: ENST00000619224.1 |
| Sequence-based reagent | hybridization chain reaction probes | Molecular Instruments Inc | *CHRNA1* | Target sequence NCBI accession number: NM_000079.4 |
| Sequence-based reagent | hybridization chain reaction probes | Molecular Instruments Inc | *RP13-143G15.4* | |
| Sequence-based reagent | hybridization chain reaction probes | Molecular Instruments Inc | *CTC-378H22.2* | Target sequence Ensembl ID: ENST00000559786.1 |
| Sequence-based reagent | hybridization chain reaction probes | Molecular Instruments Inc | *HOTAIRM1* | Target sequence NCBI accession number: NR_038366.1 |
| Sequence-based reagent | hybridization chain reaction probes | Molecular Instruments Inc | *CTD-2034I21.2* | Target sequence NCBI accession number: XR_001752169.1 |
| Sequence-based reagent | hybridization chain reaction probes | Molecular Instruments Inc | *GNAT2* | Target sequence NCBI accession number: NM_005272.5 |
| Sequence-based reagent | hybridization chain reaction probes | Molecular Instruments Inc | *NR2E3* | Target sequence NCBI accession number: NM_01249.4 |
| Commercial assay or kit | In-Fusion HD Cloning | Clontech | Clontech:639647 | |
| Commercial assay or kit | SMARTer Ultra Low RNA Kit for the Fluidigm C1 System | Clontech (Takara Bio) | CAT# 634835/634935 | |
| Commercial assay or kit | SMART-Seq V4 Ultra Low Input RNA Kit | Takara Bio | CAT# 63891 | |
| Commercial assay or kit | Nextera XT DNA Library Preparation Kit | Illumina | CAT# FC-131–1096 | |
| Commercial assay or kit | CloneAmp HiFi PCR Premix | Clontech (Takara Bio) | Takara Bio #639298 | |
| Commercial assay or kit | Quant-iT PicoGreen dsDNA Assay Kit | Life Technologies | CAT# P11496 | |
| Commercial assay or kit | Nano-Glo Dual-Luciferase Reporter Assay System | Promega, Inc | CAT# N1610 | |
| Chemical compound, drug | GS-9876 (SYK inhibitor) | MedChemExpress | CAT# HY-109091 | |

*Continued on next page*

*Continued*

| Reagent type (species) or resource | Designation | Source or reference | Identifiers | Additional information |
|---|---|---|---|---|
| Software, algorithm | TrimGalore | *Krueger, 2018* | RRID:SCR_011847 | https://github.com/FelixKrueger/TrimGalore |
| Software, algorithm | Cutadapt | *Martin, 2011*; https://journal.embnet.org/index.php/embnetjournal/article/view/200 | | https://cutadapt.readthedocs.io/en/stable/ |
| Software, algorithm | HISAT2 | *Kim et al., 2019*; https://www.nature.com/articles/s41587-019-0201-4 | RRID:SCR_015530, version 2.1.0 | https://github.com/DaehwanKimLab/hisat2 |
| Software, algorithm | StringTie | *Pertea et al., 2015*; https://www.nature.com/articles/nbt.3122 | | https://ccb.jhu.edu/software/stringtie/ |
| Software, algorithm | Snakemake/ARMOR | *Orjuela et al., 2019*; https://www.nature.com/articles/nbt.3122; *Soneson et al., 2025* | | https://github.com/cobriniklab/ARMOR |
| Software, algorithm | Minimap2 | *Li, 2018*; https://academic.oup.com/bioinformatics/article/34/18/3094/4994778?login=false | | https://lh3.github.io/minimap2/ |
| Software, algorithm | WebGestalt | *Liao et al., 2019*; https://academic.oup.com/nar/article/47/W1/W199/5494758?login=false | | http://www.webgestalt.org/ |
| Software, algorithm | pySCENIC | *Van de Sande et al., 2020*; *Van de Sande and Flerin, 2025*; https://www.nature.com/articles/s41596-020-0336-2 | RRID:SCR_025802 | https://github.com/aertslab/pySCENIC |
| Software, algorithm | Tximport | *Soneson et al., 2015*; https://f1000research.com/articles/4-1521/v1 | | https://bioconductor.org/packages/release/bioc/html/tximport.html |
| Software, algorithm | Seurat v3 (full-length scRNA-seq), Seurat v5 (3' snRNA-seq) | *Stuart et al., 2019*; https://www.cell.com/cell/fulltext/S0092-8674(19)30559-8?_returnURL=https%3A%2F%2Flinkinghub.elsevier.com%2Fretrieve%2Fpii%2FS0092867419305598%3Fshowall%3Dtrue | | https://satijalab.org/seurat/index.html |
| Software, algorithm | EnhancedVolcano | *Blighe et al., 2018* | RRID:SCR_018931 | https://github.com/kevinblighe/EnhancedVolcano |
| Software, algorithm | genesorteR (v0.4.3) | *Ibrahim and Kramann, 2019*; *Ibrahim, 2021*; https://doi.org/10.1101/676379 | | http://github.com/mahmoudibrahim/genesorteR |
| Software, algorithm | Wiggleplotr (v1.13.1) | *Alasoo, 2025*; *Alasoo, 2022*; Bioconductor package | | https://github.com/kauralasoo/wiggleplotr |
| Software, algorithm | DEXSeq | *Anders et al., 2012*, https://genome.cshlp.org/content/22/10/2008 | | https://bioconductor.org/packages/release/bioc/html/DEXSeq.html |
| Software, algorithm | velocytoR | *La Manno et al., 2018*; https://www.nature.com/articles/s41586-018-0414-6 | | https://velocyto.org/ |
| Software, algorithm | Monocle3 | *Cao et al., 2019*; https://www.nature.com/articles/s41586-019-0969-x | | https://cole-trapnell-lab.github.io/monocle3/ |
| Software, algorithm | QuPath | *Bankhead et al., 2017*; https://www.nature.com/articles/s41586-019-0969-x | | https://qupath.github.io |
| Software, algorithm | StarDist (QuPath implementation) | *Schmidt et al., 2018*; https://www.nature.com/articles/s41586-019-0969-x | | https://qupath.readthedocs.io/en/stable/docs/advanced/stardist.html |

## Primary human tissue

Human fetal samples were provided by Advanced Bioscience Resources, Inc, Alameda, CA, or collected under Institutional Review Board approval at the University of Southern California (protocol HS-13–00399), and Children's Hospital Los Angeles (protocol CHLA-14–00122). Following the patient decision for pregnancy termination, patients were offered the option of donation of the products of conception for research purposes, and those that agreed signed an informed consent. This did not alter the choice of termination procedure, and the products of conception from those who declined participation were disposed of in a standard fashion. Fetal age was determined according to the American College of Obstetrics and Gynecology guidelines (*Pettker et al., 2017*). Each retina subjected to sequencing was from a different donor.

## Cell lines

CHLA-VC-RB31 was produced as in Stachelek et al., and HEK-293T and NIH-3T3 cells obtained from the American Type Culture Collection (ATCC). CHLA-VC-RB31 and HEK-293T were authenticated by short tandem repeat (STR) analysis and all cell lines tested (negative) for mycoplasma by the University of Arizona Genetics Core, which has the original CHLA-VC-RB31 and HEK-293T STR profile records. NIH-3T3 cells were authenticated by STR analysis by the ATCC Mouse Cell Authentication Service.

## Retina dissociation and RNA-sequencing

### Single cell isolation

Retinae were dissected while submerged in ice-cold phosphate-buffered saline (PBS) and dissociated as described (*Xu et al., 2014*). Briefly, the retina was removed and placed in 200 µl Earle's balanced salt solution (EBSS) in a six-well plate on ice. 2 ml of 10 u/ml papain (Worthington LK003176) solution was added and then incubated at 37 °C for 10 min, followed by pipetting with a 1000 µl pipet tip to break up large pieces and additional 5 min incubations until cells were dissociated to single cell level. An additional 1–2 ml papain was added after 20 min if dissociation was insufficient (~40 min total). Retina culture media [Iscove's Modified Dulbecco's Medium with glutamine (Corning, #12440046), 10% fetal bovine serum (FBS, FB-01, Omega Scientific), 0.28 U/mL insulin (Eli Lilly NDC 0002-0213-01), 55 µM β-mercaptoethanol (Fisher Scientific, # 21985023), penicillin and streptomycin] (Fisher Scientific, # MT30002CI) was used to stop enzyme activity and cells were centrifuged in 14 ml round bottom tubes at 300 x g, 4 °C for 10 min. Supernatants were centrifuged at 1100 x *g* for an additional 3 min. After resuspension in Hank's balanced salt solution (Fisher Scientific, 14025092), a 10 µl volume of cells was combined with 10 µl trypan blue and counted using a hemocytometer.

### FACS isolation of single cells and cDNA synthesis

Single cells were FACS-isolated as described (*Xu et al., 2014*) with differences as follows. Dissociated cells were centrifuged and resuspended in a 5% fetal bovine serum in PBS (FBS/PBS) with CD133-PE (Miltenyi Biotec 130-113-108), CD44-FITC (BD Pharmingen 555478), and CD49b-FITC antibodies (BD Pharmingen 555498) to a final concentration of 10,000 cells/µl. After incubation at room temperature (RT) for 1 h, samples were diluted in 5% FBS/PBS, centrifuged as above, and resuspended in 300–400 µl 1% FBS/PBS containing 10 µg/ml 4',6-diamidino-2-phenylindole (DAPI). Single cells were sorted on a BD FACSAria I at 4 °C using 100 µm nozzle in single-cell mode into each of eight 1.2 µl lysis buffer droplets on parafilm-covered glass slides, with droplets positioned over pre-defined marks (S. Lee et al., *in preparation*). Gating removed low forward- and side-scatter cells before collecting a CD133-high, CD44/49b-low population. Upon collection of eight cells per slide, droplets were transferred to individual low-retention PCR tubes (eight tubes per strip; Bioplastics K69901, B57801) pre-cooled on ice to minimize evaporation. The process was repeated with a fresh piece of parafilm for up to 12 rounds to collect 96 cells. cDNA was prepared and amplified using the SMART-Seq V4 Ultra Low Input RNA Kit (Takara Bio 634891) in 10 x reduced volume reactions using a Mantis liquid handling system (S. Lee et al., *in preparation*). Samples were stored at –20 °C until quantitation and library preparation. All RNA/cDNA volumes during FACS and processing were handled using low retention siliconized pipette tips (VWR 53503–800, 535093–794).

## C1 isolation of single cells and cDNA synthesis

Dissociated cells were FACS isolated as above and collected into a single 1.5 ml tube. Cells were centrifuged and resuspended at 400–800 cells/µl, combined with C1 suspension reagent, and loaded onto the C1 chip and imaged at each site to define cell number in each well. cDNA was synthesized using SMARTer chemistry (SMARTer Ultra Low RNA Kit for the Fluidigm C1 System, Clontech 634835/634935; three retinae, Seq 1, FW17, and FW13-1), or SMART-Seq V4 (all others). Final cDNA was harvested into low-retention tube strips and stored at –20 °C.

## Quality control, library preparation, and sequencing

DNA was quantitated using Quant-iT PicoGreen dsDNA assay (Life Technologies, P11496) and quality checked by Bioanalyzer. Libraries >0.05 ng/µl were prepared using the Nextera XT DNA Library Preparation Kit (Illumina, FC-131–1096) and sequenced on the Illumina NextSeq 500 (2x75) (Seq 1) or on Illumina HiSeq 4000 (2x75) (all others).

## 5' and 3' RACE and nanopore sequencing

1 µl of amplified single cell cDNAs libraries produced as above (23 ER cells, 21 LM cells, and 5 LR cells) were re-amplified using the 5' PCR Primer II A (described for Clontech SmartSeq V4 and purchased from Integrated DNA Technologies) and using CloneAmp HiFi PCR Premix (Clontech Laboratories, Inc Takara Bio #639298; *Supplementary file 1I*) by heating to 98 °C x 2 min followed by 20 cycles of 98 °C x 10 s, 65 °C x 15 s, and 72 °C x 35 s, followed by 72 °C x 5 min. Re-amplified cDNAs were quantified using Qubit Flex Fluorometer (InVitrogen) and 50 pg of each re-amplified library was used for separate RACE PCR reactions with gene-specific *NRL, RXRγ, TRβ1, and TRβ2* primers and universal 5' RACE and 3' RACE primers (*Supplementary file 1I*) using CloneAmp HiFi PCR Premix by heating to 98 °C x 2 min followed by 30 cycles of 98 °C x 10 s, 60 °C x 15 s, and 72 °C x 35 s, followed by 72 °C x 5 min. The '5' RACE primer for SmartSeq cDNA' sequence (5'–AAGCAGTGGTATCAACGCAGAGTACGGG–3') was based on the SMART-Seq V4 SMARTer II A oligonucleotide (5'–AAGCAGTGGTATCAACGCAGAGTA CXXXXX–3') after library sequencing revealed that AAGCAGTGGTATCAACGCAGAGTAC was most often followed by GGGNN. The '3' RACE primer for SmartSeq cDNA' sequence (5'- AAGCAGTGGTAT CAACGCAGAGTACTTTT-3') was based on the SMART-Seq V4 3' SMART CDS Primer II A (5'–AAGC AGTGGTATCAACGCAGAGTACT(30)(N-1)N–3') but with only four Ts at the 3' end. Equal volumes of RACE PCR products were pooled according to cell type, and DNA concentrations of LM (32.8 ng/µl), ER (22.6 ng/µl), and LR (16.5 ng/µl) samples were determined using Qubit fluorometer. Long read sequencing libraries were prepared and samples barcoded using Oxford Nanopore Technology (ONT) Native Barcoding Kit 24 V14 as specified by the manufacturer for average DNA fragment lengths of ~2 kb. 200–300 ng of each DNA pool was subjected to repair/dA-tailing (End-Prep) with NEB Ultra II End-Prep Enzyme Mix (New England Biolabs, Ipswich, MA) followed by Native Barcode Ligation as specified by ONT with NB01 assigned to the ER pool, NB02 assigned to the LM pool, and NB03 assigned to the LR pool. Ligation reactions were quenched by addition of 4 µl EDTA and each reaction mixture (24 µl) pooled for a 72 µl volume. The final pool was incubated with Native Adapter (NA) and NEB Quick T4 ligase. The adapter-ligated pool was recovered by AMPure XP magnetic beads and then loaded onto a PromethION flow cell (R10.4.1) as specified by the manufacturer. Sequencing was performed on a PromethION 24 system using MinKNOW software for 72 hr. Raw FAST5 data files were converted to FASTQ files using ONT MinKNOW base calling software.

## Computational methods

### Software packages used in this study are described in the Key Resources Table

#### Read processing and dimensionality reduction

Adapter sequences were removed using the *trimgalore* wrapper for *Cutadapt* (*Martin, 2011*). Trimmed FASTQ files were used as input for *HISAT2* (*Kim et al., 2019*) with a non-canonical splice penalty of 20, maximum and minimum penalties for mismatch of 0 and 1, and maximum and minimum penalties for soft-clipping set to 3 and 1. Aligned bam files were quantified with *StringTie* (*Pertea et al., 2015*) and *Tximport* (*Soneson et al., 2015*), yielding cell-by-transcript and cell-by-gene count matrices annotated according to Ensembl build 87. All of the above operations can be reproduced

using a custom snakemake (*Orjuela et al., 2019*) workflow accessible at https://github.com/cobrin-iklab/ARMOR (copy archived at *Stachelek, 2025*).

For analysis of short-read sequencing of full-length scRNA-seq, dimensionality reduction and data visualization were performed using the Seurat (V3) toolset (*Butler et al., 2018*; *Stuart et al., 2019*). Expression counts from all sequencing datasets were integrated using Seurat's standard integration workflow. Seven sample sequencing batches were integrated after default normalization and scaling using the top 2000 most variable features in each set to identify anchor features. Normalized read counts for gene expression are reported as raw feature read counts divided by total cell read counts, then multiplied by a scaling factor of 10,000 and natural log transformed. These features of the integrated dataset were used to calculate principal component analysis (PCA) and UMAP embeddings (*McInnes et al., 2018*; *Becht et al., 2019*). A nearest-neighbors graph was constructed from the PCA embedding and clusters were identified using a Louvain algorithm at low and high resolutions (0.4 and 1.6; *Blondel et al., 2008*). Read coverage plots for genes of interest were generated using *wiggleplotr* (*Alasoo, 2025*) to visualize BigWig files with ENSEMBL isoforms. Individual exon counts were identified using *DEXseq* (*Anders et al., 2012*), which takes exons from available isoforms and bins them into intervals before assigning any read counts that overlap a bin to that same bin. These counts were normalized for length for calculation of fold change or relative difference of exon use. Reads crossing target splice donor sites were evaluated as spliced or unspliced from BAM files to determine intron readthrough.

For analysis of 3' snRNA-seq, processed data provided with *Zuo et al., 2024* (CZ CELLxGENE Discover with accession code 5900dda8-2dc3-4770- b604084eac1c2c82) was downloaded as a Seurat Object. A pre-processed AnnData object containing inferred NRPC fate annotations was generously provided by Dr. Rui Chen, cone and rod fated NRPC annotations were transferred to the primary Seurat Object, and data visualization was performed using the Seurat (V5) toolset (*Butler et al., 2018*; *Stuart et al., 2019*).

Analysis of long-read sequencing data was performed with the nf-core nanoseq version 3.1.0 pipeline (*Ewels et al., 2020*) (https://nf-co.re/nanoseq/3.1.0). Quality control on raw reads was conducted using FastQC. Reads were aligned using minimap2 (*Li, 2018*). SAM files were converted to coordinate-sorted BAM files and mapping metrics were obtained using SAMtools. bigWig coverage tracks were created for visualization using BEDTools and bedGraphToBigWig. bigBed coverage tracks were created using BEDTools and bedToBigBed. Transcripts were reconstructed and quantified using bambu. Quality control results for raw reads and alignment results were presented using MultiQC. All reads whose ends overlapped gene-specific RACE primers were displayed and quantitated using the Integrated Genome Browser (IGV). For *THRB* 3' RACE reactions, reads inferred to initiate through exonic internal oligo(dT) priming were removed based on the following features adjacent to the 3' end of the alignments: six continuous adenines (As), more than seven As in a 10 nucleotide window, AG-runs of six or more nucleotides, eight or more As or Gs in a 10 nucleotide window, eight or more As or high A/T content (27 out of 30 bases), or 12 or more adenines present in an 18 nt window (*Svoboda et al., 2022*). The proportion of alternative *NRL* exon 2 splicing was calculated based on the maximum read coverage between the exon 2 splice acceptor and the first splice donor and the minimum coverage after the internal splice donor as reported in IGV.

## Differential expression and overrepresentation analyses

Marker features for each cluster were identified using the Wilcoxon rank sum tests and specificity scores were computed using the *genesorteR* (*Ibrahim and Kramann, 2019*) R package. Only genes with an adjusted p-value (pAdj) <0.05 were considered for further analyses. Differential expression analysis of full-length scRNA-seq was performed using the Seurat FindMarkers function with the Wilcoxon rank sum test on log-normalized counts. Results were displayed as volcano plots made with *EnhancedVolcano* (*Blighe et al., 2018*). Differential expression analysis of a 3' snRNA-seq was performed using the Seurat FindMarkers function with the Wilcoxon rank sum test on log-normalized counts. Results were displayed as volcano plots made with *EnhancedVolcano* (*Blighe et al., 2018*).

Overrepresentation analyses were performed with WebGestalt (*Liao et al., 2019*). Analyses were run with the default settings (5–2000 genes per category, Benjamini-Hochberg multiple-testing correction), gene list enrichment was compared to the genome reference set and ontologies were displayed with a false discovery rate (FDR)<0.05 and weighted set cover redundancy reduction. All

gene sets evaluated were provided within WebGestalt and the same three were used unless otherwise indicated: GO – Biological Process, KEGG, and Hallmark50.

### Transcription factor regulons

Transcription factor regulons were identified using pySCENIC, the python-based implementation of Single-Cell Regulatory Network Inference and Clustering (SCENIC; *Van de Sande et al., 2020*; *Aibar et al., 2017*). The tool was run using the basic settings as shown (https://pyscenic.readthedocs.io/en/latest/tutorial.html). Initial filtering required a gene to have a minimum of 3 raw counts in 1% of cells to be considered for inclusion in a regulon. AUC scores were z-score normalized for comparison between regulons.

### Trajectory analysis

RNA velocity analysis was performed with R package *velocyto* (*La Manno et al., 2018*). Spliced and unspliced count matrices were assembled using the run_smartseq2 subcommand with repeat-masked ENSEMBL build 87. RNA velocity was calculated using cell kNN pooling with a gamma fit based on a 95% quantile, and velocities were overlaid on integrated UMAP visualizations using *velocyto*. Pseudotemporal trajectories were calculated using Monocle 3 (*Cao et al., 2019*). The SeuratWrappers package provided integration between Seurat's final integrated dimensional reduction and Monocle. The data was subset to include only cone-directed iCPs and L/M cones, then a principal graph was fit across the UMAP plot and a root cell chosen to represent the latest maturation point in the $OPN1LW^+$ late-maturing cone group. After assigning pseudotime values to each cell, genes that significantly changed as a function of pseudotime were identified and those with a correlation q-value of <0.05 with expression >0.5 in at least 5% of the cells present in the pseudotime were grouped into modules of co-regulation by performing UMAP and Louvain community analysis.

## Histology

### Fixation and cryosectioning of fetal retina

Retinae were procured and dissected in cold 1 x PBS. The cornea and lens were removed with a cut around the limbus of the iris leaving the front of the eye open. The tissue was submerged in ~25 ml of 4% paraformaldehyde (PFA) in 1 x TBS and placed at 4 °C on a rocker at low speed for ~16–18 hrs. The tissue was washed three times with 1 x PBS before incubation in 30% sucrose [sucrose plus 1 x PBS] overnight at 4 °C. A mold was made by cutting off the top 5 ml of a 50 ml conical tube and placed on dry ice. A solution of 1:2 OCT Compound:30% sucrose was mixed and centrifuged to remove bubbles. The mold was partially filled before adding the dissected eye and then covering fully. 10 µm frozen sections were cut at ~−24 °C with the front of the eye facing side on to the blade. Explanted retinae were fixed for 15 min in 4% PFA, washed with 1 x PBS, incubated in 30% sucrose for 15 min, after which the membrane was cut from its frame and transferred to a mold containing 1:2 OCT:30% sucrose and frozen.

### Immunohistochemical staining

For SYK and ARR3 co-staining, tissue sections were warmed at RT until dry and then washed in Coplin jars twice with 1 x PBS for 5 min at RT. Samples were permeabilized in 1 x PBS-T for 5 min, washed again in 1 x PBS twice for 5 min, blocked for 1 hr in blocking solution (1 x PBS, 0.1% Triton X-100, 5% normal donkey serum, 5% normal horse serum). Primary mouse anti-SYK and rabbit anti-ARR3 (LUMIF-hCAR) antibodies were mixed in blocking solution and applied to samples overnight (16–18 hr) at 4 °C. Slides were washed three times with 1 x PBS for 10 min each, probed with secondary antibodies (Key Resources Table) in blocking solution for 1 hr at RT, washed twice 1 x PBS for 15 min each, incubated in 1x PBS containing 10 µg/ml DAPI for 10 min, and mounted with Mowiol solution. Additional immunohistological staining of tissue sections and cells on coverslips was performed as described (*Singh et al., 2018*) but without initial EDTA wash.

### Combined in situ hybridization and immunohistological staining

Human retinal sections were prepared as above but using RNase-free reagents for all stages (PBS, TBS) and ultrapure sucrose (JT Baker, 4097–04) and cryostats cleaned with ELIMINase (Decon Labs,

Inc) and 70% ethanol prior to use. RNA FISH hybridization chain reaction probes were designed by and obtained from Molecular Instruments, Inc, with 20 20 bp single strand DNA probes per target RNA sequence except Tr-NRL exon 1 (4 probes), FL-NRL exons 1 and (8 probes), and *HOTAIRM1* (14 probes). The accession numbers for target sequences are shown in the Key Resources Table, except for *RP13-143G15.4*, which included regions of several isoforms for maximum gene coverage. The in situ protocol was performed with fluorescent hairpins and buffers from Molecular Instruments using the manufacturer's instructions for mammalian cells on a slide (https://files.molecularinstruments. com/MI-Protocol-RNAFISH-MammalianCellSlide-Rev7.pdf), beginning with a 4% PFA treatment after thawing and drying slides at RT for 10 min, with the following changes: samples were incubated in 70% ethanol overnight in a Coplin jar; probes were added to the final hybridization volume at 4 x the recommended concentration for tissue sections (2.4 pmol in 150 µl of hybridization buffer); tissues were pre-hybridized at 42 °C and then lowered to 37 °C once probes were added; and hairpins were used at a volume of 1 µl (6 pmol) in 50 µl amplification buffer per section. After removing the amplification solution and washing, samples were directly used in the immunofluorescence protocol. Briefly, samples were blocked in a basic RNase-free solution (1% BSA, 0.1% Triton X-100, 0.05% Tween-20 in PBS) for 1 hr and primary antibodies mixed in blocking solution and applied to samples overnight at 4 °C. Samples were washed three times for 10 min with 1 x PBS, then incubated with secondary antibodies in blocking solution for 40 min at RT. Samples were washed as above with DAPI in the third and final wash, and mounted with Mowiol before imaging using a Leica STELLARIS 5 inverted confocal microscope. For *GNAT2* and *NR2E3* FISH, samples were washed with DAPI in the third and final wash, mounted, and then imaged as above. After this first imaging, coverslips were removed, and slides were washed in PBS and then used in the immunofluorescence protocol with an alternate blocking solution (2.5% Horse, donkey, human sera, 1% BSA, 0.1% Triton X-100, 0.05% Tween-20 in TBS).

## Quantitation of FISH

For *NRL*, *CHRNA1*, and lncRNAs, QuPath *Bankhead et al., 2017* was used to identify nuclei, predict cell expansion, classify cells, and count FISH puncta. Apical retina regions containing RXRγ+ and/or NRL+ cells were visually selected for evaluation. The StarDist extension *Schmidt et al., 2018* was used to outline nuclei and cell expansion zones with deep learning model dsb2018_heavy_augment.pb on the DAPI image channel (threshold = 0.6, pixelSize = 0.15). Model results were reviewed and cells with partial, multiple, or overlapping nuclei were removed from consideration. FISH puncta were thresholded, then output, using the subcellular detection function expected spot size = 0.5 µm², min spot size = 0.2 µm², max spot size = 0.7 µm² (*NRL* isoforms) or 1 µm² (*CHRNA1* and lncRNAs) to capture the most puncta above background. Image positions were determined by tracing the retina apical edge with line segments in FIJI, identifying the closest point on the line to the midpoint of each image, then measuring the distance from the line starting position. For *GNAT2* and *NR2E3* FISH, images were analyzed in FIJI. The outer layer of cells was delimited manually based on the RXRγ+ cells and the number of RXRγ+ and/or *GNAT2*+ and/or *NR2E3*+ cells were manually counted, examining each color channel separately. For the inner layer, the number of *NR2E3*-RNA-positive cells and NR2E3-protein-positive cells was counted.

## *NRL* isoform analyses

### Proteasome inhibition of CHLA-VC-RB-31

$10^6$ CHLA-VC-RB31 cells *Stachelek et al., 2023* were cultured in 2 ml retina culture medium in a 24-well plate with or without 10 µM retinoic acid (Cayman Chemical 11017) and incubated at 37 °C in 5% $CO_2$. MG132 (Sigma-Aldrich 474790) was added at 23 hr to a final concentration of 10 µM and cells incubated for 4 hr. Cells were collected and centrifuged at 300 x *g* for 4 min, resuspended in 60 µl cold 1 x RIPA buffer (Millipore 20–188), 10% sodium dodecyl sulfate (SDS), protease and phosphatase inhibitors diluted per manufacturer instructions (Sigma-Aldrich 5892970001, 4906837001), incubated on ice for 30 min, then centrifuged at 4 °C for 20 min at 14,000 RPM. 15 µl of supernatant were mixed with 3 µl 6 X sample buffer (300 mM Tris, 60% glycerol, 12% SDS, 86 mM B-mercaptoethanol, 0.6 mg/ ml bromophenol blue), heated to 95 °C for 5 min, and separated on a 4–12% Bis-Tris gel (Invitrogen, NP0321) using 1xMOPS running buffer (Life Technologies NP0001). Protein was wet transferred to a PVDF membrane in Towbin transfer buffer (25 mM Tris, 192 mM glycine pH 8.3,10% methanol) at 20 V ON at RT. The membrane was blocked in 1xTBS-T containing 5% w/v nonfat dry milk for 1 hr, washed

with TBS-T three times for 5 min, incubated overnight in 0.1% dry milk TBS-T solution containing primary antibody at 4 °C with slow rocking, washed three times for 15 min in TBS-T, incubated with HRP-conjugated secondary antibody for 1 hr at RT in the 0.1% milk solution, washed six times for 15 min, incubated in chemiluminescent substrate (Thermo Fisher, 34094) and imaged.

## NRL isoform and reporter constructs

The pcDNA4-His-Max-C-Nrl (*Cheng et al., 2004*; gift of A. Swaroop) CMV promoter was replaced with EF1α promoter to produce pcDNA4-His-Max-C-EF1α-FL-NRL. Using In-Fusion (Takara Bio), the His-Max tag was removed to produce full-length NRL expression plasmid pCDNA4-C-EF1α-FL-NRL (primers #1,2). The His-Max tag plus first 417 bp of NRL N-terminal coding sequence were removed to produce truncated NRL expression plasmid pCDNA4-C-EF1α-Tr-NRL (primers #1,2,3,4). pcDNA4-C-EF1α-empty transfection carrier plasmid was made by removing NRL coding sequence from pCDNA4-C-EF1α-Fl-NRL (primers #5,6). pGL3-PDE6B-146-luc was produced by amplifying a 146 bp promoter region of *PDE6B Lerner et al., 2001* from H9 hESC genomic DNA and inserting into a pGL3-SV40 (Promega, primers #7,8), then removing the SV40 promoter (primers #9,10) to generate pGL3-PDE6B-146. The promoter-less pGL3-empty was produced by removing SV40 sequences from pGL3-SV40 (primers #9,11). pUltra-EGFP-P2A-Tr-NRL was produced by amplification of NRL coding sequence from pcDNA4-C-EF1α-FL-NRL and inserting into BamHI digested pUltra-EGFP (Addgene 24129; primers #12, 13). All primers are listed in the Key Resources Table.

## NRL expression and analysis

pcDNA4-C-EF1α-NRL constructs were transfected into HEK-293T cells. 4 µg plasmid in 1 ml DMEM (Life Technologies, 10313039) was combined with 1 ml DMEM plus 12 µg polyethylenimine (PEI MAX; MW 40,000, Polysciences 24765–1) and combined with $10^6$ trypsinized HEK-293T cells, plated into two wells of a 6-well plate with an additional 1 ml of DMEM plus 10% FBS, then cultured overnight in a 37 °C incubator. Media was replaced with fresh 293T media (DMEM, 10% FBS, 1% Penicillin-Streptomycin) the next day and after 2 additional days cells were collected, lysed, and protein quantitated via BCA assay (Thermo Scientific PI23227). 30 µg of each well was used for western blot as above, with membranes probed with antibody raised against the NRL N-terminus (Santa Cruz SC-374277) or against full-length NRL (R&D AF2945).

## Luciferase assays

575 ng of total DNA (50 ng reporter, 100 ng FL-NRL and/or 100–300 ng Tr-NRL expression vector, and the remainder pcDNA4-C-EF1α empty vector) in 5 µl DMEM was mixed with 2.3 µg PEI in another 5 µl DMEM (per condition, in triplicate) and then plated into one well of a white bottom 96-well cell culture plate with 50,000 NIH3T3 cells or 50,000 of HEK-293T cells in 100 µl 293T media. The luciferase assay was carried out 48 hr after transfection with the Nano-Glo Dual-Luciferase Reporter Assay System (Promega N1610) per manufacturer instructions. Luminescence was recorded using the Promega Glomax Multi+ Detection System.

## Truncated NRL overexpression in fetal retina

Concentrated pUltra-EGFP-NRL-205 and control pUltra-EGFP lentivirus were produced as described (*Lee and Cobrinik, 2020*). At 16–24 hr after HEK-293T transfection in 15 cm dishes, media was exchanged for 20 ml UltraCULTURE (Lonza Inc, BP12-725F) containing 1% HEPES (Sigma, 0887), 1% GlutaMax (Life Technologies 35050061), and 1% Penicillin-Streptomycin (Thermo Fisher Scientific, MT30002CI). After 60–64 hr, media was harvested, centrifuged at 3000 x *g* for 10 min at 4 °C, filtered through 0.45 µm PVDF flask filter, concentrated via tangential flow filtration using a MidiKros 20 cm 500 KD 0.5 mm column (D02-5500-05-S), with both input and final supernatant containers kept in ice. The concentrate was re-filtered using a 0.45 µm PVDF syringe filter (Millipore), stored in aliquots at –80 °C, and titered using a p24 ELISA kit (ZeptoMetrix 801002).

As described (*Singh et al., 2018*), fetal eyes were washed in 70% ethanol, washed three times in 1 x sterile PBS, and dissected in cold 1 x PBS. The cornea was removed and tissue cut ~one-third of the distance towards the posterior pole along our lines approximately equidistant and between attached tendons. The retina was removed in this flattened state after cutting the optic nerve, transferred to

a polytetrafluoroethylene culture insert (Millipore, PICM0RG50) with photoreceptor side down, and placed in a six-well plate with 1200 µl retinal culture medium. Retinae were infected as described, cultured for 7 days with half media changes every other day, then fixed in 4% PFA for 15 min, washed three times in 1 x PBS, equilibrated in sucrose (30% in PBS) for 15 min, and embedded and sectioned as above. Immunostaining and imaging were performed as above.

### *RB1* knockdown and SYK inhibitor treatment

FW18.5 retina was partially dissociated with papain, cultured overnight in retinal culture media at 37 °C in a six-well plate, then frozen in 10% DMSO solution and stored in liquid nitrogen. Samples were thawed and revived in retinal culture media overnight and infected with lentivirus carrying shRNAs targeting *RB1* (sh*RB1-733*; *Xu et al., 2014*) or a scrambled control (sh*SCR*) and diluted β-mercaptoethanol, insulin, glutamine, penicillin, and streptomycin to the final retina culture medium concentrations along with 5 µg/ml Polybrene, for 24 hr. After replacing 2/3 of media in each well with fresh media, cells were treated with GS-9876 (MedChemExpress, HY-109091) in DMSO at or with the same volume of DMSO vehicle control. After 12 days, cells were attached to poly-L-Lysine-coated coverslips for 3 hr, fixed in 4% PFA for 10 min, washed in 1 x PBS three times, and stored at –20 °C until immunocytological staining with antibodies to Ki67, RXRγ, and GFP (for YFP staining) as described (*Xu et al., 2014*).

### Statistical analysis

Statistical methods and packages are cited in the Methods text and figure legends. Statistical comparison between more than two groups of cells for gene expression, regulon activity, or exon count proportions used the non-parametric Kruskal-Wallis test followed by pairwise post-hoc Dunn tests with Benjamini-Hochberg correction. Mean expression levels of *NRL* and *RXRG* isoforms across cell clusters were evaluated by Welch's t-tests for groups with unequal variance; p-values for these tests were estimated using bootstrapping with 1,000,000 replications per comparison. Changes in FISH puncta counts were examined within each channel individually using zero-inflated negative binomial regression models with robust standard error terms. Distance across the sample tissue was used to model degenerate zero inflation. P values reported in the manuscript text and legends are Wald Chi-Squared tests that evaluate differences in count between inflection points. Analyses were performed using Microsoft Excel, the R statistical language in RStudio, or Stata SE v14.2.

## Acknowledgements

We thank Melissa L Wilson (USC Department of Preventive Medicine) and Family Planning Associates for assistance in obtaining fetal tissue, Tingting Yang and Rui Chen for sharing data, the Stem Cell Analytics Core Facility, FACS Core Facility, and Imaging Core Facility of The Saban Research Institute of Children's Hospital Los Angeles for technical support, and Biraj Mahato and Sumitha Bharathan for critical reading an earlier version of the manuscript. National Institutes of Health grant R01EY026661 (DC). National Institutes of Health grant R01CA137124 (DC). National Institutes of Health grant R01AG076956 (MAB). National Institutes of Health grant 5T32HD060549 to USC Department of Development, Stem Cells, and Regenerative Medicine (DWHS). USC Provost Fellowship (DWHS). Saban Research Institute of Children's Hospital Los Angeles fellowship (JB). Research to Prevent Blindness (unrestricted grant to USC Dept. of Ophthalmology). Larry and Celia Moh Foundation (DC). Neonatal Blindness Research Fund (DC). AB Reins Foundation (DC).

## Additional information

### Funding

| Funder | Grant reference number | Author |
| --- | --- | --- |
| National Eye Institute | R01EY026661 | David Cobrinik |
| National Cancer Institute | R01CA137124 | David Cobrinik |

| Funder | Grant reference number | Author |
| --- | --- | --- |
| National Institutes of Health | R01AG076956 | Michael A Bonaguidi |
| National Institutes of Health | 5T32HD060549 | Dominic WH Shayler |
| Research to Prevent Blindness | Unrestricted Grant to USC Department of Ophthalmology | Cheryl Mae Craft David Cobrinik |
| Saban Research Institute | Predoctoral Fellowship | Jinlun Bai |
| University of Southern California | Provost Fellowship | Dominic WH Shayler |
| Larry and Celia Moh Foundation | | David Cobrinik |
| Neonatal Blindness Research Fund | | David Cobrinik |
| AB Reins Foundation | | David Cobrinik |

The funders had no role in study design, data collection and interpretation, or the decision to submit the work for publication.

## Author contributions

Dominic WH Shayler, Conceptualization, Data curation, Formal analysis, Validation, Investigation, Visualization, Methodology, Writing – original draft, Writing – review and editing; Kevin Stachelek, Data curation, Software, Visualization, Methodology, Writing – review and editing; Linda Cambier, Validation, Investigation, Methodology, Writing – review and editing; Sunhye Lee, Jinlun Bai, Daniel J Weisenberger, Hardeep P Singh, Investigation, Methodology; Bhavana Bhat, Data curation, Software, Visualization, Methodology; Mark W Reid, Formal analysis; Jennifer G Aparicio, Investigation, Writing – review and editing; Yeha Kim, Zachary Fouladian, Investigation; Mitali Singh, Visualization; Maxwell Bay, Eamon K Doyle, Stephan G Erberich, Software; Matthew E Thornton, Brendan H Grubbs, Resources; Michael A Bonaguidi, Supervision, Writing – review and editing; Cheryl Mae Craft, Resources, Writing – review and editing; David Cobrinik, Conceptualization, Data curation, Supervision, Funding acquisition, Methodology, Writing – original draft, Project administration, Writing – review and editing

## Author ORCIDs

Bhavana Bhat http://orcid.org/0009-0000-0534-0942
Daniel J Weisenberger http://orcid.org/0000-0001-8303-2603
Matthew E Thornton https://orcid.org/0000-0002-1083-2703
Michael A Bonaguidi https://orcid.org/0000-0002-9295-4762
Cheryl Mae Craft http://orcid.org/0000-0002-5219-3444
David Cobrinik https://orcid.org/0000-0002-4478-2417

## Ethics

Human fetal samples were provided by Advanced Bioscience Resources, Inc, Alameda, CA, or collected under Institutional Review Board approval at the University of Southern California (protocol HS-13-00399), and Children's Hospital Los Angeles (protocol CHLA-14-00122). Following the patient decision for pregnancy termination, patients were offered the option of donation of the products of conception for research purposes, and those that agreed signed an informed consent.

Reviewer #1 (Public review): https://doi.org/10.7554/eLife.101918.3.sa1
Reviewer #2 (Public review): https://doi.org/10.7554/eLife.101918.3.sa2
Reviewer #3 (Public review): https://doi.org/10.7554/eLife.101918.3.sa3
Author response https://doi.org/10.7554/eLife.101918.3.sa4

## Additional files

### Supplementary files
Supplementary file 1. Source data Excel files for selected figures. (A) Differentially expressed genes for low resolution rod clusters. (B) Differentially expressed genes for low resolution cone clusters. (C) Differentially expressed genes for low resolution late- vs early-maturing L/M cone populations. (D) Differentially expressed genes in rods vs rod precursors in a 3' snRNA-seq dataset (data of Zuo et al. PMID 39117640). (E) Differentially expressed genes in cones vs cone precursors in a 3' snRNA-seq dataset data of Zuo et al. PMID 39117640. (F) SCENIC regulon specificity scores for low resolution clusters. (G) Differentially expressed genes for high resolution MG and RPC clusters. (H) SCENIC regulon specificity scores for high resolution clusters. (I) Pseudotime-correlated genes and gene modules. (J) Differentially expressed genes for low resolution early rod and L/M cone clusters.

MDAR checklist

### Data availability
All data needed to evaluate the conclusions in the paper are present in the paper and/or the Supplementary Materials. A web portal providing access to a Shiny app for manipulation of the presented full-length scRNA-seq data (including UMAP or tSNE display of clusters, gene or isoform expression, and sample metadata; cluster marker gene and violin plots; exon coverage plots; Ensembl isoform assignments) is available at https://docker.saban.chla.usc.edu/cobrinik/app/seuratApp/. The preprocessed Seurat Object for full-length scRNAseq (and for the above-linked Shiny app) is available at https://doi.org/10.5281/zenodo.15231489. The Seurat Object for 3' snRNA-seq was downloaded from CZ CELLxGENE Discover with accession code 5900dda8-2dc3-4770- b604084eac1c2c82. Single-cell RNA-sequencing files and processed final gene and transcript matrices are publicly available at the GEO database under accession number GSE207802.

The following datasets were generated:

| Author(s) | Year | Dataset title | Dataset URL | Database and Identifier |
|---|---|---|---|---|
| Shayler DW, Stachalek K, Lee S, Singh M, Thornton M, Grubbs B, Singh HP, Cobrinik D | 2023 | Single-cell gene expression profiles of human retinal progenitor cells and photoreceptor precursors from deep full-length scRNA-seq | https://www.ncbi.nlm.nih.gov/geo/query/acc.cgi?acc=GSE207802 | NCBI Gene Expression Omnibus, GSE207802 |
| Shayler DWH, Stachelek K, Cobrinik D | 2025 | Human fetal retina FL scRNA-seq processed Seurat object | https://doi.org/10.5281/zenodo.15231489 | Zenodo, 10.5281/zenodo.15231489 |

The following previously published dataset was used:

| Author(s) | Year | Dataset title | Dataset URL | Database and Identifier |
|---|---|---|---|---|
| Zuo Z, Cheng X, Ferdous S, Shao J, Li J, Bao Y, Li J, Lu J, Jacobo Lopez A, Wohlschlegel J, Prieve A, Thomas MG, Reh TA, Li Y, Moshiri A, Chen R | 2024 | Single Cell Multiome Atlas of the Human Fetal Retina | https://cellxgene.cziscience.com/collections/5900dda8-2dc3-4770-b604-084eac1c2c82 | CZ CELLxGENE Discover, 5900dda8-2dc3-4770-b604084eac1c2c82 |

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
