## [Editor Report · eLife Assessment]

In this **important** paper, the authors use single-cell RNA sequencing to understand post-mitotic cone and rod developmental states and identify cone-specific features that contribute to retinoblastoma genesis. The authors report findings that have practical implications for retinal development, gene expression, and cell fate specification. The evidence is **compelling** as the experimental design and analysis are exceptionally rigorous.

---

## [Referee Report · Reviewer #1 (Public review)]

Summary:

The authors have used full length single cell sequencing on a sorted population of human fetal retina to delineate expression patterns associated with the progression of progenitors to rod and cone photoreceptors. They find that rod.cone precursors contain a mix of rod/cone determinants, with a bias in both amounts and isoform balance likely deciding the ultimate cell fate. Markers of early rod/cone hybrids are clarified, and a gradient of lncRNAs is uncovered in maturing cones. Comparison of early rods and cones exposes an enriched MYCN regulon, as well as expression of SYK, which may contribute to tumor initiation in RB1 deficient cone precursors.

Strengths:

The insight into how cone and rod transcripts are mixed together at first is important and clarifies a long-standing notion in the field.

The discovery of distinct active vs inactive mRNA isoforms for rod and cone determinants is crucial to understand how cells make the decision to form one or the other cell type. This is only really possible with full length scRNAseq analysis.

New markers of subpopulations are also uncovered, such as CHRNA1 in rod/cone hybrids that seem to give rise to either rods or cones.

Regulon analyses provide insight into key transcription factor programs linked to rod or cone fates.

The gradient of lncRNAs in maturing cones is novel, and while the functional significance is unclear, it opens up a new line of questioning around photoreceptor maturation.

The finding that SYK mRNA is naturally expressed in cone precursors is novel, as previously it was assumed that SYK expression required epigenetic rewiring in tumors.

Weaknesses:

Functional data on many new hypothesis regarding potential players in cone genesis are not performed, but these are beyond the scope of the current work.

Validation of the SYK inhibitor data e.g. by genetic means, is not included, but the authors acknowledge this caveat throughout.

---

## [Referee Report · Reviewer #2 (Public review)]

Summary:

The authors used deep full-length single-cell sequencing to study the human photoreceptor development, with a particular emphasis on the characteristics of photoreceptors that may contribute to retinoblastoma.

Strengths:

This single-cell study captures gene regulation in photoreceptors across different developmental stages, defining post-mitotic cone and rod populations by highlighting their unique gene expression profiles through analyses such as RNA velocity and SCENIC. By leveraging full-length sequencing data, the study identifies differentially expressed isoforms of NRL and THRB in L/M cone and rod precursors, illustrating the dynamic gene regulation involved in photoreceptor fate commitment. Additionally, the authors performed high-resolution clustering to explore markers defining developing photoreceptors across the fovea and peripheral retina, particularly characterizing SYK's role in the proliferative response of cones in the RB loss background. The study provides an in-depth analysis of developing human photoreceptors, with the authors conducting thorough analyses using full-length single-cell RNA sequencing. The strength of the study lies in its design, which integrates single-cell full-length RNA-seq, long-read RNA-seq, and follow-up histological and functional experiments to provide compelling evidence supporting their conclusions. The model of cell type-dependent splicing for NRL and THRB is particularly intriguing. Moreover, the potential involvement of the SYK and MYC pathways with RB in cone progenitor cells aligns with previous literature, offering additional insights into RB development.

Weaknesses:

The manuscript feels somewhat unfocused, with a lack of a strong connection between the analysis of developing photoreceptors, which constitutes the bulk of the manuscript, and the discussion on retinoblastoma. Additionally, given the recent publication of several single-cell studies on developing human retina, it is important for the authors to cross-validate their findings and adjust their statements where appropriate.

Comments on revisions:

The authors have done quite thorough work addressing concerns raised by myself and other reviewers. The identification of unresolved developing state of rod/cone precursor cell is interesting and intriguing. I do not have much more to add.

---

## [Referee Report · Reviewer #3 (Public review)]

Summary:

The authors use high-depth, full-length scRNA-Seq analysis of fetal human retina to identify novel regulators of photoreceptor specification and retinoblastoma progression.

Strengths:

The use of high-depth, full-length scRNA-Seq to identify functionally important alternatively spliced variants of transcription factors controlling photoreceptor subtype specification, and identification of SYK as a potential mediator of RB1-dependent cell cycle reentry in immature cone photoreceptors.

Weaknesses:

Relatively minor. This is a technically strong and thorough study that is broadly useful to investigators studying retinal development and retinoblastoma.

Comments on revisions:

The authors have addressed all points raised in the review and considerably strengthened the manuscript. No additional changes are required.

---

## [Author Response]

The following is the authors’ response to the original reviews

**Public Reviews:**

**Reviewer #1 (Public review):**
Summary:The authors have used full-length single-cell sequencing on a sorted population of human fetal retina to delineate expression patterns associated with the progression of progenitors to rod and cone photoreceptors. They find that rod and cone precursors contain a mix of rod/cone determinants, with a bias in both amounts and isoform balance likely deciding the ultimate cell fate. Markers of early rod/cone hybrids are clarified, and a gradient of lncRNAs is uncovered in maturing cones. Comparison of early rods and cones exposes an enriched MYCN regulon, as well as expression of SYK, which may contribute to tumor initiation in RB1 deficient cone precursors.Strengths:(1) The insight into how cone and rod transcripts are mixed together at first is important and clarifies a long-standing notion in the field.(2) The discovery of distinct active vs inactive mRNA isoforms for rod and cone determinants is crucial to understanding how cells make the decision to form one or the other cell type. This is only really possible with full-length scRNAseq analysis.(3) New markers of subpopulations are also uncovered, such as CHRNA1 in rod/cone hybrids that seem to give rise to either rods or cones.(4) Regulon analyses provide insight into key transcription factor programs linked to rod or cone fates.(5) The gradient of lncRNAs in maturing cones is novel, and while the functional significance is unclear, it opens up a new line of questioning around photoreceptor maturation.(6) The finding that SYK mRNA is naturally expressed in cone precursors is novel, as previously it was assumed that SYK expression required epigenetic rewiring in tumors.

We thank the reviewer for describing the study’s strengths, reflecting the major conclusions of the initially submitted manuscript. However, based on new analyses – including the requested analyses of other scRNA-seq datasets, our revision clarifies that:

- related to point (1), cone and rod transcripts do not appear to be mixed together at first (i.e., in immediately post-mitotic immature cone and rod precursors) but appear to be coexpressed in subsequent cone and rod precursor stages; and

- related to point (3), *CHRNA1* appears to mark immature cone precursors that are distinct from the maturing cone and rod precursors that co-express cone- and rod-related RNAs (despite the similar UMAP positions of the two populations in our dataset).

Weaknesses:(1) The writing is very difficult to follow. The nomenclature is confusing and there are contradictory statements that need to be clarified.(2) The drug data is not enough to conclude that SYK inhibition is sufficient to prevent the division of RB1 null cone precursors. Drugs are never completely specific so validation is critical to make the conclusion drawn in the paper.

We thank the reviewer for noting these important issues. Accordingly, in the revised manuscript:

(1) We improve the writing and clarify the nomenclature and contradictory statements, particularly those noted in the Reviewer’s Recommendations for Authors.

(2) We scale back claims related to the role of SYK in the cone precursor response to *RB1* loss, with wording changes in the Abstract, Results, and Discussion, which now recognize that the inhibitor studies only support the possibility that cone-intrinsic SYK expression contributes to retinoblastoma initiation, as detailed in our responses to Reviewer’s Recommendations for Authors. We agree and now mention that genetic perturbation of *SYK* is required to prove its role.

**Reviewer #2 (Public review):**
Summary:The authors used deep full-length single-cell sequencing to study human photoreceptor development, with a particular emphasis on the characteristics of photoreceptors that may contribute to retinoblastoma.Strengths:This single-cell study captures gene regulation in photoreceptors across different developmental stages, defining post-mitotic cone and rod populations by highlighting their unique gene expression profiles through analyses such as RNA velocity and SCENIC. By leveraging fulllength sequencing data, the study identifies differentially expressed isoforms of NRL and THRB in L/M cone and rod precursors, illustrating the dynamic gene regulation involved in photoreceptor fate commitment. Additionally, the authors performed high-resolution clustering to explore markers defining developing photoreceptors across the fovea and peripheral retina, particularly characterizing SYK's role in the proliferative response of cones in the RB loss background. The study provides an in-depth analysis of developing human photoreceptors, with the authors conducting thorough analyses using full-length single-cell RNA sequencing. The strength of the study lies in its design, which integrates single-cell full-length RNA-seq, longread RNA-seq, and follow-up histological and functional experiments to provide compelling evidence supporting their conclusions. The model of cell type-dependent splicing for NRL and THRB is particularly intriguing. Moreover, the potential involvement of the SYK and MYC pathways with RB in cone progenitor cells aligns with previous literature, offering additional insights into RB development.

We thank the reviewer for summarizing the main findings and noting the compelling support for the conclusions, the intriguing cell type-dependent splicing of rod and cone lineage factors, and the insights into retinoblastoma development.

Weaknesses:The manuscript feels somewhat unfocused, with a lack of a strong connection between the analysis of developing photoreceptors, which constitutes the bulk of the manuscript, and the discussion on retinoblastoma. Additionally, given the recent publication of several single-cell studies on the developing human retina, it is important for the authors to cross-validate their findings and adjust their statements where appropriate.

We agree that the manuscript covers a range of topics resulting from the full-length scRNAseq analyses and concur that some studies of developing photoreceptors were not well connected to retinoblastoma. However, we also note that the connection to retinoblastoma is emphasized in several places in the Introduction and throughout the manuscript and was a significant motivation for pursuing the analyses. We suggest that it was valuable to highlight how deep, fulllength scRNA-seq of developing retina provides insights into retinoblastoma, including (i) the similar biased expression of NRL transcript isoforms in cone precursors and RB tumors, (ii) the cone precursors’ co-expression of rod- and cone-related genes such as *NR2E3* and *GNAT2*, which may explain similar co-expression in RB cells, and (iii) the expression of *SYK* in early cones and RB cells. While the earlier version had mainly highlighted point (iii), the revised Discussion further refers to points (i) and (ii) as described further in the response to the Reviewer’s Recommendations for Authors.

We address the Reviewer’s request to cross-validate our findings with those of other single-cell studies of developing human retina by relating the different photoreceptor-related cell populations identified in our study to those characterized by Zuo et al (PMID 39117640), which was specifically highlighted by the reviewer and is especially useful for such cross-validation given the extraordinarily large ~ 220,000 cell dataset covering a wide range of retinal ages (pcw 8–23) and spatiotemporally stratified by macular or peripheral retina location. Relevant analyses of the Zuo et al dataset are shown in Supplementary Figures S3G-H, S10B, S11A-F, and S13A,B.

**Reviewer #3 (Public review):**
Summary:The authors use high-depth, full-length scRNA-Seq analysis of fetal human retina to identify novel regulators of photoreceptor specification and retinoblastoma progression.Strengths:The use of high-depth, full-length scRNA-Seq to identify functionally important alternatively spliced variants of transcription factors controlling photoreceptor subtype specification, and identification of SYK as a potential mediator of RB1-dependent cell cycle reentry in immature cone photoreceptors.Human developing fetal retinal tissue samples were collected between 13-19 gestational weeks and this provides a substantially higher depth of sequencing coverage, thereby identifying both rare transcripts and alternative splice forms, and thereby representing an important advance over previous droplet-based scRNA-Seq studies of human retinal development.Weaknesses:The weaknesses identified are relatively minor. This is a technically strong and thorough study, that is broadly useful to investigators studying retinal development and retinoblastoma.

We thank the reviewer for describing the strengths of the study. Our revision addresses the concerns raised separately in the Reviewer’s Recommendations for Authors, as detailed in the responses below.

**Recommendations for the authors:**

**Reviewing Editor Comments:**
The reviewers have completed their reviews. Generally, they note that your work is important and that the evidence is generally convincing. The reviewers are in general agreement that the paper adds to the field. The findings of rod/cone fate determination at a very early stage are intriguing. Generally, the paper would benefit from clarifications in the writing and figures. Experimentally, the paper would benefit from validation of the drug data, for example using RNAi or another assay. Alternatively, the authors could note the caveats of the drug experiments and describe how they could be improved. In terms of analysis, the paper would be improved by additional comparisons of the authors' data to previously published datasets.

We thank the reviewing editor for this summary. As described in the individual reviewer responses, we clarify the writing and figures and provide comparisons to previously published datasets in particular, the large snRNA-seq dataset of Zuo et al., 2024 (PMID 39117640). With regard to the drug (i.e., SYK inhibitor) studies, we opted to provide caveats and describe the need for genetic approaches to validate the role of SYK, owing to the infeasibility of completing genetic perturbation experiments in the appropriate timeframe. We are grateful for the opportunity to present our findings with appropriate caveats.

**Reviewer #1 (Recommendations for the authors):**
Shayler cell sort human progenitor/rod/cone populations then full-length single cell RNAseq to expose features that distinguish paths towards rods or cones. They initially distinguish progenitors (RPCs), immature photoreceptor precursors (iPRPs), long/medium wavelength (LM) cones, late-LM cones, short wavelength (S) cones, early rods (ER) and late rods (LR), which exhibit distinct transcription factor regulons (Figures 1, 2). These data expose expected and novel enriched genes, and support the notion that S cones are a default state lacking expression of rod (NRL) or cone (THRB) determinants but retaining expression of generic photoreceptor drivers (CRX/OTX2/NEUROD1 regulons). They identify changes in regulon activity, such as increasing NRL activity from iPRP to ER to LR, but decreasing from iPRP to cones, or increasing RAX/ISL2/THRB regulon activity from iPRP to LM cones, but decreasing from iPRP to S cones or rods.They report co-expression of rod/cone determinants in LM and ER clusters, and the ratios are in the expected directions (NRLTHRB or RXRG in ER). A novel insight from the FL seq is that there are differing variants generated in each cell population. Full-length NRL (FL-NRL) predominates in the rod path, whereas truncated NRL (Tr-NRL) does so in the cone path, then similar (but opposite) findings are presented for THRB (Fig 3, 4), whereas isoforms are not a feature of RXRG expression, just the higher expression in cones.The authors then further subcluster and perform RNA velocity to uncover decision points in the tree (Figure 5). They identify two photoreceptor precursor streams, the Transitional Rods (TRs) that provide one source for rod maturation and (reusing the name from the initial clustering) iPRPs that form cones, but also provide a second route to rods. TR cells closest to RPCs (immediately post-mitotic) have higher levels of the rod determinant NR2E3 and NRL, whereas the higher resolution iPRPs near RPCs lack NR2E3 and have higher levels of ONECUT1, THRB, and GNAT2, a cone bias. These distinct rod-biased TR and cone-biased high-resolution iPRPs were not evident in published scRNAseq with 3′ end-counting (i.e. not FL seq). Regulon analysis confirmed higher NRL activity in TR cells, with higher THRB activity in highresolution iPRP cells.Many of the more mature high-resolution iPRPs show combinations of rod (GNAT1, NR2E3) and cone (GNAT2, THRB) paths as well as both NRL and THRB regulons, but with a bias towards cone-ness (Figure 6). Combined FISH/immunofluorescence in fetal retina uncovers cone-biased RXRG-protein-high/NR2E3-protein-absent cone-fated cells that nevertheless expressed NR2E3 mRNA. Thus early cone-biased iPRP cells express rod gene mRNA, implying a rod-cone hybrid in early photoreceptor development. The authors refer to these as "bridge region iPRP cells".In Figure 7, they identify CHRNA1 as the most specific marker of these bridge cells (overlapping with ATOH7 and DLL3, previously linked to cone-biased precursors), and FISH shows it is expressed in rod-biased NRL protein-positive and cone-biased RXRG proteinpositive cones at fetal week 12.Figure 8 outlines the graded expression of various lncRNAs during cone maturation, a novel pattern.Finally (Figure 9), the authors identify differential genes expressed in early rods (ER cluster from Figure 1) vs early cones (LM cluster, excluding the most mature opsin+ cells), revealing high levels of MYCN targets in cones. They also find SYK expression in cones. SYK was previously linked to retinoblastoma, so intrinsic expression may predispose cone precursors to transformation upon RB loss. They finish by showing that a SYK inhibitor blocks the proliferation of dividing RB1 knockdown cone precursors in the human fetal retina.Overall, the authors have uncovered interesting patterns of biased expression in cone/rod developmental paths, especially relating to the isoform differences for NRL and THRB which add a new layer to our understanding of this fate choice. The analyses also imply that very soon after RPCs exit the cell cycle, they generate post-mitotic precursors biased towards a rod or cone fate, that carry varying proportions of mixed rod/cone determinants and other rod/cone marker genes. They also introduce new markers that may tag key populations of cells that precede the final rod/cone choice (e.g. CHRNA1), catalogue a new lncRNA gradient in cone maturation, and provide insight into potential genes that may contribute to retinoblastoma initiation, like SYK, due to intrinsic expression in cone precursors. However, as detailed below, the text needs to be improved considerably, and overinterpretations need to be moderated, removed, or tested more rigorously with extra data.Major CommentsThe manuscript is very difficult to follow. The nomenclature is at times torturous, and the description of hybrid rod/cone hybrid cells is confusing in many aspects.(1) A single term, iPRP, is used to refer to an initial low-resolution cluster, and then to a subset of that cluster later in the paper.

We agree that using immature photoreceptor precursor (iPRP) for both high-resolution and lowresolution clusters was confusing. We kept this name for the low-resolution cluster (which includes both immature cone and immature rod precursors), renamed the high-resolution iPRP cluster immature cone precursors (iCPs). and renamed their transitional rod (TR) counterparts immature rod precursors (iRPs). These designations are based on

- the biased expression of *THRB*, *ONECUT1*, and the THRB regulon in iCPs (Fig. 5D,E);

- the biased expression of *NRL*, *NR2E3*, and NRL regulon iRPs (Fig. 5D,E);

- the partially distinct iCP and iRP UMAP positions (Figure 5C); and

- the evidence of similar immature cone versus rod precursor populations in the Zuo et al 3’ snRNA-seq dataset, as noted below and described in two new paragraphs starting at the bottom of p. 12.

(2) To complicate matters further, the reader needs to understand the subset within the iPRP referred to as bridge cells, and we are told at one point that the earliest iPRPs lack NR2E3, then that they later co-express NR2E3, and while the authors may be referring to protein and RNA, it serves to further confuse an already difficult to follow distinction. I had to read and re-read the iPRP data many times, but it never really became totally clear.

We agree that the description of the high-resolution iPRP (now “iCP”) subsets was unclear, although our further analyses of a large 3’ snRNA-seq dataset in Figure S11 support the impression given in the original manuscript that the earliest iCPs lack *NR2E3* and then later coexpress *NR2E3* while the earliest iRPs lack *THRB* and then later express *THRB*. As described in new text in the Two post-mitotic immature photoreceptor precursor populations section (starting on line 7 of p. 13):

When considering only the main cone and rod precursor UMAP regions, early (pcw 8 – 13) cone precursors expressed *THRB* and lacked *NR2E3* (Figure S11D,E, blue arrows), while early (pcw 10 – 15) rod precursors expressed NR2E3 and lacked THRB (Figure S11D,E, red arrows), similar to RPC-localized iCPs and iRPs in our study (Figure 5D).

Next, as summarized in new text in the Early cone and rod precursors with rod- and conerelated RNA co-expression section (new paragraph at top of p. 16):

Thus, a 3’ snRNA-seq analysis confirmed the initial production of immature photoreceptor precursors with either L/M cone-precursor-specific *THRB* or rod-precursor-specific *NR2E3* expression, followed by lower-level co-expression of their counterparts, *NR2E3* in cone precursors and *THRB* in rod precursors. However, in the Zuo et al. analyses, the co-expression was first observed in well-separated UMAP regions, as opposed to a region that bridges the early cone and early rod populations in our UMAP plots. These findings are consistent with the notion that cone- and rod-related RNA co-expression begins in already fate-determined cone and rod precursors, and that such precursors aberrantly intermixed in our UMAP bridge region due to their insufficient representation in our dataset.

Importantly, and as noted in our ‘Public response’ to Reviewer 1, “*CHRNA1* appears to mark immature cone precursors that are distinct from the maturing cone and rod precursors that coexpress cone- and rod-related RNAs (despite the similar UMAP positions of the two populations in our dataset).” In support of this notion, the immature cone precursors expressing *CHRNA1* and other populations did not overlap in UMAP space in the Zuo et al dataset. We hope the new text cited above along with other changes will significantly clarify the observations.

(3) The term "cone/rod precursor" shows up late in the paper (page 12), but it was clear (was it not?) much earlier in this manuscript that cone and rod genes are co-expressed because of the coexpressed NRL and THRB isoforms in Figures 3/4.

We thank the reviewer for noting that the differential *NRL* and *THRB* isoform expression already implies that cone and rod genes are co-expressed. However, as we now state, the co-expression of RNAs encoding an additional cone marker (*GNAT2*) and rod markers (*GNAT1*, *NR2E3*) was

“suggestive of a proposed hybrid cone/rod precursor state more extensive than implied by the coexpression of different *THRB* and *NRL* isoforms” (first paragraph of “Early cone and rod …” section on p. 14; new text underlined).

(4) The (incorrect) impression given later in the manuscript is that the rod/cone transcript mixture applies to just a subset of the iPRP cells, or maybe just the bridge cells (writing is not clear), but actually, neither of those is correct as the more abundant and more mature LM and ER populations analyzed earlier coexpress NRL and THRB mRNAs (Figures 2, 3). Overall, the authors need to vastly improve the writing, simplify/clarify the nomenclature, and better label figures to match the text and help the reader follow more easily and clearly. As it stands, it is, at best, obtuse, and at worst, totally confusing.

We thank the reviewer for bringing the extent of the confusing terminology and wording to our attention. We revised the terminology (as in our response to point 1) and extensively revised the text. We also performed similar analyses of the Zuo et al. data (as described in more detail in our response to Reviewer 2), which clarifies the distinct status of cells with the “rod/cone transcript mixture” and cells co-expressing early cone and rod precursor markers.

To more clearly describe data related to cells with rod- and cone-related RNA co-expression, we divided the former Figure 6 into two figures, with Figure 6 now showing the cone- and rodrelated RNA co-expression inferred from scRNA-seq and Figure 7 showing *GNAT2* and *NR2E3* co-expression in FISH analyses of human retina plus a new schematic in the new panel 7E.

To separate the conceptually distinct analyses of cone and rod related RNA co-expression and the expression of early photoreceptor precursor markers (which were both found in the so-called bridge region – yet now recognized to be different subpopulations), we separated the analyses of the early photoreceptor precursor markers to form a new section, “Developmental expression of photoreceptor precursor markers and fate determinants,” starting on p. 16.

Additionally, we further review the findings and their implications in four revised Discussion paragraphs starting at the bottom of p. 23.

(5) The data showing that overexpressing Tr-NRL in murine NIH3T3 fibroblasts blocks FL-NRL function is presented at the end of page 7 and in Figure 3G. Subsequent analysis two paragraphs and two figures later (end page 8, Figure 5C + supp figs) reveal that Tr-NRL protein is not detectable in retinoblastoma cells which derive from cone precursors cells and express Tr-NRL mRNA, and the protein is also not detected upon lentiviral expression of Tr-NRL in human fetal retinal explants, suggesting it is unstable or not translated. It would be preferable to have the 3T3 data and retinoblastoma/explant data juxtaposed. E.g. they could present the latter, then show the 3T3 that even if it were expressed (e.g. briefly) it would interfere with FL-NRL. The current order and spacing are somewhat confusing.

We thank the reviewer for this suggestion and moved the description of the luciferase assays to follow the retinoblastoma and explant data and switched the order of Figure panels 3G and 3H.

(6) On page 15, regarding early rod vs early cone gene expression, the authors state: "although MYCN mRNA was not detected....", yet on the volcano plot in Figure S14A MYCN is one of the marked genes that is higher in cones than rods, meaning it was detected, and a couple of sentences later: "Concordantly, the LM cluster had increased MYCN RNA". The text is thus confusing.

With respect, we note that the original text read, “although *MYC* RNA was not detected,” which related to a statement in the previous sentence that the gene ontology analysis identified “MYC targets.” However, given that this distinction is subtle and may be difficult for readers to recognize, we revised the text (now on p. 19) to more clearly describe expression of *MYCN* (but not *MYC*) as follows:

“The upregulation of MYC target genes was of interest given that many MYC target genes are also targets of MYCN, that MYCN protein is highly expressed in maturing (ARR3^+^) cone precursors but not in NRL^+^ rods (Figure 10A), and that MYCN is critical to the cone precursor proliferative response to pRB loss8–10. Indeed, whereas *MYC* RNA was not detected, the LM cone cluster had increased *MYCN* RNA …”

(7) The authors state that the SYK drug is "highly specific". They provide no evidence, but no drug is 100% specific, and it is possible that off-target hits are important for the drug phenotype. This data should be removed or validated by co-targeting the SYK gene along with RB1.

We agree that our data only show *the potential* for SYK to contribute to the cone proliferative response; however, we believe the inhibitor study retains value in that a negative result (no effect of the SYK inhibitor) would disprove its potential involvement. To reflect this, we changed wording related to this experiment as follows:

In the Abstract, we changed:

(1) “*SYK*, which contributed to the early cone precursors’ proliferative response to *RB1* loss” To: “*SYK*, which was implicated in the early cone precursors’ proliferative response to *RB1* loss.”

(2) “These findings reveal … and a role for early cone-precursor-intrinsic *SYK* expression.” To: “These findings reveal … and suggest a role for early cone-precursor-intrinsic *SYK* expression.”

In the last paragraph of the Results, we changed:

(1) “To determine if SYK contributes…” To: “To determine if SYK might contribute…”

(2) “the highly specific SYK inhibitor” To: “the selective SYK inhibitor”

(3) “indicating that cone precursor intrinsic SYK activity is critical to the proliferative response” To: “consistent with the notion that cone precursor intrinsic SYK activity contributes to the proliferative response.”

In the Results, we added a final sentence:

“However, given potential SYK inhibitor off-target effects, validation of the role of SYK in retinoblastoma initiation will require genetic ablation studies.”

In the Discussion (2nd-to-last paragraph), we changed:

“SYK inhibition impaired pRB-depleted cone precursor cell cycle entry, implying that native SYK expression rather than de novo induction contributes to the cone precursors’ initial proliferation.” To: “…the pRB-depleted cone precursors’ sensitivity to a SYK inhibitor suggests that native SYK expression rather than de novo induction contributes to the cone precursors’ initial proliferation, although genetic ablation of *SYK* is needed to confirm this notion.” In the Discussion last sentence, we changed:

“enabled the identification of developmental stage-specific cone precursor features that underlie retinoblastoma predisposition.” To: “enabled the identification of developmental stage-specific cone precursor features that are associated with the cone precursors’ predisposition to form retinoblastoma tumors.”

Minor/TyposFigure 7 legend, H should be D.

We corrected the figure legend (now related to Figure 8).

**Reviewer #2 (Recommendations for the authors):**
(1) The author should take advantage of recently published human fetal retina data, such as PMID:39117640, which includes a larger dataset of cells that could help validate the findings. Consequently, statements like "To our knowledge, this is the first indication of two immediately post-mitotic photoreceptor precursor populations with cone versus rod-biased gene expression" may need to be revised.

We thank the reviewer for noting the evidence of distinct immediately post-mitotic rod and cone populations published by others after we submitted our manuscript. In response, we omitted the sentence mentioned and extensively cross-checked our results including:

- comparison of our early versus late cone and rod maturation states to the cone and rod precursor versus cone and rod states identified by Zuo et al (new paragraph on the top half of p. 6 and new figure panels S3G,H);

- detection of distinct immediately post-mitotic versus later cone and rod precursor populations (two new paragraphs on pp. 12-13 and new Figures S10B and S11A-E);

- identification of cone and rod precursor populations that co-express cone and rod marker genes (two new paragraphs starting at the bottom of p. 15 and new Figures S11D-F);

- comparison of expression patterns of immature cone precursor (iCP) marker genes in our and the Zuo et al dataset (new paragraph on top half of p. 17 and new Figure S13).

We also compare the cell states discerned in our study and the Zuo et al. study in a new Discussion paragraph (bottom of p. 23) and new Figure S17.

(2) The data generated comes from dissociated cells, which inherently lack spatial context. Additionally, it is unclear whether the dataset represents a pool of retinas from multiple developmental stages, and if so, whether the developmental stage is known for each cell profiled. If this information is available, the authors should examine the distribution of developmental stages on the UMAP and trajectory analysis as part of the quality control process.

We thank the reviewer for highlighting the importance of spatial context and developmental stage.

Related to whether the dataset represents a pool of retinae from multiple developmental stages, the different cell numbers examined at each time point are indicated in Figure S1A. To draw the readers’ attention to this detail, Figure S1A is now cited in the first sentence of the Results.

Related to the age-related cell distributions in UMAP plots, the distribution of cells from each retina and age was (and is) shown in Fig. S1F. In addition, we now highlight the age distributions by segregating the FW13, FW15-17, and FW17-18-19 UMAP positions in the new Figure 1C. We describe the rod temporal changes in a new sentence at the top of p. 5:

“Few rods were detected at FW13, whereas both early and late rods were detected from FW15-19 (Figure 1C), corroborating prior reports [15,20].”

We describe the cone temporal changes and note the likely greater discrimination of cell state changes that would be afforded by separately analyzing macula versus peripheral retina at each age in a new sentence at the bottom of p. 5:

“L/M cone precursors from different age retinae occupied different UMAP regions, suggesting age-related differences in L/M cone precursor maturation (Figure 1C).”

Moreover, they should assess whether different developmental stages impact gene expression and isoform ratios. It is well established that cone and rod progenitors typically emerge at different developmental times and in distinct regions of the retina, with minimal physical overlap. Grouping progenitor cells based solely on their UMAP positioning may lead to an oversimplified interpretation of the data.

(2a) We agree that different developmental stages may impact gene expression and isoform ratios, and evaluated stages primarily based on established Louvain clustering rather than UMAP position. However, we also used UMAP position to segregate so-called RPC-localized and nonRPC-localized iCPs and iRPs, as well as to characterize the bridge region iCP sub-populations. In the revision, we examine whether cell groups defined by UMAP positions helped to identify transcriptomically distinct populations and further examine the spatiotemporal gene expression patterns of the same genes in the Zuo et al. 3’ snRNA-seq dataset.

(2b) Related to analyses of immediately post-mitotic iRPs and iCPs, the new Figure S10A expanded the violin plots first shown in Figure 5D to compare gene expression in RPC-localized versus non-RPC-localized iCPs and iRPs and subsequent cone and rod precursor clusters (also presented in response to Reviewer 3). The new Figure S10C, shows a similar analysis of UMAP region-specific regulon activities. These figures support the idea that there are only subtle UMAP region-related differences in the expression of the selected gene and regulons.

To further evaluate early cone and rod precursors, we compared expression patterns in our cluster- and UMAP-defined cell groups to those of the spatiotemporally defined cell groups in the Zuo et al. 3’ snRNA-seq study. The results revealed similar expression timing of the genes examined, although the cluster assignments of a subset of cells were brought into question, especially the assigned rod precursors at pcw 10 and 13, as shown in new Figures S10B (grey columns) and S11, and as described in two new paragraphs starting near the bottom of p.12.

(2c) Related to analyses of iCPs in the so-called bridge region, our analyses of the Zuo et al dataset helped distinguish early cone and rod precursor populations (expressing early markers such as *ATOH7* and *CHRNA1*) from the later stages exhibiting rod- and cone-related gene coexpression, which had intermixed in the UMAP bridge region in our dataset. Further parsing of early cone precursor marker spatiotemporal expression revealed intriguing differences as now described in the second half of a new paragraph at the top of p. 17, as follows:

“Also, different iCP markers had different spatiotemporal expression: *CHRNA1* and *ATOH7* were most prominent in peripheral retina with *ATOH7* strongest at pcw 10 and *CHRNA1* strongest at pcw 13; *CTC-378H22.2* was prominently expressed from pcw 10-13 in both the macula and the periphery; and *DLL3* and *ONECUT1* showed the earliest, strongest, and broadest expression (Figure S13B). The distinct patterns suggest spatiotemporally distinct roles for these factors in cone precursor differentiation.”

(3) I would commend the authors for performing a validation experiment via RNA in situ to validate some of the findings. However, drawing conclusions from analyzing a small number of cells can still be dangerous. Furthermore, it is not entirely clear how the subclustering is done. Some cells change cell type identities in the high-resolution plot. For example, some iPRP cells from the low-resolution plots in Figure 1 are assigned as TR in high-resolution plots in Figure 5.The authors should provide justification on the identifies of RPC localized iPRP and TR.Comparison of their data with other publicly available data should strengthen their annotation

We agree that drawing conclusions from scRNA-seq or in situ hybridization analysis of a small number of cells can be dangerous and have followed the reviewer’s suggestion to compare our data with other publicly available data, focusing on the 3’ snRNA-seq of Zuo et al. given its large size and extensive annotation. Our analysis of the Zuo et al. dataset helped clarify cell identities by segregating cone and rod precursors with similar gene expression properties in distinct UMAP regions. However, we noted that the clustering of early cone and rod precursors likely gave numerous mis-assigned cells (as noted in response 2b above and shown in the new Figure S11). It would appear that insights may be derived from the combination of relatively shallow sequencing of a high number of cells *and* deep sequencing of substantially fewer cells.

Related to how subclustering was done, the Methods state, “A nearest-neighbors graph was constructed from the PCA embedding and clusters were identified using a Louvain algorithm at low and high resolutions (0.4 and 1.6)[70],” citing the Blondel et al reference for the Louvain clustering algorithm used in the Seurat package. To clarify this, the results text was revised such that it now indicates the levels used to cluster at low resolution (0.4, p. 4, 2nd paragraph) and at high resolution (1.6, top of p. 11) .

Related to the assignment of some iPRP cells from the low-resolution plots in Figure 1 to the TR cluster (now called the ‘iRP’ ‘cluster) in the high-resolution plots in Figure 5, we suggest that this is consistent with Louvain clustering, which does not follow a single dendrogram hierarchy.

The justification for referring to these groups as RPC-localized iCPs and iRPs relates to their biased gene and regulon expression in Fig. 5D and 5E, as stated on p. 12:

“In the RPC-localized region, iCPs had higher *ONECUT1*, *THRB*, and *GNAT2*, whereas iRPs trended towards higher NRL and NR2E3 (p = 0.19, p=0.054, respectively).”

(4) Late-stage LM5 cluster Figure 9 is not defined anywhere in previous figures, in which LM clusters only range from 1 to 4. The inconsistency in cluster identification should be addressed.

We revised the text related to this as follows:

“Indeed, our scRNA-seq analyses revealed that *SYK* RNA expression increased from the iCP stage through cluster LM4, in contrast to its minimal expression in rods (Figure 10E). Moreover, *SYK* expression was abolished in the five-cell group with properties of late maturing cones (characterized in Figure 1E), here displayed separately from the other LM4 cells and designated LM5 (Figure 10E).” (p. 19-20)

(5) Syk inhibitor has been shown to be involved in RB cell survival in previous studies. The manuscript seems to abruptly make the connection between the single-cell data to RB in the last figure. The title and abstract should not distract from the bulk of the manuscript focusing on the rod and cone development, or the manuscript should make more connection to retinoblastoma.

We appreciate the reviewer’s concern that the title may seem to over-emphasize the connection to retinoblastoma based solely on the SYK inhibitor studies. However, we suggest the title also emphasizes the identification and characterization of early human photoreceptor states, *per se*, and that there are a number of important connections beyond the SYK studies that could warrant the mention of cell-state-specific retinoblastoma-related features in the title.

Most importantly, a prior concern with the cone cell-of-origin theory was that retinoblastoma cells express RNAs thought to mark retinal cell types other than cones, especially rods. The evidence presented here, that cone precursors also express the rod-related genes helps resolve this issue. The issue is noted numerous times in the manuscript, as follows:

In the Introduction, we write:

“However, retinoblastoma cells also express rod lineage factor *NRL* RNAs, which – along with other evidence – suggested a heretofore unexplained connection between rod gene expression and retinoblastoma development[12,13]. Improved discrimination of early photoreceptor states is needed to determine if co-expression of rod- and cone-related genes is adopted during tumorigenesis or reflects the co-expression of such genes in the retinoblastoma cell of origin.” (bottom, p. 2) And:

“In this study, we sought to further define the transcriptomic underpinnings of human photoreceptor development and their relationship to retinoblastoma tumorigenesis.” (last paragraph, p. 3)

The Discussion also alluded to this issue and in the revised Discussion, we aimed to make the connection clearer. We previously ended the 3rd-to-last paragraph with,

“iPRP [now iCP] and early LM cone precursors’ expression of *NR2E3* and *NRL* RNAs suggest that their presence in retinoblastomas[12,13] reflects their normal expression in the L/M cone precursor cells of origin.”

We now separate and elaborate on this point in a new paragraph as follows:

“Our characterization of cone and rod-related RNA co-expression may help resolve questions about the retinoblastoma cell of origin. Past studies suggested that retinoblastoma cells co-express RNAs associated with rods, cones, or other retinal cells due to a loss of lineage fidelity[12]. However, the early L/M cone precursors’ expression of NR2E3 and NRL RNAs suggest that their presence in retinoblastomas[12,13] reflects their normal expression in the L/M cone precursor cells of origin. This idea is further supported by the retinoblastoma cells’ preferential expression of cone-enriched NRL transcript isoforms (Figure S5B).” (middle of p. 24) Based on the above, we elected to retain the title.

Minor comments:(1) It is difficult to see the orange and magenta colors in the Fig 3E RNA-FISH image. The colors should be changed, or the contrast threshold needs to be adjusted to make the puncta stand out more.

We re-assigned colors, with red for *FL-NRL* puncta and green for *Tr-NRL* puncta.

(2) Figure 5C on page 8 should be corrected to Supplementary Figure 5C.

We thank the reviewer for noting this error and changed the figure citation.

**Reviewer #3 (Recommendations for the authors):**
(1) Minor concernsa. Abbreviation of some words needs to be included, example: FW.

We now provide abbreviation definitions for FW and others throughout the manuscript.

b. Cat # does not matches with the 'key resource table' for many reagents/kits. Some examples are: CD133-PE mentioned on Page # 22 on # 71, SMART-Seq V4 Ultra Low Input RNA Kit and SMARTer Ultra Low RNA Kit for the Fluidigm C1 Sytem on Page # 22 on # 77, Nextera XT DNA Library preparation kit on Page # 23 on # 77.

We thank the reviewer for noting these discrepancies. We have now checked all catalog numbers and made corrections as needed.

c. Cat # and brand name of few reagents & kits is missing and not mentioned either in methods or in key resource table or both. Eg: FBS, Insulin, Glutamine, Penicillin, Streptomycin, HBSS, Quant-iT PicoGreen dsDNA assay, Nextera XT DNA LibraryPreparation Kit, 5' PCR Primer II A with CloneAmp HiFi PCR Premix.

Catalog numbers and brand names are now provided for the tissue culture and related reagents within the methods text and for kits in the Key Resources Table. Additional descriptions of the primers used for re-amplification and RACE were added to the Methods (p. 28-29).

d. Spell and grammar check is needed throughout the manuscript is needed. Example. In Page # 46 RXRγlo is misspelled as RXRlo.

Spelling and grammar checks were reviewed.

(2) Methods & Key Resource table.a. In Page # 21, IRB# needs to be stated.

The IRB protocols have been added, now at top of p. 26.

b. In Page # 21, Did the authors dissociate retinae in ice-cold phosphate-buffered saline or papain?

The relevant sentence was corrected to “dissected while submerged in ice-cold phosphatebuffered saline (PBS) and dissociated as described10.” (p. 26)

c. In Page # 21, How did the authors count or enumerate the cell count? Provide the details.

We now state, “… a 10 µl volume was combined with 10 µl trypan blue and counted using a hemocytometer” (top of p. 27)

d. Why did the authors choose to specifically use only 8 cells for cDNA preparation in Page # 22? State the reason and provide the details.

The reasons for using 8 cells (to prevent evaporation and to manually transfer one slide-worth of droplets to one strip of PCR tubes) and additional single cell collection details are now provided as follows (new text underlined):

“Single cells were sorted on a BD FACSAria I at 4°C using 100 µm nozzle in single-cell mode into each of eight 1.2 µl lysis buffer droplets on parafilm-covered glass slides, with droplets positioned over pre-defined marks … . Upon collection of eight cells per slide, droplets were transferred to individual low-retention PCR tubes (eight tubes per strip) (Bioplastics K69901, B57801) pre-cooled on ice to minimize evaporation. The process was repeated with a fresh piece of parafilm for up to 12 rounds to collect 96 cells. (p. 27, new text underlined)

e. Key resource table does not include several resources used in this study. Example - NR2E3 antibody.

We added the NR2E3 antibody and checked for other omissions.

(3) Results & Figures & Figure Legendsa. Regulon-defined RPC and photoreceptor precursor statesi. On page # 4, 1 paragraph - Clarify the sentence 'Exclusion of all cells with <100,000 cells read and 18 cells.........Emsembl transcripts inferred'. Did the authors use 18 cells or 18FW retinae?

The sentence was changed to:

“After sequencing, we excluded all cells with <100,000 read counts and 18 cells expressing one or more markers of retinal ganglion, amacrine, and/or horizontal cells (*POU4F1, POU4F2, POU4F3, TFAP2A, TFAP2B, ISL1*) and concurrently lacking photoreceptor lineage marker *OTX2*. This yielded 794 single cells with averages of 3,750,417 uniquely aligned reads, 8,278 genes detected, and 20,343 Ensembl transcripts inferred (Figure S1A-C).” (p. 4, new words underlined)

To clarify that 18 retinae were used, the first sentence of the Results was revised as follows:

“To interrogate transcriptomic changes during human photoreceptor development, dissociated RPCs and photoreceptor precursors were FACS-enriched from 18 retinae, ages FW13-19 …” (p. 4).

Why did the authors 'exclude cells lacking photoreceptor lineage marker OTX2' from analysis especially when the purpose here was to choose photoreceptor precursor states & further results in the next paragraph clearly state that 5 clusters were comprised of cells with OTX2 and CRX expression. This is confusing.

We apologize for the imprecise diction. We divided the evidently confusing sentence into two sentences to more clearly indicate that we removed cells that did not express *OTX2*, as in the first response to the previous question.

ii. In Page # 5, the authors reported the number of cell populations (363 large and 5 distal) identified in the THRB+ L/M-cone cluster. What were the # of cell populations identified in the remaining 5 clusters of the UMAP space?

We added the cell numbers in each group to Fig. 1B. We corrected the large LM group to 366 cells (p. 5) and note 371 LM cells , which includes the five distal cells, in Figure 1B.

b. Differential expression of NRL and THRB isoforms in rod and cone precursorsi. In Figure 3B, the authors compare and show the presence of 5 different NRL isoforms for all the 6 clusters that were defined in 3A. However, in the results, the ENST# of just 2 highly assigned transcript isoforms is given. What are the annotated names of the three other isoforms which are shown in 3B? Please explain in the Results.

As requested, we now annotate the remaining isoforms as encoding full-length or truncated NRL in Fig. 3B and show isoform structures in new Supplementary Figure S4B. We also refer to each transcript isoform in the Results (p. 7, last paragraph) and similarly evaluate all isoforms in RB31 cells (Fig. S5B).

ii. What does the Mean FPM in the y-axis of Fig 3C refer to?

Mean FPM represents mean read counts (fragments per million, FPM) for each position across Ensembl NRL exons for each cluster, as now stated in the 6th line of the Fig. 3 legend.

iii. A clear explanation of the results for Figures 3E-3F is missing.

We revised the text to more clearly describe the experiment as follows:

“The cone cells’ higher proportional expression of Tr-NRL first exon sequences was validated by RNA fluorescence in situ hybridization (FISH) of FW16 fetal retina in which NRL immunofluorescence was used to identify rod precursors, RXRg immunofluorescence was used to identify cone precursors, and FISH probes specific to truncated Tr-NRL exon 1T or FL-NRL exons 1 and 2 were used to assess Tr-NRL and FL-NRL expression (Figure 3E,F).” (p. 8, new text underlined).

c. Two post-mitotic photoreceptor precursor populationsi. Although deep-sequencing and SCENIC analysis clarified the identities of four RPC-localized clusters as MG, RPC, and iPRP indicative of cone-bias and TR indicative of rod-bias. It would be interesting to see the discriminating determinant between the TR and ER by SCENIC and deep-sequencing gene expression violin/box plots.

We agree it is of interest to see the discriminating determinant between the TR [now termed *iRP*] and ER clusters by SCENIC and deep-sequencing gene expression violin/box plots. We now provide this information for selected genes and regulons of interest in the new Supplementary Figures S10A and S10C, along with a similar comparison between the prior high-resolution iPRP (now termed *iCP*) cluster and the first high-resolution LM cluster, LM1, as described for gene expression on p. 12:

“Notably, *THRB* and *GNAT2* expression did not significantly change while *ONECUT1* declined in the subsequent non-RPC-localized iCP and LM1 stages, whereas *NR2E3* and *NRL* dramatically increased on transitioning to the ER state (Figure S10A).”

And as described for regulon activities on pp. 13-14:

“Finally, activities of the cone-specific THRB and ISL2 regulons, the rod-specific NRL regulon, and the pan-photoreceptor LHX3, OTX2, CRX, and NEUROD1 regulons increased to varying extents on transitioning from the immature iCP or iRP states to the early-maturing LM1 or ER states (Figure 10C).”

We also show expression of the same genes for spatiotemporally grouped cells from the Zuo et al. dataset in the new Figure S10B, which displays a similar pattern (apart from the possibly mixed pcw 10 and pcw13 designated rod precursors).

d. Early cone precursors with cone- and rod-related RNA expressioni. On page #12, the last paragraph where the authors explain the multiplex RNA FISH results of RXRγ and NR2E3 by citing Figure S8E. However, in Fig S8E, the authors used NRL to identify the rods. Please clarify which one of the rod markers was used to perform RNA FISH?

Figure S8E (where NRL was used as a rod marker) was cited to remind readers that RXRg has low expression in rods and high expression in cones, rather than to describe the results of this multiplex FISH section. To avoid confusion on this point, Figure S8E is now cited using “(as earlier shown in Figure S8E).” With this issue clarified, we expect the markers used in the FISH + IF analysis will be clear from the revised explanation,

“… we examined *GNAT2* and *NR2E3* RNA co-expression in RXRg+ cone precursors in the outermost NBL and in RXRg+ rod precursors in the middle NBL … .” (p. 14-15).

To provide further clarity, we provide a diagram of the FISH probes, protein markers, and expression patterns in the new Figure 7E.

ii. The Y-axis of Fig 6G-6H needs to be labelled.

The axes have been re-labeled from “Nb of cells” to “Number of RXRg+ outermost NBL cells in each region” (original Fig. 6G, now Fig. 7C) and “Number of RXRg+ middle NBL cells in each region” (original Fig. 6H, now Fig. 7D).

iii. The legends of Figures 6G and 6H are unclear. In the Figure 6G legend, the authors indicate 'all cells are NR2E3 protein-'. Does that imply the yellow and green bars alone? Similarly, clarify the Figure 6H legend, what does the dark and light magenta refer to? What does the light magenta color referring to NR2E3+/ NR2E3- and the dark magenta color referring to NR2E3+/ NR2E3+ indicate?

We regret the insufficient clarity. We revised the Fig. 6G (now Fig. 7C) key, which now reads

“All outermost NBL cells are NR2E3 protein-negative.” We added to the figure legend for panel 7C,D “(*n.b.*, italics are used for RNAs, non-italics for proteins).” The new scheme in Figure 7E shows the RNAs in italics proteins in non-italics. We hope these changes will clarify when RNA or protein are represented in each histogram category.

Overall, the results (on page # 13) reflecting Figures 6E-6H & Figure S11 are confusing and difficult to understand. Clear descriptions and explanations are needed.

We revised this results section described in the paragraph now spanning p. 14:

- We now refer to the bar colors in Figures 7C and 7D that support each statement.

- We provide an illustration of the findings in Figure 7E.

iv. Previously published literature has shown that cells of the inner NBL are RXRγ+ ganglion cells. So, how were these RXRγ+ ganglion cells in the inner NBL discriminated during multiplex RNA FISH (in Fig 6E-6H and in Fig S11)?

We thank the reviewer for requesting this clarification. We agree that “inner NBL” is the incorrect term for the region in which we examined RXRg+ photoreceptor precursors, as this could include RXRγ+ nascent RGCs. We now clarify that

“we examined *GNAT2* and *NR2E3* RNA co-expression in RXRg+ cone precursors in the outermost NBL and in RXRg+ rod precursors in the middle NBL … .” (p. 14-15) We further state,

“Limiting our analysis to the outer and middle NBL allowed us to disregard RXRγ+ retinal ganglion cells in the retinal ganglion cell layer or inner NBL (top of p. 15)”

Figure 7E is provided to further aid the reader in understanding the positions examined, and the legend states “RXRg+ retinal ganglion cells in the inner NBL and ganglion cell layer not shown.

v. In Figure 6E, what marker does each color cell correspond to?

In this figure (now panel 7A), we declined to provide the color key since the image is not sufficiently enlarged to visualize the IF and FISH signals. The figure is provided solely to document the regions analyzed and readers are now referred to “see Figure S12 for IF + FISH images” (2nd line, p. 15), where the marker colors are indicated.

vi. In Figure S11 & 6E, Protein and RNA transcript color of NR2E3, GNAT2 are hard to distinguish. Usage of other colors is recommended.

We appreciate the reviewer’s concern related to the colors (in the now redesignated Figure S12 and 7A); however, we feel this issue is largely mitigated by our use of arrows to point to the cells needed to illustrate the proposed concepts in Figure S12B. All quantitation was performed by examining each color channel separately to ensure correct attribution, which is now mentioned in the Methods (2nd-to-last line of Quantitation of FISH section, p. 35).

vii.

With due respect, we suggest that labeling each box (now in Figure 8B) makes the figure rather busy and difficult to infer the main point, which is that boxed regions were examined at various distanced from the center (denoted by the “C” and “0 mm”) with distances periodically indicated. We suggest the addition of such markers would not improve and might worsen the figure for most readers.

e. An early L/M cone trajectory marked by successive lncRNA expressioni. In Figure 8C - color-coded labelling of LM1-4 clusters is recommended.

We note Fig. 8C (now 9C) is intended to use color to display the pseudotemporal positions of each cell. We recognize that an additional plot with the pseudotime line imposed on LM subcluster colors could provide some insights, yet we are unaware of available software for this and are unable to develop such software at present. To enable readers to obtain a visual impression of the pseudotime vs subcluster positions, we now refer the reader to Figure 5A in the revised figure legend, as follows: (“The pseudotime trajectory may be related to LM1-LM4 subcluster distributions in Figure 5A.”).

ii. In Figure 8G - what does the horizontal color-coded bar below the lncRNAs name refer to? These bars are similar in all four graphs of the 8G figure.

As stated in the Fig. 8G (now 9G) legend, “Colored bars mark lncRNA expression regions as described in the text.” We revised the text to more clearly identify the color code. (p. 18-19)

f. Cone intrinsic SYK contributions to the proliferative response to pRB lossi. In Fig 9F - The expression of ARR3^+^ cells (indicated by the green arrow in FW18) is poorly or rarely seen in the peripheral retina.

We thank the reviewer for finding this oversight. In panel 9F (now 10F), we removed the green arrows from the cells in the periphery, which are ARR3- due to the immaturity of cones in this region.

ii. In Figure 9F - Did the authors stain the FW16 retina with ARR3?

Unfortunately, we did not stain the FW16 retina for ARR3 in this instance.

iii. Inclusion of DAPI staining for Fig 9F is recommended to justify the ONL & INL in the images.

We regret that we are unable to merge the DAPI in this instance due to the way in which the original staining was imaged. A more detailed analysis corroborating and extending the current results is in progress.

iv. Immunostaining images for Figure 9G are missing & are required to be included. What does shSCR in Fig 9G refer to?

We now provide representative immunostaining images below the panel (now 10G). The legend was updated: “Bottom: Example of Ki67, YFP, and RXRg co-immunostaining with DAPI^+^ nuclei (yellow outlines). Arrows: Ki67+, YFP+, RXRg+ nuclei.” The revised legend now notes that shSCR refers to the scrambled control shRNA.

v. For Figure 9H - Is the presence and loss of SYK activity consistent with all the subpopulations (S & LM) of early maturing and matured cones?

We appreciate the reviewer’s question and interest (relating to the redesignated Figure 10H); however, we have not yet completed a comprehensive evaluation of SYK expression in all the subpopulations (S & LM) of early maturing and matured cones and will reserve such data for a subsequent study. We suggest that this information is not critical to the study’s major conclusions.

vi. Figure 9A is not explained in the results. Why were MYCN proteins assessed along with ARR3 and NRL? What does this imply?

We thank the reviewer for noting that this figure (now Figure 10A) was not clearly described.

As per the response to Reviewer 1, point 6 , the text now states,

“The upregulation of MYC target genes was of interest given that many MYC target genes are also MYCN targets, that MYCN protein is highly expressed in maturing (ARR3^+^) cone precursors but not in NRL^+^ rods (Figure 10A), and that MYCN is critical to the cone precursor proliferative response to pRB loss [8–10].” (middle, p. 19, new text underlined).

Hence, the figure demonstrates the cone cell specificity of high MYCN protein. This is further noted in the Fig. 10a legend: “A. Immunofluorescent staining shows high MYCN in ARR3^+^ cones but not in NRL^+^ rods in FW18 retina.”